# Growth of complete ammonia oxidizers on guanidine

Marton Palatinszky[1], Craig W. Herbold[1,10], Christopher J. Sedlacek[1], Dominic Pühringer[2,3], Katharina Kitzinger[1], Andrew T. Giguere[1], Kenneth Wasmund[1,4,11], Per H. Nielsen[4], Morten K. D. Dueholm[4], Nico Jehmlich[5], Richard Gruseck[1,6], Anton Legin[7], Julius Kostan[2,3], Nesrete Krasnici[2,3], Claudia Schreiner[2,3], Johanna Palmetzhofer[1,6], Thilo Hofmann[1], Michael Zumstein[1], Kristina Djinović-Carugo[2,3,8,9], Holger Daims[1,8] & Michael Wagner[1,4,8 ✉]

Guanidine is a chemically stable nitrogen compound that is excreted in human urine and is widely used in manufacturing of plastics, as a flame retardant and as a component of propellants, and is well known as a protein denaturant in biochemistry[1–3]. Guanidine occurs widely in nature and is used by several microorganisms as a nitrogen source, but microorganisms growing on guanidine as the only substrate have not yet been identified. Here we show that the complete ammonia oxidizer (comammox) *Nitrospira inopinata* and probably most other comammox microorganisms can grow on guanidine as the sole source of energy, reductant and nitrogen. Proteomics, enzyme kinetics and the crystal structure of a *N. inopinata* guanidinase homologue demonstrated that it is a bona fide guanidinase. Incubation experiments with comammox-containing agricultural soil and wastewater treatment plant microbiomes suggested that guanidine serves as substrate for nitrification in the environment. The identification of guanidine as a growth substrate for comammox shows an unexpected niche of these globally important nitrifiers and offers opportunities for their isolation.

Recently, microbial guanidine metabolism has received a lot of attention, as the identification of four different classes of riboswitches (RNA elements that bind metabolites or metal ions as ligands and regulate mRNA expression) selective for guanidine in the genomes of many bacteria in various phyla suggested that guanidine is an important metabolite of microorganisms. However, the microbial pathways for the formation and degradation of this compound and its ecological importance largely remain to be explored[4–8]. During the past few years, bacterial guanidine production by at least three pathways has been demonstrated. Some bacteria encode an ethylene-forming enzyme that has an important role in bioethylene production and produces guanidine from arginine and 2-oxoglutarate[9]. Furthermore, during synthesis of the antibiotic naphthyridinomycin by *Streptomyces lusitanus*, the arginine-4,5-desaturase NapI leads to guanidine formation as a side reaction[10]. Moreover, bacteria can transform guanylurea to ammonia and guanidine[11]. This is particularly important as guanylurea is one of the most common contaminants in nature formed by degradation of the type 2 diabetes drug metformin (a biguanidine and one of the most prescribed drugs globally) and of the fertilizer additive cyanoguanidine (dicyandiamide)[11–13]. However, additional guanidine-forming mechanisms must exist in bacteria, as guanidine has been detected under nutrient-poor conditions in *Escherichia coli* lacking the aforementioned pathways[4]. Notably, in plants and algae, guanidine is also produced by homoarginine-6-hydroxylases[10,14].

Our knowledge of guanidine degradation by microorganisms is also still in its infancy, but recent research has revealed two degradation pathways for guanidine that are used by bacteria to not only detoxify guanidine but also to use it as a nitrogen source and, ultimately, to produce three molecules of ammonia from one molecule of guanidine. A widespread sequential decomposition pathway (Fig. 1a) involves a biotin-containing and ATP-dependent guanidine carboxylase, a heteromeric carboxyguanidine deiminase and an allophanate hydrolase to convert guanidine to ammonia and $CO_2$[15,16]. In 2021, a Ni-containing guanidinase, mediating the direct hydrolysis of free guanidine to urea (which is further converted by urease to ammonia and $CO_2$) and ammonia, was described in a *Synechocystis* strain (Fig. 1b). This guanidinase has a crucial role in detoxifying guanidine during bioethylene production and enables this cyanobacterium to tap the guanidine pool as a nitrogen source without spending ATP for its degradation[17,18].

## Guanidine use by nitrifying microorganisms

Chemolithoautotrophic nitrifying microorganisms catalyse the aerobic oxidation of ammonia through nitrite to nitrate and therefore have

[1]Centre for Microbiology and Environmental Systems Science, University of Vienna, Vienna, Austria. [2]Department of Structural and Computational Biology, Center for Molecular Biology, University of Vienna, Vienna, Austria. [3]Max Perutz Labs, Vienna Biocenter Campus (VBC), Vienna, Austria. [4]Center for Microbial Communities, Department of Chemistry and Bioscience, Aalborg University, Aalborg, Denmark. [5]Helmholtz-Centre for Environmental Research-UFZ, Department of Molecular Systems Biology, Leipzig, Germany. [6]Doctoral School in Microbiology and Environmental Science, University of Vienna, Vienna, Austria. [7]Institute of Inorganic Chemistry, Faculty of Chemistry, University of Vienna, Vienna, Austria. [8]The Comammox Research Platform, University of Vienna, Vienna, Austria. [9]European Molecular Biology Laboratory (EMBL), Grenoble, France. [10]Present address: Te Kura Pūtaiao Koiora (School of Biological Sciences), Te Whare Wānanga o Waitaha (University of Canterbury), Ōtautahi (Christchurch), Aotearoa New Zealand. [11]Present address: School of Biological Sciences, University of Portsmouth, Portsmouth, UK. ✉e-mail: michael.wagner@univie.ac.at

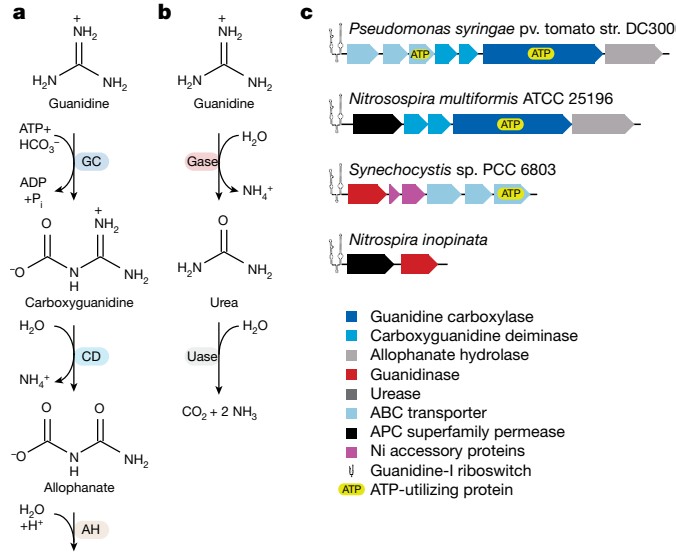

**Fig. 1 | Pathways and genes involved in guanidine degradation. a**, The guanidine carboxylase pathway. AH, allophanate hydrolase; CD, carboxyguanidine deaminase; Gase, guanidinase; GC, guanidine carboxylase; $P_i$, inorganic phosphate; Uase, urease. **b**, The guanidinase pathway. **c**, Arrangements of genes encoding proteins involved in guanidine degradation under the regulation of type I guanidine riboswitches in the comammox microorganism *N. inopinata*, the AOB *Nitrosospira multiformis* and two bacterial model organisms for guanidine catabolism. Each of the four genomes also encodes urease (not displayed) at locations that are not regulated by guanidine riboswitches.

an essential role in global biogeochemical nitrogen cycling, lead to massive fertilizer loss in agriculture, contribute to the emission of the potent greenhouse gas and ozone-depleting substance nitrous oxide, and are essential for nutrient removal in wastewater treatment plants (WWTPs)[19,20]. In addition to ammonia-oxidizing bacteria (AOB) and archaea (AOA), which convert ammonia to nitrite (that is subsequently oxidized to nitrate by nitrite-oxidizing bacteria), the recently identified complete ammonia oxidizers (comammox) of the genus *Nitrospira* use ammonia as a substrate and oxidize it through nitrite to nitrate[21,22]. In contrast to canonical nitrite oxidizers of the genus *Nitrospira*[23,24], the recognized range of energy sources used by ammonia-oxidizing microorganisms is very narrow. In addition to ammonia, only urea and cyanate have been experimentally confirmed to support the growth of some ammonia-oxidizing microorganisms by serving as an ammonia source after being converted enzymatically by cytoplasmic ureases and cyanases, respectively[25–27], although meta-omic studies of comammox strains in WWTPs suggested a more pronounced metabolic versatility[28].

We wondered whether ammonia-oxidizing microorganisms might be able to exploit guanidine as a source of energy, reductant and nitrogen. We therefore screened all available genomes from this guild for genes possibly encoding enzymes involved in the two known pathways that would enable guanidine to be used as an ammonia source (Fig. 1 and Supplementary Table 1). We found genes for guanidine degradation in most betaproteobacterial AOB (135 out of 145 genomes) and comammox strains (76 out of 83 genomes) but none within the gammaproteobacterial AOB or the various members of the AOA. Notably, most ammonia oxidizers encoding enzymes involved in guanidine utilization also possess an amino acid/polyamine/organocation permease (APC superfamily) in the same genomic region, which is predicted to enable the import of guanidine without using ATP (possibly through proton symport; Fig. 1). By contrast, *Pseudomonas syringae* pv. tomato str. DC3000 and *Synechocystis* sp. PCC 6803 (the model organisms for the guanidine carboxylase and guanidinase pathways, respectively)

encode an ATP-dependent ABC transporter for guanidine transport in the neighbourhood of the genes encoding either guanidine utilization pathway[15,17,18]. Furthermore, the betaproteobacterial AOB exclusively possess genes for the ATP-requiring guanidine carboxylase pathway. By contrast, only two comammox genomes encode this pathway, whereas the majority of comammox genomes instead are predicted to be equipped with the more energy-efficient pathway using a Ni-containing guanidinase (Extended Data Fig. 1). Most comammox genomes and metagenome-assembled genomes (MAGs; 75 out of 83) also encode a urease for conversion of urea formed by guanidinase to ammonia. Taken together, ammonia oxidizers equipped for guanidine metabolism do not rely on an ATP-consuming guanidine transporter and comammox microorganisms additionally use an ATP-independent guanidinase. Thus, these ammonia oxidizers and, in particular, comammox, if indeed capable of using guanidine, would be much more energy efficient than the previously characterized *Pseudomonas* and *Synechocystis* strains (that use guanidine as a nitrogen source) in converting guanidine to three molecules of ammonia.

With the distribution patterns of guanidine utilization genes in mind, we tested the guanidine metabolization ability using equally dense cultures of five AOB strains that possess the complete gene set of the pathway (one lacking only the urease), and the only described comammox pure culture *N. inopinata*. Notably, after incubation for 2 weeks with 50 µM guanidine as the sole substrate, only *N. inopinata* was able to almost fully degrade guanidine as a sole substrate and produce nitrite and nitrate. Three of the AOB strains were able to convert a fraction of the added guanidine to nitrite but only in the concomitant presence of ammonium (Extended Data Fig. 2).

Consistent with a guanidine metabolism, a guanidine-I riboswitch (*ykkC-yxkD*) is found immediately upstream of the APC superfamily permease and guanidinase genes in *N. inopinata* and many other comammox microorganisms (Fig. 1c, Extended Data Fig. 1 and Supplementary Tables 2 and 3). In comammox organisms, ABC transporters and, unexpectedly, also Ni chaperones were absent from the genomic regions surrounding the guanidine-I riboswitch. This riboswitch class acts as a transcriptional suppressor, permitting transcription of downstream genes only in the presence of guanidine[4]. Phylogenetic analyses revealed that the comammox guanidinases belong to the ureohydrolase enzyme superfamily (Fig. 2a). Substrates converted by this enzyme family typically contain a guanidine moiety, which is hydrolysed to release urea. In the guanidinase from *Synechocystis* sp. PCC 6803, specificity to guanidine was dependent on two amino acids, threonine at position 97 and tryptophan at position 305 (ref. 18). The guanidinase from *N. inopinata* belongs to the same clade of this superfamily and also possesses threonine and tryptophan at the key sites for guanidine specificity (Fig. 2a,b). Comparison of the comammox guanidinase phylogenetic tree with the comammox ammonia monooxygenase tree revealed a clear separation between the comammox A and B groups, with several subclades of comammox group A also showing similar clustering and approximate branching order in the trees (Fig. 2c). These phylogenetic consistencies, and the very widespread distribution of putative guanidinases among comammox microorganisms, suggest that guanidine use is probably an ancient and persistent trait of comammox organisms. Together with the observation that most comammox strains contain the enzymatic repertoire for guanidine degradation, this strongly indicates that the ability for guanidine degradation confers a selective advantage to comammox strains and has therefore been retained in their genomes during evolution.

## Growth of *N. inopinata* on guanidine

To further characterize guanidine use by *N. inopinata*, and to test whether this comammox organism can grow on guanidine as the sole source of energy, reductant and nitrogen, batch incubation experiments were performed. For quantification of the guanidine

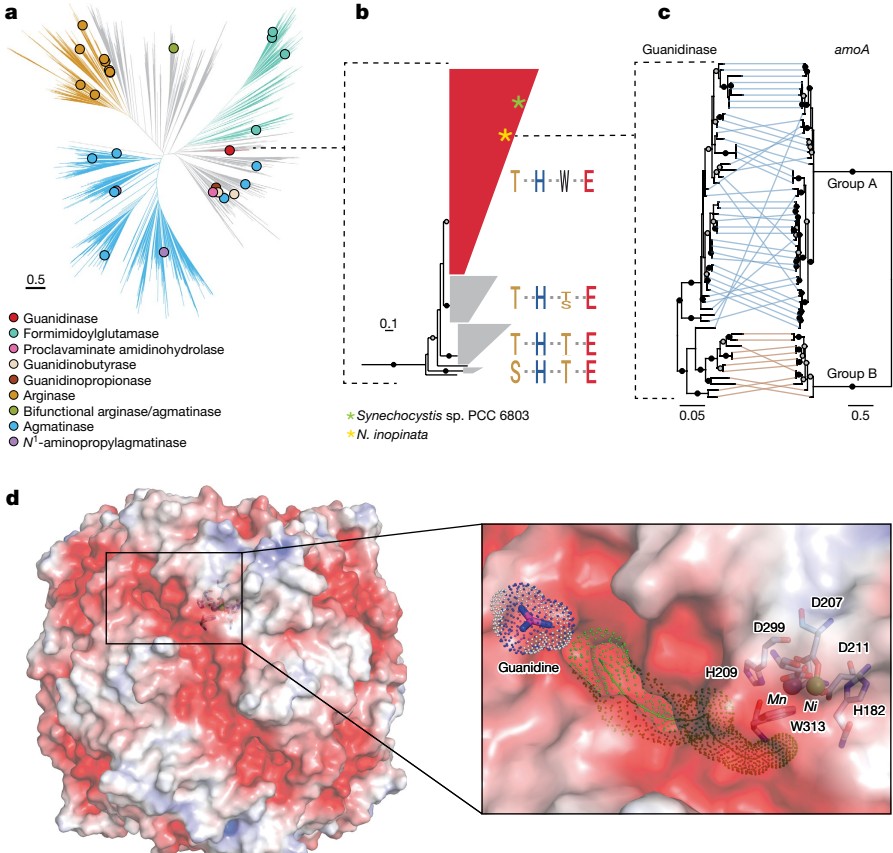

**Fig. 2 | Phylogeny and structure of comammox guanidinases. a**, Phylogeny of the ureohydrolase superfamily. The circles indicate functionally characterized members. Characterized $N^1$-aminopropylagmatinases were included in the indicated agmatinase clade. Branches in grey do not correspond to any known function or are not monophyletic for function. **b**, Simplified phylogeny of guanidinases, using biochemically characterized arginase family members as the outgroup (a full tree is shown in Extended Data Fig. 4a). Specific amino acid residues that might be important for guanidine catalysis according to previous studies[15,17,18] (*N. inopinata* guanidinase positions Thr105, His222, Trp313 and Glu344; Supplementary Table 4) are indicated. **c**, Comparison of the phylogenetic relationships of comammox guanidinases (amino acids) and ammonia monooxygenases (*amoA* nucleotides). Tree tips are connected for genes found in the same genome. **d**, Electrostatic surface representation of guanidinase (scaled from −5 kT/*e* (red) to +5 kT/*e* (blue)), with the suggested entry of a tunnel towards the active site highlighted (left). Right, magnification of the region indicated by a box. The tunnel as determined by CAVER[57] is shown as a green line, and its width is indicated by a dot representation. Active-site residues are shown as sticks, and nickel (Ni) and manganese (Mn) ions are shown as green and purple spheres, respectively. Guanidine is shown as sticks with corresponding van der Waals atomic radii indicated as dots.

concentrations, we adapted an analytical workflow based on derivatization of guanidine with benzoin[29]. In addition to the previously reported derivatization product, our analysis revealed an additional product that showed a higher signal intensity. We optimized the derivatization protocol to increase the fraction of this product and thereby improve the analytical sensitivity, confirmed the specificity of the derivatization (Supplementary Fig. 1 and Supplementary Table 5) and then used this protocol (with calibration solutions containing 1–100 μM guanidine) for all measurements.

In batch incubation experiments with around 50 μM of guanidine, *N. inopinata* converted guanidine stoichiometrically to nitrite and nitrate (Fig. 3 and Extended Data Fig. 3f). No abiotic guanidine degradation was detectable with dead *N. inopinata* biomass (Fig. 3a). During incubation with guanidine, *N. inopinata* cell numbers increased significantly in comparison to the controls without guanidine (Fig. 3c). Furthermore, *N. inopinata* assimilated nitrogen and carbon from isotopically labelled guanidine and bicarbonate, clearly demonstrating chemolithoautotrophic growth of *N. inopinata* on guanidine as a sole substrate (Fig. 3f,g). *N. inopinata* growth on guanidine was slower compared with growth on ammonium with maximum growth rates (division rates) of 0.076 and 0.632 d$^{-1}$, respectively (Fig. 3c, Extended Data Fig. 3c and Supplementary Table 6), and nitrite/nitrate production in the presence of both ammonium and guanidine was slightly (but significantly)

slower compared with ammonium only (Extended Data Fig. 3g), yet the cell yield (that is, the number of cells formed per mol of combined nitrite and nitrate produced) did not differ between treatments receiving guanidine, ammonium or both substrates (Extended Data Fig. 3i). Notably, when both ammonium and guanidine were supplied, both were used simultaneously, rather than in a diauxic growth pattern (Extended Data Fig. 3d). Guanidine utilization rates per cell did not differ between treatments receiving only guanidine, or both ammonium and guanidine (Extended Data Fig. 3h). These observations imply that guanidine is used by comammox microorganisms whenever it becomes available, consistent with the 'on/off' type regulation of guanidinase expression by the guanidine-dependent riboswitch.

Protein expression of *N. inopinata* grown on guanidine was compared to growth on ammonium. Pure cultures were repeatedly fed with either 50 μM guanidine or 150 μM ammonium after substrate depletion. After incubation for 3 weeks, three proteins showed significant differential expression between the guanidine and ammonium treatments: guanidinase (log$_{10}$[fold change (FC)] = 1.2, adjusted $P$ ($P_{adj}$) = 0.001), a putative fatty acid methyltransferase (FAMT, log$_{10}$[FC] = 1.1, $P_{adj}$ = 0.036) and a putative RNA-binding protein (RbpB, log$_{10}$[FC] = 2.5, $P_{adj}$ = 0.036) (Extended Data Fig. 4b). All three showed higher expression levels in the guanidine incubation than in the ammonium incubation, with guanidinase showing the highest significance

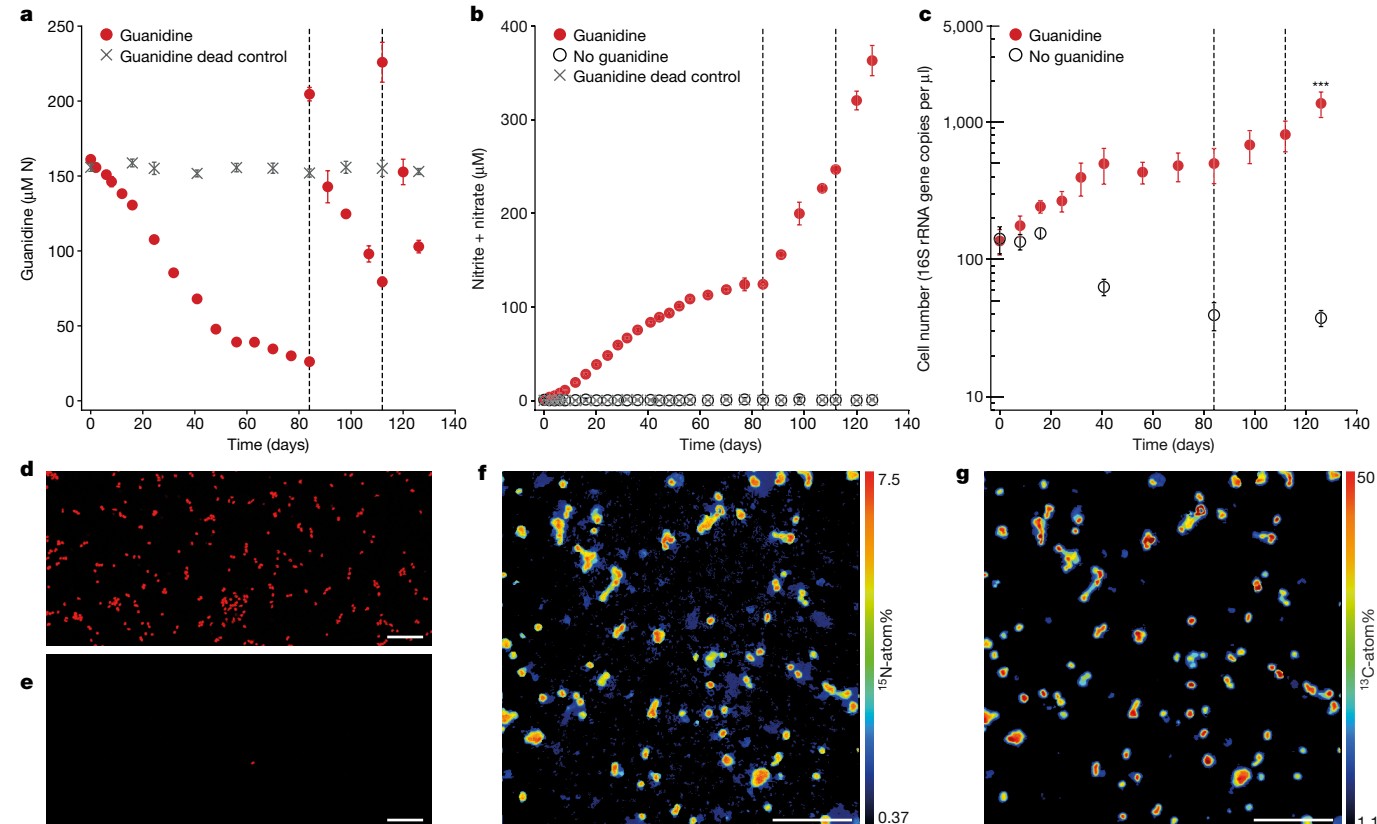

**Fig. 3 | Growth of *N. inopinata* on guanidine as the sole source of energy, reductant and nitrogen. a**, Biodegradation of guanidine over time. Around 50 µM (150 µM N) isotopically labelled guanidine was added to a washed culture of *N. inopinata* (after pre-incubation with guanidine and ammonium for 1 month) and incubated for 126 days. The control for abiotic guanidine decay was performed using autoclaved *N. inopinata* biomass. On days 84 and 112, additional spikes of around 50 µM of guanidine were added to living biomass incubations (dashed grey lines). Ammonium never increased above the level of detection (5 µM) and urea concentrations remained below 2.5 µM (Supplementary Table 6). **b**, $NO_2^-$ and $NO_3^-$ production (combined). At the end of the incubation, 78% of the total added guanidine nitrogen was oxidized to $NO_2^-$ and $NO_3^-$. Nitrogen balances are shown in Extended Data Fig. 3f. **c**, qPCR analysis of 16S rRNA gene copy numbers. Statistical analysis was performed using Welch two-sample *t*-tests; ***$P$ = 0.0049, $t$ = 10.348, d.f. = 4, comparing

*N. inopinata* cell numbers incubated with guanidine versus without guanidine at the 126 day timepoint. For **a**–**c**, data are mean ± s.d. across five biological replicates. **d**,**e**, Representative images of DAPI-stained *N. inopinata* cells (red, 10 ml culture filtered onto 0.2 µm GTTP filter) after 107 days of incubation with guanidine (**d**) and without guanidine (**e**). The same results were observed for all five biological replicates. **f**,**g**, Nitrogen (**f**) and corresponding carbon (**g**) isotopic enrichment of *N. inopinata* cells after 107 days of incubation with $^{15}$N-guanidine and $^{13}$C-bicarbonate as measured using nanoSIMS. The $^{13}$C-enrichment is lowered by dilution of the $^{13}$C-bicarbonate with $CO_2$ from the headspace air, $CO_2$ formation from pyruvate and breakdown of isotopically unlabelled guanidine in the medium. Data from one biological replicate are shown; a second replicate was measured with the same results. Scale bars, 10 µm (**d**–**g**).

($P_{adj}$ = 0.001). The APC permease, the putative guanidine transporter of *N. inopinata*, was detected in only one ammonium-treatment replicate, but was observed in 4 out of 5 guanidine-treatment replicates (Extended Data Fig. 4d). The APC permease and guanidinase are both under the control of the *ykkC-yxkD* riboswitch (Fig. 1c) and, therefore, both were expected to respond to guanidine in the same manner. However, the APC superfamily permease as an integral membrane protein is difficult to detect using proteomic approaches despite our use of protocols to maximize the recovery of such proteins. Neither FAMT nor RbpB are in the vicinity of a guanidine-dependent riboswitch in *N. inopinata*. Both were detected as differentially expressed with marginal significance ($P_{adj}$ = 0.036) and were not pursued further.

## Comammox guanidinase characterization

The guanidinase enzyme of *N. inopinata* was expressed in *E. coli*, purified to homogeneity and characterized as a homohexamer with a molecular mass of 240 kDa (Extended Data Fig. 5a), resembling the recently discovered guanidinase of *Synechocystis* sp. PCC6803 (GdmH)[18]. Crystal structure analyses of the heterologously expressed *N. inopinata*

guanidinase (resolution of 1.58 Å) revealed that it exhibits the same α-β-α fold of the arginase subfamily as its cyanobacterial homologue[18], but with each subunit (Fig. 2d and Extended Data Fig. 6) containing one nickel and one manganese ion with the overall structure being highly similar to the *Synechocystis* enzyme with an root mean squared deviation of 0.74 Å over an aligned length of 361 Cα atoms. However, the *N. inopinata* guanidinase possesses an N-terminal extension, which is lacking in the cyanobacterial enzyme (Extended Data Fig. 6b,c). This extension stabilizes the tertiary and quaternary structure of the guanidinase through extensive interactions, consistent with its notably high thermostability and activity in a wide pH range (Extended Data Fig. 5b,d and Supplementary Table 7). On the other hand, in comparison to the cyanobacterial enzyme, the *N. inopinata* guanidinase lacks a C-terminal extension (Supplementary Table 8 and Extended Data Fig. 6f).

In the active site of the heterologously expressed *N. inopinata* guanidinase, one nickel and one manganese ion are coordinated through conserved histidine and aspartic acid residues (Fig. 2d, Extended Data Fig. 6 and Supplementary Table 9). The positions and identity of metal sites were determined by the diffraction anomalous dispersion signal (Extended Data Fig. 6h,i) and corroborated by inductively coupled

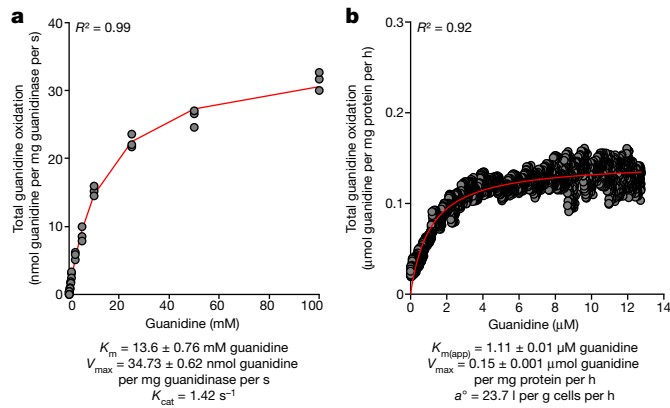

**a** $R^2 = 0.99$

$K_m = 13.6 \pm 0.76$ mM guanidine
$V_{max} = 34.73 \pm 0.62$ nmol guanidine
per mg guanidinase per s
$K_{cat} = 1.42$ s$^{-1}$

**b** $R^2 = 0.92$

$K_{m(app)} = 1.11 \pm 0.01$ µM guanidine
$V_{max} = 0.15 \pm 0.001$ µmol guanidine
per mg protein per h
$a^o = 23.7$ l per g cells per h

**Fig. 4 | Guanidine oxidation kinetics of purified guanidinase and of**
*N. inopinata* **cells. a,** Kinetic characterization of the heterologously expressed
*N. inopinata* guanidinase. A Michaelis–Menten model (red line) was fit to
triplicate guanidine consumption rates by the guanidinase (pH 7.5, 37 °C) and
used to determine the $K_m$ and $V_{max}$. **b,** The whole-cell guanidine oxidation rates
of *N. inopinata* were determined for cells that were pre-induced with guanidine
for around 12 h with microsensor measurements (two additional biological
replicates are shown in Extended Data Fig. 7a,b). A Michaelis–Menten model
(red line) was used to determine the apparent half-saturation constant ($K_{m(app)}$),
$V_{max}$ and substrate specific affinity ($a^o$) based on guanidine oxidation rates.

plasma mass spectrometry (ICP-MS) metal analysis (Extended Data
Table 1). Another notable difference between the active sites of the
*Synechocystis* and the *N. inopinata* enzymes is a tilt of *N. inopinata*
guanidinase Trp313 towards the active-site nickel ion. The tilted
Trp313 in *N. inopinata* guanidinase permits access to the active site
from the surface, in contrast to Trp305 of the cyanobacterial enzyme,
which blocks it (Extended Data Fig. 6g). The hexamer's ridges display
a negative electrostatic potential, and each cavity marks the start of
a 17 Å negatively charged tunnel towards the active site (Fig. 2d and
Extended Data Fig. 6e). This is suggestive of electrostatic attraction of
the positively charged substrate guanidinium (the predominant form
of guanidine under physiological conditions).

Kinetic analysis of the purified guanidinase of *N. inopinata* revealed
a substrate affinity ($K_m$) of 13.6 ± 0.76 mM for guanidine at pH 7.5 and a
temperature of 37 °C (Fig. 4a) when expressed in the presence of 1 mM
NiSO$_4$, highly similar to the guanidinase of the cyanobacterium *Synecho-
cystis* sp. PCC6803[17,18], although some kinetic properties were depend-
ent on the nickel concentration used during heterologous expression
(Extended Data Table 1). Moreover, the maximum reaction rate ($V_{max}$),
enzyme turnover ($K_{cat}$) and enzymatic catalytic efficiency ($K_{cat}/K_m$)
are all highly similar to the values obtained in a previous study[17] after
guanidinase overexpression in *Synechocystis*, but much lower than
the reported values of the latter guanidinase overexpressed along
with nickel-loading chaperones in *E. coli*[18] (Extended Data Table 1). By
contrast, the nickel-loading mechanism of *N. inopinata* is unclear as
no such chaperones were found in the genome and, therefore, could
not be co-expressed with the guanidinase in *E. coli*. Thus, it is tempting
to speculate that the lower $V_{max}$ and $K_{cat}$ values of the heterologously
expressed *N. inopinata* guanidinase reflect that the enzyme was not
completely loaded with metals. This is consistent with an inferred aver-
age occupancy of 0.33 for nickel ions (0.66 per subunit) in the *N. inopi-
nata* enzyme and with the ICP-MS data of the purified enzyme, which
revealed a stoichiometry of 0.42 nickel ions per subunit (Extended Data
Table 1). An alternative explanation for the low affinity for guanidine
could be that the enzyme of *N. inopinata* converts guanidine as a side
reaction and is actually specialized on another substrate (which would
require in vivo its recognition by the riboswitch, which has a very high
substrate specificity and has been shown to not recognize metabolites
with a guanidino group[30]). However, consistent with previous work on

the cyanobacterial guanidinase[18], we could not detect urea formation
by the comammox guanidinase from the putative alternative guanidine
compounds methylguanine, arginine, creatine, guanidino-butyrate and
guanidino-propionate. Only for agmatine, a very minor urea production
was detected that was much lower than the enzyme activity measured
with guanidine as substrate (Extended Data Fig. 5e). This lack of activity
for the alternative substrates aligns well with structural analysis, which
indicates that the small entry channel can accommodate guanidine
only when accounting for dynamic channel 'breathing'. The calculated
molecular volumes for the potential alternative substrates are 127% to
287% larger than that of guanidine[31].

We also determined the whole-cell kinetic properties of *N. inopi-
nata* using guanidine through substrate-dependent O$_2$ consumption
rates. This analysis reflects more accurately the in vivo ability of this
organism to convert guanidine, as it assesses the entire pathway from
guanidine uptake and degradation through ammonia and nitrite oxi-
dation to terminal oxidase activity in the actual cellular environment.
Guanidine oxidation rates increased with increasing guanidine con-
centrations in a Michaelis–Menten kinetic profile, which was used to
model the apparent substrate affinity (mean $K_{m(app)}$, 1.34 ± 0.25 µM
guanidine; $n = 3$) and maximum guanidine oxidation rate (mean $V_{max}$,
0.16 ± 0.02 µM guanidine per mg protein per h; $n = 3$) (Fig. 4b and
Extended Data Fig. 7a,b). The whole-cell affinity for guanidine was
much higher than the affinity of the purified heterologously expressed
enzyme. This might, for example, reflect a more efficient metal loading
of the enzyme in vivo, differences in post-translational modification
in the natural host or crowding effects in the cytoplasm. At guanidine
concentrations above around 400 µM, which probably do not occur in
environments in which comammox thrive, and which are much higher
than the guanidine concentrations used for the growth experiments,
additional guanidine resulted in a decreased rate of respiratory activity.
Guanidine inhibition at high concentrations was not specifically related
to inhibition of the guanidinase (see the enzyme kinetic experiments)
and also occurred when guanidine was added to ammonia-oxidizing
*N. inopinata* cells (Extended Data Fig. 7c,d). The ability of *N. inopinata*
to scavenge guanidine from a dilute pool, such as in an environmental
setting, can be assessed by its substrate-specific affinity ($a^o$), incorpo-
rating the whole-cell $K_{m(app)}$ and $V_{max}$ (ref. 32). *N. inopinata* has a mean $a^o$
for guanidine of 21.6 l per g cells per h. In comparison, when pregrown
under the same conditions, *N. inopinata* has a comparable mean $a^o$ of
74.5 l per g cells per h for urea (although a higher $a^o$ for this substrate
has been reported recently[33]) and an $a^o$ of 528 to 2,262 l per g cells per
h for total ammonium (Extended Data Fig. 7e–h). This highlights the
hierarchy of substrate acquisition and use by *N. inopinata*. Although
the whole-cell affinity of *N. inopinata* for guanidine is by two orders
of magnitude below its affinity for NH$_3$, it still resembles or exceeds
the whole-cell affinities for NH$_3$ of several terrestrial AOA from the
*Nitrososphaerales* and *Nitrosocaldales* phylogenetic lineages and of
many AOB[34].

## Comammox use guanidine in WWTPs

Nitrification is an essential step for the efficient removal of nitro-
gen compounds in WWTPs. Comammox thrives in some but not all
of these systems. As guanidine has been detected in human urine at
concentrations between 2 and 20 µM (refs. 35,36), and we detected
guanidine in the influent of a municipal WWTP (concentration,
0.5 µM; Extended Data Table 2), we hypothesized that comammox
microorganisms in WWTPs degrade guanidine under competitive
conditions and studied how they respond to guanidine pulses. For our
experiments, we selected two municipal WWTPs (Ribe and Haderslev,
Denmark), in which comammox *Nitrospira* were abundant nitrifiers
(Extended Data Fig. 8a–d). A previous metagenomic analysis had
reconstructed an abundant comammox MAG from each of the two
WWTPs[37]. Both comammox MAGs contained the guanidine-I riboswitch

upstream of the APC superfamily permease and guanidinase genes (Extended Data Fig. 1). After addition of 50 µM guanidine to starved biomass from these WWTPs, considerable guanidine degradation was observed, whereas a much slower degradation rate occurred in a control experiment with biomass from an Austrian WWTP in which comammox could not be detected by PCR[38] (Extended Data Fig. 8e–g). Activated sludge from the Ribe WWTP was additionally incubated with urea or ammonium and one replicate of each substrate amendment experiment was analysed by metatranscriptomics, using the different timepoints as replicates for comparison. At $T = 0$, baseline transcription of the guanidinase gene was 227 transcripts per million (TPM) in the Ribe sludge. In the Ribe WWTP, only the comammox guanidinase ($\log_2[FC] = 2.72$; $P_{adj} = 1.3 \times 10^{-8}$) and APC superfamily permease ($\log_2[FC] = 2.62$; $P_{adj} = 7.5 \times 10^{-11}$) were more highly transcribed in the guanidine-spiked incubation compared with in the ammonia-spiked incubation, demonstrating the functionality of the riboswitch. Three genes were more highly transcribed in the guanidine-spiked incubation compared with in the urea-spiked incubation: guanidinase ($\log_2[FC] = 2.63$; $P_{adj} = 8.2 \times 10^{-8}$), APC superfamily permease ($\log_2[FC] = 2.57$; $P_{adj} = 3.6 \times 10^{-10}$) and a CBS-domain-containing protein with unknown function (MBK9946601.1: $\log_2[FC] = 2.19$; $P_{adj} = 3.9 \times 10^{-3}$). High transcriptional levels of guanidinase and APC superfamily permease in the guanidine-spiked incubation were maintained throughout the experiment compared with the ammonia-spiked and urea-spiked incubations, and incubations with no experimental substrate addition (Fig. 5a), consistent with a high concentration of guanidine (>40 µM) over the time course (Fig. 5c and Extended Data Fig. 8f).

Similar metatranscriptomic experiments were performed with biomass from the Haderslev WWTP, but no ammonia- or urea-spiked incubations were included, and the guanidine-spiked incubation was compared to a no-substrate control. At $T = 0$, the baseline transcription of the guanidinase gene was 316 TPM in the Haderslev sludge. In total, 373 genes were found to be differentially transcribed, including the guanidinase ($\log_2[FC] = 1.91$; $P_{adj} = 1.3 \times 10^{-6}$). The APC superfamily permease transcripts increased at the early timepoints but were not found to have a significant difference between the guanidine-spiked incubation and the no-substrate control over the entire experiment ($\log_2[FC] = 1.58$; $P_{adj} = 0.09$). The guanidine concentration in the Haderslev replicate time series used for transcriptomics dropped substantially between 8 and 16 h of incubation; this decrease was accompanied by a decrease in the transcriptional activity of the guanidinase and the permease (Fig. 5b). These results corroborate the findings from the Ribe sludge experiment, in that the guanidine concentration appears to exert strong control over the transcription of the comammox guanidine permease and guanidinase in WWTP communities.

## Guanidine metabolism in soil

As we detected guanidine at low concentrations in urine and faecal samples from cows, pigs, chicken and sheep (Extended Data Table 2), guanidine metabolism in a comammox-containing agricultural soil, which had been fertilized with 525 kg total N per ha per year from solid cattle manure, was also investigated. As expected, the soil microbiome showed strong nitrification activity and added ammonium (30 µg N per g dry-weight soil) was rapidly nitrified (<2 days). Soil nitrification (ammonium consumption and nitrate formation) was completely inhibited by acetylene (Fig. 6). Notably, guanidine added to the soil was nitrified to nitrate over 27 days whether provided as the sole nitrogen source (30 µg N per g dry-weight soil) or in combination with ammonium (each ammonium and guanidine, 15 µg N per g dry-weight soil). However, when nitrification was inhibited, guanidine persisted in the soil at a high concentration for more than 27 days. Although these data do not prove guanidine degradation by comammox, they highlight the large contribution of ammonia-oxidizing microorganisms (or potentially

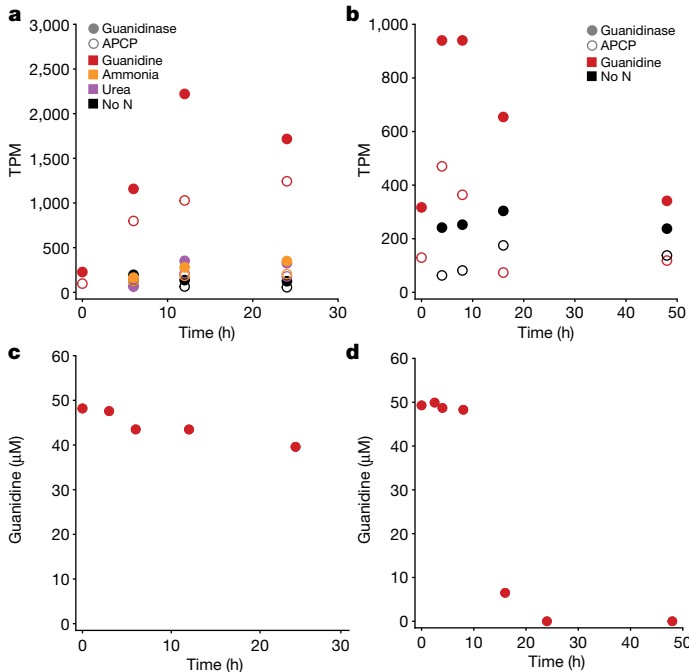

**Fig. 5 | Metatranscriptomic response of nitrifying activated sludges to guanidine amendment. a,b,** The transcriptional response of the guanidine APC superfamily permease (APCP) and guanidinase to substrate-spiked (guanidine, ammonium, urea) incubations in abundant comammox organisms from the Ribe (**a**) and Haderslev (**b**) WWTPs. Metatranscriptomic reads from the Ribe and Haderslev experiments were mapped to the respective Ribe and Haderslev comammox MAGs; transcriptional levels of the genes of interest are shown in TPM for all timepoints. **c,d,** Corresponding concentrations of guanidine in the replicate used for metatranscriptomics at time of sampling for Ribe (**c**) and Haderslev (**d**). Note that transcripts of the comammox guanidinases were also found at the start of the experiment, although the activated sludge samples had been starved during transport and storage, showing that transcription is either also occurring in the WWTP or that the sludge produced guanidine that was immediately consumed.

other acetylene-sensitive microorganisms) to the observed guanidine degradation (Fig. 6). In all treatments, quantitative PCR (qPCR)-based quantification of AOB, AOA, and comammox clade A and B using the functional marker gene ammonia monooxygenase subunit A did not detect growth of ammonia-oxidizing microorganisms over the course of 27 days ($P > 0.05$; Supplementary Table 10). This is not surprising, as conditions that support growth of ammonia oxidizers in soil vary widely among soils[39–41]. Growth of AOB is typically observed at higher nitrogen fertilization treatments than those used in our study[42,43], while the few studies reporting on comammox growth in soil observed it in soils unamended with nitrogen, or after substantially higher urea-N additions than those used in this study[41].

## Discussion

Comammox *Nitrospira* are to our knowledge the first organisms identified to grow with guanidine as their sole source of energy, reductant and nitrogen. It is well documented that functionally redundant microorganisms can coexist by partitioning low-concentration substrates even though they compete for one dominant substrate[44,45]. Thus, it is tempting to speculate that the ability to use the low-concentration substrate guanidine provides an additional niche for comammox organisms in guanidine-containing environments and is probably important for their co-existence with other ammonia-oxidizing microorganisms, with which they compete for the dominant substrate ammonia. Co-existence instead of competitive exclusion enables functional redundancy of

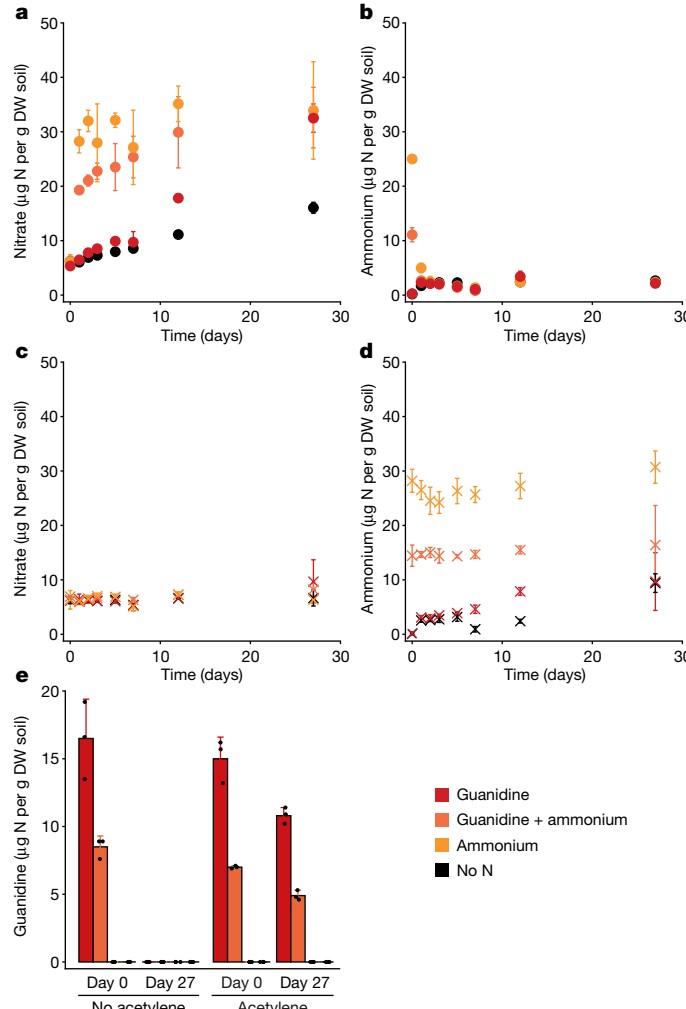

**Fig. 6 | Soil nitrification activity in the presence and absence of externally added ammonium and guanidine.** Nitrogen applications used in the soil microcosm incubations are comparable to those applied in field nitrogen fertilizer additions[58]. The red symbols represent addition of 30 μg guanidine-N per g dry weight (DW) soil; the orange symbols represent addition of 15 μg guanidine-N + 15 μg $NH_4^+$-N per g dry-weight soil; the yellow symbols represent addition of 30 μg $NH_4^+$-N per g dry-weight soil; and the black symbols represent no N addition. The circles represent treatments with active nitrification, and the crosses represent treatments with nitrification inhibited with 0.02% acetylene. For **a**–**d**, data are mean ± s.d. across three biological replicates. **a**, Net nitrate accumulation with active nitrification. In all treatments, urea and nitrite remained below the detection limit for the entire time course. **b**, Ammonium consumption with active nitrification. **c**, Nitrate and nitrite concentrations with nitrification inhibited with acetylene. **d**, Ammonium concentrations in the presence of acetylene. **e**, Guanidine concentrations (detection limit of 50 nM in the soil extract, corresponding to 14.1 ng N per g dry-weight soil) at day 0 and day 27 in the absence and presence of acetylene. Data are mean ± s.d. across three biological replicates. The overlaid dots show values from biological triplicates.

ammonia oxidizers, is frequently observed in many environments and probably preserves their overall ecosystem service.

The specific adaptation of comammox to using guanidine for nitrification also opens interesting perspectives for research and applications. Guanidine is the first substrate that has the potential to be selective for any lineage of ammonia oxidizers and has been reported to inhibit the AOB *Nitrosomonas europaea* at 1 mM concentration[46,47]. Thus, mineral cultivation media containing guanidine as the only substrate should be tested for purification of new comammox *Nitrospira*

strains from enrichments and environmental samples. Currently, only one comammox isolate (*N. inopinata*) and one high enrichment are available and have been partly physiologically characterized[21,48–50]. In comparison to ammonia oxidation by AOB, complete nitrification by *N. inopinata* yields significantly less nitrous oxide ($N_2O$) as a by-product under oxic and hypoxic conditions[49]. Future work is needed to examine whether the long-term application of guanidine-containing nitrogen fertilizers will influence the composition of the ammonia-oxidizer community in agricultural soils and the emissions of the greenhouse gas and ozone-depleting $N_2O$. Comammox *Nitrospira* could also have an important role for the removal of the widely used drug metformin in the environment. It is tempting to speculate that the presence of metformin—for example, in wastewater[51,52]—opens together with other guanidine sources a niche for comammox in exposed habitats. In this case, the continuous release of metformin into the environment might influence the composition of nitrifying populations.

As we are just beginning to understand the environmental concentrations of guanidine, the importance of this nitrogen-rich compound for the biogeochemical nitrogen cycle cannot yet be quantified. For this purpose, a more sensitive analytical method for measuring in situ guanidine concentrations with high specificity is urgently needed. As shown previously for cyanate, even low standing concentrations of a nitrification substrate can be linked to relatively high production and consumption rates[53]. Thus, future studies should aim to determine both the steady-state concentrations and the turnover rates of guanidine in pristine and human-affected ecosystems. Moreover, the diversity of natural and synthetic guanidinium compounds and their degradation pathways, such as of many neonicotinoid pesticides[54] and nitroguanidine, which is used in explosives[55,56], indicates that a multitude of hitherto mostly overlooked, biotic and abiotic guanidine-producing reactions may exist. Insights into the environmental dynamics of guanidine and its derivatives will close yet another gap in our picture of the nitrogen cycle and shed light onto the ecological roles of comammox *Nitrospira* and their enigmatic functional separation from the canonical nitrifiers.

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

## Methods

### Gene collection and functional classification

Supplementary Table 11 contains details for the collection and analyses parameters of each gene family and the following description is generalized. Predicted proteins from all publicly available genomes in GenBank as of 1 July 2022 (429,896 genomes) were screened with hmmsearch[59] using 'collection HMMs' for genes related to guanidine metabolism (Fig. 1). The resulting genes were used as query sequences against a combined HMM set of PFAMs, TIGRFAMs, NCBIFAMs and PANTHERFAMs for a set of acceptable 'cross-check HMMs'. Genes were further screened using specific e-value and coverage cut-offs. Cross-check HMM names were used to query UniProt and results were filtered for reviewed entries with evidence at the protein level to identify functionally characterized proteins and download them if not already present in the dataset. The portion of the protein sequence that was aligned to the 'collection HMM' was extracted and clustered using usearch[60] with specified -id and -query_cov values to identify centroids. HMM-based alignments of centroid sequences generated from the initial hmmsearch were used in FastTree[61] to generate phylogenetic trees for each gene family of interest. For the APC superfamily permease and the allophanate hydrolase, all proteins that passed the e-value and coverage cut-offs were inferred to possess the expected function. Guanidinases were also required to possess threonine at N. inopinata position 105 (PF00491 HMM position 15), histidine at N. inopinata position 222 (PF00491 HMM position 134) and tryptophan at N. inopinata position 313 (PF00491 HMM position 223)[18]. Guanidine carboxylases were required to possess a conserved aspartic acid at K. lactis position 1,584 (TIGR02712 HMM position 956) and were further differentiated from urea carboxylase by having an aspartic acid at K. lactis position 1,330 (TIGR02712 HMM position 701)[15]. Carboxyguanidine deiminases were defined using the common ancestor of the two subunits (CgdA and CgdB) in the general tree for PF09347. This common ancestral node then gave rise to CgdA and CgdB as two monophyletic clades defined as such.

### Riboswitch collection

Genomes from ammonia-oxidizing microorganisms were screened using infernal (v.1.1.3)[62] using established RFAM covariance models for guanidine riboswitches I (RF00442), II (RF01068) and III (RF01763) and a model for the recently described guanidine IV riboswitch, which was constructed with infernal using 'GGAM-1-curated.sto'[63]. Scaffold IDs, coordinates and orientation were recorded and cross-referenced against gff files to identify downstream genes. A gene was considered to be under the control of a riboswitch if it was in the same orientation as the riboswitch and the 5′ end of the gene was within 1,000 nucleotides of the riboswitch. The inferred operon was then extended downstream until genes could be found with the opposite orientation.

### Phylogenetic analyses

**Ureohydrolase.** HMM-based alignments of centroid sequences defined above were used in FastTree2[61] with the default parameters to generate a phylogenetic tree. The tree was midpoint rooted using the function midpoint() from the phangorn package, and functional clades were defined using the getMRCA() function within the ape package and visualized using the ggtree package in R.

**Guanidinase.** The most recent common ancestor of all guanidinases (as defined above) was identified in the ureohydrolase tree using the getMRCA() function and the descendant centroids were collected using the Descendants() function, both from the ape package[64]. All HMM alignments of ureohydrolases that were represented by centroids collected using the Descendants() function were additionally required to have covered PF00491 over 90% of its length reclustered using usearch (-id 0.9 -query_cov 0.9). The HMM-aligned portion of this sequence dataset was used to calculate phylogeny with IQ-TREE2[65], using the

best model (LG+I+I+R5), and bipartition support was evaluated using ultrafast bootstraps. Logos for each resulting clade of guanidinases and close relatives were generated using the ggseqlogo package in R.

For co-phylogeny analyses of comammox guanidinases and ammonia monooxygenases, comammox genomes were screened for the presence of amoA and guanidinase genes. As most available genomes were MAGs, it was required that exactly one copy of each gene was identified per genome. This resulted in 54 genomes for analysis. The AlignTranslation and AlignSeqs functions from the Decipher package were used to align amoA nucleotide and guanidinase amino acid sequences, respectively. IQ-TREE2 was used to identify the best models (amoA, TPM3u+F+I+I+R3; guanidinase protein, LG+I+G4), calculate trees and evaluate bipartition support with ultrafast bootstraps. Trees were visualized in R using a combination of the cophylo function from the phytools package and the ggtree package.

### Guanidine quantification

The following chemicals were purchased from Sigma-Aldrich: guanidine hydrochloride (≥99%, G3272), benzoin (≥99%, 8.01776), potassium hydroxide (≥85%, 1.05033), ethanol (≥99.8%, 02851), formic acid (≥98%, 5.43804), β-mercaptoethanol (≥99%, 8.05740), L-arginine (≥99.5%, 11009) and sodium sulfite (≥98%, 239321). Hydrogen chloride solution (32%, 20254.321) and acetonitrile (≥99.9%, 20060.320) were purchased from VWR. 2-Methoxyethanol (≥99.5%, 10582945) was purchased from Thermo Fisher Scientific. MilliQ water was obtained from a water purification system (0.071 μS cm$^{-1}$; Elga Veolia, PURELAB Chorus). The derivatization protocol for guanidine was adapted from a previous study[29]. In brief, 150 μl of an aqueous solution potentially containing guanidine was cooled to 0 °C in a 0.5 ml plastic tube (Eppendorf, Protein LoBind, 0030108434) and spiked with 75 μl of a benzoin solution in ethanol (4 mM), 75 μl of an aqueous solution containing both β-mercaptoethanol (0.1 M) and sodium sulfite (0.2 M), and 150 μl of an aqueous solution of potassium hydroxide (1.6 M). The resulting solution was mixed, heated in a bath of boiling water for 10 min and cooled in an ice bath for 2 min. Subsequently, 25 μl of an aqueous solution of hydrogen chloride (4.8 M) was added. The resulting solution was mixed and transferred to a 1.5 ml plastic tube (Eppendorf, 0030120086) and centrifuged at 10,000g for 2 min. Before analysis, the supernatant was diluted to obtain analyte concentrations in the optimal quantification range of the analytical instrument (that is, 0.05–5 μM). The predominant derivatization product (proposed structure in Supplementary Fig. 1c) was analysed using liquid chromatography (Agilent 1290 Infinity II) coupled to triple quadrupole mass spectrometry (Agilent, 6470) with a retention time of 3.73 min. We used the InfinityLab Poroshell 120 Bonus-RP (Agilent, 2.7 μm, 2.1 × 150 mm) column for separation, an injection volume of 2 μl, a flow rate of 0.4 ml min$^{-1}$, a column compartment temperature of 40 °C and the following eluents: aqueous (A): MilliQ water with 0.1% (v/v) formic acid; organic (B): acetonitrile with 0.1% (v/v) formic acid. The eluent gradient was as follows: 0–1.5 min, 5% B; 1.5–4 min, 5–61% B; 4–4.5 min, 61–95% B; 4.5–7 min, 95% B; 7–8 min, 95-5% B; 8–10 min, 5% B. The source parameters were set as follows: positive mode electrospray ionization; gas temperature, 250 °C; gas flow, 10 l min$^{-1}$; nebulizer, 45 psi; sheath gas temperature, 280 °C; sheath gas flow, 11 l min$^{-1}$; capillary voltage, 3.5 kV; nozzle voltage, 0.5 kV. The following product ions of the derivatization product (m/z of parent ion: 252.2) were monitored: m/z: 182.1 (quantifier) and m/z: 104.1 (qualifier). The resulting chromatographs were integrated using MassHunter (Agilent, v.10.1). For absolute quantification, we used a series of guanidine solutions with a concentration range after dilution between 0.05 and 5 μM. For pure culture medium, activated sludge and soil extracts, calibration solutions were prepared in the respective matrix. Animal urine and faeces were quantified with calibration solutions in water. Accurate quantification for animal samples was confirmed by spiking 20 μM guanidine to an animal faeces sample with a recovery of >83%. Animal manure samples were freeze-dried, and

subsamples were dispersed in 2 M KCl solution (1 ml per 100 mg sample) and bead-beated for 15 min in a Lysing matrix A tube (MPBiomedicals), then centrifuged at 20,000*g* for 15 min. Wastewater treatment plant influent was quantified by standard addition. Calibration solutions were derivatized and analysed in the same way as and in parallel to the respective samples. For Orbitrap (high-resolution) MS analyses, we used liquid chromatography coupled to the Thermo QExactive mass spectrometer with the following parameters: positive electrospray ionization; capillary temperature, 275 °C; sheath gas, 15; aux gas, 10; sweep gas, 1; S-lens RF, 50.0; resolution, 140,000 (MS full-scan), 17,500 (MS/MS); NCE (stepped), 10,20,30. For growth experiment samples containing heavy-isotope-labelled guanidine, the total guanidine concentrations were inferred by assuming the measured isotopically unlabelled guanidine concentrations to correspond to 90% (we used 10% isotopically labelled guanidine).

## Physiology experiments with *N. inopinata* and ammonia-oxidizing bacteria

The cells were grown in medium containing 54.4 mg l$^{-1}$ KH$_2$PO$_4$, 74.4 mg l$^{-1}$ KCl, 49.3 mg l$^{-1}$ MgSO$_4$·7 H$_2$O, 584 mg l$^{-1}$ NaCl, 147 mg l$^{-1}$ CaCl$_2$, 34.4 µg l$^{-1}$ MnSO$_4$·1H$_2$O, 50 µg l$^{-1}$ H$_3$BO$_3$, 70 µg l$^{-1}$ ZnCl$_2$, 72.6 µg l$^{-1}$ Na$_2$MoO$_4$·2H$_2$O, 1 mg l$^{-1}$ FeSO$_4$·7 H$_2$O, 20.0 µg l$^{-1}$ CuCl$_2$·2 H$_2$O, 80 µg l$^{-1}$ CoCl$_2$·6 H$_2$O, 3 µg l$^{-1}$ Na$_2$SeO$_3$·5H$_2$O, 4 µg l$^{-1}$ Na$_2$WO$_4$·2H$_2$O, 24 µg l$^{-1}$ NiCl$_2$·6 H$_2$O and 0.5 mM pyruvate. The medium was buffered by addition of 4 mM HEPES, with the pH set to 8. For regular culture maintenance, cultures were kept in closed Schott bottles at 37 °C without shaking in the dark. When indicated, guanidine hydrochloride was added from a filter-sterilized 0.1 M stock solution to a final concentration of 50 µM.

For comparing guanidine utilization by pure cultures of *N. inopinata* and AOB, all strains were induced in 1 l batch cultures for 6 weeks with 0.5 mM ammonium and 1 µM guanidine fed weekly. Subsequently, the same amount of biomass per culture as determined using the Pierce BCA protein quantification kit (Thermo Fisher Scientific; calculated final concentration in the incubation, 10 µg ml$^{-1}$) was collected, washed and resuspended in fresh medium in equal volumes and transferred to 96-well, flat bottom culture plates (Greiner Bio-One). In these plates the following incubations were done with either 50 µM guanidine only; 50 µM guanidine plus 150 µM ammonium; or 150 µM ammonium only for 14 days at 28 °C in the dark and without agitation (optimal growth conditions for the ammonia oxidizing organisms used, while 9 °C colder than the optimum for *N. inopinata*).

For growth experiments, *N. inopinata* pure culture cells pregrown on 10 µM guanidine and 0.5 mM ammonium (with weekly refeedings) for 1 month were collected by centrifugation (4,500*g*, 30 min), washed with N-free medium three times and resuspended in fresh medium. Aliquots of 200 ml were distributed into 250 ml serum bottles. Aliquots used as dead controls were autoclaved (120 °C, 20 min) before substrate additions. The following N substrates were added (always 150 µM N) to five replicate bottles each: (A) $^{15}$N-guanidine (10% $^{15}$N-guanidine hydrobromide, 90% guanidine hydrochloride); (B) guanidine (as guanidine hydrochloride); (C) $^{15}$N-guanidine (10% $^{15}$N-guanidine hydrobromide, 90% guanidine hydrochloride) and ammonium (each 150 µM N); (D) ammonium only; (E) no N addition (starved control); (F) dead (autoclaved) control with $^{15}$N-guanidine (10% $^{15}$N-guanidine hydrobromide, 90% guanidine hydrochloride). Moreover, all bottles received $^{13}$C-bicarbonate additions ($^{13}$C-NaHCO$_3$; 1 mM final concentration, 99% $^{13}$C) to detect chemolithoautotrophic growth and 0.5 mM sodium pyruvate as a reactive-oxygen-species scavenger[66]. Serum bottles were closed with sterile, HCl-cleaned blue butyl rubber stoppers (Chemglass) and incubated at 37 °C in the dark without agitation. Samples of 2 ml for the determination of cell numbers (using qPCR) and of N-compound concentrations were taken with sterile syringes and needles and replaced with air every 1 to 14 days (frequent sampling in the beginning of the experiment,

more spaced-out sampling after incubations containing ammonium were ended) over a time course of 126 days (12 days for treatments containing ammonium). Substrates were replenished after depletion. After 107 days of incubation, 10 ml samples were removed from treatments A, B, E and F, fixed with 3% formaldehyde (final concentration) for 30 min at room temperature, filtered onto polycarbonate filters (0.2 µm pore size, GTTP, 40 nm gold sputtered), washed with sterile 1× PBS, dried and stored frozen until further use. Cells were visualized after staining with 4′,6-diamidino-2-phenylindole (DAPI, 10 µg ml$^{-1}$, 5 min at room temperature) using a confocal laser-scanning microscope (inverted Leica TCS SP8X CLSM equipped with a 405 nm UV diode). At the end of the growth experiments, the absence of heterotrophic contaminants was confirmed by inoculation into heterotrophic growth medium (LB and TSY).

Ammonium, urea, nitrite and nitrate concentrations were measured by colorimetric protocols published previously[67]. In brief, combined ammonia and ammonium concentrations were determined using the indophenol blue method. Nitrite concentrations were measured spectrophotometrically using the Griess method after reacting with sulfanilamide and *N*-1-naphthyl-ethylenediamine dihydrochloride. Nitrate was measured by the same method after reduction to nitrite with vanadium chloride. Urea concentrations were measured using the thiosemicarbazide-diacetylmonoxime method[68], according to a previous study[18].

For quantification of *N. inopinata* cell numbers, qPCR was performed using the primers 515F/806R, targeting the V4 region of the 16S rRNA gene as described previously[69,70]. Standards were generated from purified PCR products generated from *N. inopinata* genomic DNA as template. The standards were quantified according to the Qubit dsDNA HS Assay Kit instructions. Standards containing 10$^9$ gene copies per µl were aliquoted and stored frozen at −20 °C until further use. Each standard aliquot was used and defrosted only once to freshly prepare tenfold serial dilutions (10$^8$–10$^2$ gene copies per µl). The qPCR assays were performed as follows: the frozen culture aliquots were four times freeze-thawed for cell disruption. A total of 0.25 µM of each primer was used in a mixture of 10 µl SYBR Green Supermix (Bio-Rad), 2 µl cell lysate or standard, and water in a final volume of 22 µl per reaction. The qPCR cycler (C1000-CFX96, Bio-Rad) settings were as follows: 95 °C for 15 min; 40 cycles of 95 °C for 30 s, 50 °C for 1 min and 72 °C for 45 s (plate read); and finishing with 72 °C for 2 min and a melting curve performance from 40 °C to 95 °C with an increase of 0.5 °C every 5 s. The efficiencies of the standard curves had an average of 86% and an $R^2$ of 0.999. Growth rates (division rates) were calculated as follows:

$$v(\text{d}^{-1}) = \log_2(N_{i+1}/N_i)/t \qquad (1)$$

where $v$ is the rate of division (d$^{-1}$), $N$ is the qPCR determined cell number at timepoint $i+1$ and $i$, and $t$ is the time interval between time point $i+1$ and $i$ in days.

For visualization of stable N and C isotope assimilation into *N. inopinata* cells from the supplied $^{15}$N-guanidine and $^{13}$C-bicarbonate, gold-sputtered filters containing cells from two replicate bottles (Treatment A, replicate A1 and A2) and a natural abundance (NA) control were glued onto antimony-doped silicon wafers (7.1 × 7.1 × 0.11 mm, Active Business Company) using superglue (Loctide). NanoSIMS measurements were performed on the NanoSIMS 50L instrument (Cameca) at the Large-Instrument Facility for Environmental and Isotope Mass Spectrometry at the University of Vienna. Before image acquisition, each analysis area was preconditioned by sequence of high and extreme low ion impact energy (EXLIE) Cs$^+$ depositions as follows: high energy (16 keV) at 50 pA beam current to a fluence of 5 × 10$^{14}$ ions cm$^{-2}$; EXLIE (50 eV) at 400 pA beam current to a fluence of 5 × 10$^{16}$ ions cm$^{-2}$; high energy to an additional fluence of 2.5 × 10$^{14}$ ions cm$^{-2}$. Data were acquired as multilayer image stacks by sequential scanning with a finely focused primary Cs$^+$ ion beam (approximately 80 nm probe size at a

2 pA beam current) over $45 \times 45\ \mu m^2$ areas with $512 \times 512$ pixel image resolution. The primary ion beam dwell time varied between 1 ms (A1, 74 planes; NA, 50 planes) and 5 ms (A2, 21 planes) per pixel per cycle. The detectors of the multicollection assembly were positioned to enable parallel detection of $^{12}C_2^-$, $^{12}C^{13}C^-$, $^{12}C^{14}N^-$, $^{12}C^{15}N^-$, $^{31}P^-$ and $^{32}S^-$ secondary ions. Image data analysis was performed using the OpenMIMS ImageJ plugin (OpenMIMS v.3.0.5, ImageJ v.1.54f), where the acquired datasets were aligned, deadtime and QSA corrected, processed (for example, accumulation, stable isotope ratio calculation) and exported for visualization of $^{13}C$ and $^{15}N$ enrichment (as $^{13}C$ and $^{15}N$ atom%).

## *N. inopinata* shotgun proteomics

For protein analysis, biomass was dissolved in lysis buffer (8 M urea, 2 M thiourea, 1 mM PMSF). Protein extraction was done by incubation at 95 °C, while shaking at 1,400 rpm for 5 min. Subsequently, the samples were treated for 3 min in an ultrasonication water bath (Elmasonic S30 H). To the cell suspension, 6.75 µl 25 mM 1,4 dithiothreitol (in 20 mM ammonium bicarbonate) was added and incubated for 1 h at 60 °C and 1,400 rpm shaking. Next, 150 µl 10 mM iodoacetamide (in 20 mM ammonium bicarbonate) was added and incubated for 30 min at 37 °C with 1,400 rpm shaking in the dark. Finally, 200 µl of 20 mM ammonium bicarbonate was added and the protein lysates were proteolytically cleaved overnight at 37 °C with trypsin (2.5 µl of 0.1 µg µl$^{-1}$ trypsin, Promega). The cleavage was stopped by adding 50 µl 10% formic acid. The peptide lysates were desalted using ZipTip µC18 tips (Merck Millipore). The peptide lysates were resuspended in 15 µl 0.1% formic acid and analysed using nanoliquid chromatography–MS (UltiMate 3000 RSLC-nano, Dionex, Thermo Fisher Scientific). MS analyses of eluted peptide lysates were performed on the Q Exactive HF mass spectrometer (Thermo Fisher Scientific) coupled with a TriVersa NanoMate (Advion). Peptide lysates were injected onto a trapping column (Acclaim PepMap 100 C18, 3 µm, nanoViper, 75 µm × 2 cm, Thermo Fisher Scientific) with 5 µl min$^{-1}$ by using 98% water/2% acetonitrile with 0.5% trifluoroacetic acid, and separated on an analytical column (Acclaim PepMap 100 C18, 3 µm, nanoViper, 75 µm × 25 cm, Thermo Fisher Scientific) at a flow rate of 300 nl min$^{-1}$. Mobile phase was 0.1% formic acid in water (A) and 80% acetonitrile/0.08% formic acid in water (B). Full MS spectra (350–1,550 *m/z*) were acquired in the Orbitrap at a resolution of 120,000 with automatic gain control target value of $3 \times 10^6$ ions.

Acquired LC–MS data were analysed with the Proteome Discoverer (v.2.5, Thermo Fischer Scientific) using SEQUEST HT and INFERYS Rescoring. Protein identification was performed using a database constructed from predicted proteins of *N. inopinata* downloaded from MicroScope[71] and common contaminating proteins. Searches were conducted with the following parameters: trypsin as enzyme specificity and two missed cleavages allowed. A peptide ion tolerance of 10 ppm and an MS/MS tolerance of 0.02 Da were used. As modifications, oxidation (methionine) and carbamidomethylation (cysteine) were selected. Peptides that scored $q > 1\%$ based on a decoy database and with a peptide rank of 1 were considered identified. Differential expression of proteins was evaluated using the DEqMS[72]. Normalized spectral abundance factors were also calculated for visualization purposes only.

## Heterologous expression and purification of *N. inopinata* guanidinase

The guanidinase gene of *N. inopinata* was amplified with self-designed, specific PCR primers which already contained the vector-specific linker overhangs for Gibson cloning (5′-CTGGAAGTTCTGTTCCA GGGGCCCATGGCGAAAAAGAGAACGTACC-3′ and 5′-CCCCAGAA CATCAGGTTAATGGCGTCAGCGTTTCTTTCGATTGCC-3′), using high-fidelity Phusion Plus PCR Master Mix (Thermo Fisher Scientific). The purified product was cloned into the pCoofy4 (pETM44; His6-MBP) expression vector by using the GeneArt Gibson Assembly EX Cloning Kit (Invitrogen) according to the manufacturer's protocol. The sequence of the insert was verified by Sanger sequencing.

Cultures were grown at 37 °C in auto-induction ZYP-5052 medium[73] supplemented with 0.5 µM, 20 µM, or 1 mM $NiSO_4$ for 5 h before cooling down at 4 °C for 15 min, followed by overnight expression at 20 °C. Cells were lysed in the presence of a protease inhibitor cocktail in 50 mM HEPES, 200 mM NaCl, 5% glycerol, pH 7.4 using a cell disruptor (Constant Systems) and centrifuged at 4 °C and 45,000g for 30 min. Guanidinase fused N-terminally to a His-MBP-tag was purified by affinity chromatography using MBPTrap HP columns (Cytiva). Subsequently, the His-MBP-tag was cleaved overnight with HRV-3C protease added at a mass ratio of protease to protein of 1:50. Guanidinase was further purified by MBPTrap HP columns (Cytiva), followed by size-exclusion chromatography on the HiLoad Superdex 200 26/600pg column (Cytiva) equilibrated with 20 mM HEPES, 200 mM NaCl, 5% glycerol, pH 7.4. For the 20 µM nickel in expression batch, this nickel concentration was maintained in all buffers during purification.

The sample was concentrated to around 10 mg ml$^{-1}$ by ultrafiltration by using Vivaspin centrifugal concentrators (Sartorius) and flash-frozen and stored at −80 °C. Protein identity and purity were analysed using SDS–PAGE.

## Size-exclusion chromatography combined with multiangle light scattering

SEC-MALS was performed using a Superdex 200 increase 10/300 GL (Cytiva) operated at 20 °C on the 1260 Infinity HPLC system (Agilent Technologies) coupled to a miniDawn Treos MALS detector (Wyatt Technology). The samples were injected (80 µl at 1 mg ml$^{-1}$) onto a column extensively equilibrated with 20 mM HEPES, 150 mM NaCl, pH 7.4. Measurement was performed using BSA as a control. The protein concentration was measured with a RI-101 refractive index detector (Shodex) and the average molecular mass was calculated using the program Astra (Wyatt Technology). The first-order fit Zimm formalism was used for analysis of light-scattering data as a data process procedure in Astra, and a generic protein dn/dc value of 0.185 ml g$^{-1}$ was used for guanidinase and BSA.

## Protein $T_m$ determination

The Prometheus NT.48 instrument (NanoTemper Technologies) was used to determine the melting temperatures ($T_m$). Before measurements, samples of the guanidinase expressed with 0.5 µM Ni$^{2+}$ were centrifuged for 10 min at 16,000g at 4 °C to remove any large aggregates. To identify the buffer/pH, at which the $T_m$ of the protein was the highest, the protein was diluted using a DSF-buffer/pH screen containing different buffers and pH values[74]. The capillaries were filled with 10 µl of sample and placed onto the sample holder. A temperature gradient of 1 °C min$^{-1}$ from 20 to 95 °C was applied and the intrinsic protein fluorescence at 330 and 350 nm was recorded. Data were processed using MoltenProt[75], where the melting temperatures from the curves were estimated using the two-state reversible unfolding model.

## MS for heterologous expression experiments

Protein identity and purity were verified by intact protein mass spectrometry. A total of 40 ng of the sample was injected into a column on the LC–MS system: Dionex nano HPLC, Waters XBridge C4, flow rate 250 µl min$^{-1}$ step gradient 12–40–80% ACN Synapt G2Si, resolution mode. Reconstruction of average mass was done with MaxEnt1 software[76].

## Metal determination by ICP-MS

To quantify Ni$^{2+}$ and Mn$^{2+}$ concentrations of the purified guanidinase, the samples were acid-digested and measured using ICP-MS. For acid digestion, HCl 30% (Supelco Suprapur, 100318, Merck), HNO$_3$ 65% (3-fold subboiled, provided in analytically pure quality; 1.00441.1000, Merck) were used. $H_2O_2$ (31%, ROTIPURAN Ultra, HN69.1) was purchased from Carl Roth. Deionized water was produced with 0.075 µS cm$^{-1}$ using an Elga Veolia, PURELAB Chorus 3 RO. 180 µl of the sample was pipetted

into 7 ml PFA vials (Savillex), corresponding to a total sample amount of between 2 and 2.5 mg. Subsequently, 0.5 ml HCl and 1.5 ml $HNO_3$ were added. After closing the vials gas tight, they were heated to 120 °C on a hot plate (Savillex). The temperature was kept constant for 12 h. After the samples had cooled down to room temperature, a total of 500 μl of $H_2O_2$ was added in 50 μl steps. Vials were closed again and heated at 120 °C for 12 h. Subsequently, vials were opened and the samples were brought to dryness at 120 °C. After cooling, the digestions were dissolved in 2 ml $HNO_3$ and brought again to dryness at 140 °C. Finally, the digestions were dissolved in 1 ml $HNO_3$ and 2 ml deionized water. Vials were closed and heated again at 120 °C for 12 h to ensure complete dissolution. The digestions were then quantitatively transferred to 15 ml centrifuge tubes (polypropylene, metal free) and filled up to 10 ml with deionized water. Twofold dilutions of the digestions were measured with an Agilent 7900 Single Quad ICP-MS instrument (Agilent Technologies) in no-gas mode. The operation parameters for the plasma were set to the following values: RF power: 1,550 W; RF matching, 1.80 V; sample depth, 10 mm; nebulizer gas flow, 0.8 l min$^{-1}$; makeup/dilution gas, 0.4 l min$^{-1}$. The parameters for data acquisition were as follows: acquisition mode, spectrum; sweeps/ replicate, 80; replicates, 3; integration time/mass, 0.1 sec. External calibration standards with an element concentration of 0.025 to 25 μg l$^{-1}$ were used for quantification. The limit of quantification values achieved for $Mn^{2+}$ and $Ni^{2+}$ were ≤0.17 μg l$^{-1}$ and ≤0.61 μg l$^{-1}$, respectively. The limit of detection (LOD) was ≤0.05 μg l$^{-1}$ for $Mn^{2+}$ and ≤0.18 μg l$^{-1}$ for $Ni^{2+}$. The measured concentrations of the diluted digestions ranged from 1.9 to 21.6 μg l$^{-1}$ for $Mn^{2+}$ and from <LOD to 22.8 μg l$^{-1}$ for $Ni^{2+}$. To exclude any contamination by the buffer used, this was also digested and measured. Here the concentrations ranged from <LOD to 0.07 μg l$^{-1}$ for $Mn^{2+}$ and from <LOD to 0.29 μg l$^{-1}$ for $Ni^{2+}$. Given that the concentrations of metals in all of the analysed samples were either significantly above the limit of quantification or below the LOD, the presence of these metals in the buffer solution was deemed not to have a relevant impact on the overall results.

## Protein crystallization

For initial screening, guanidinase expressed with 0.5 μM $Ni^{2+}$ was concentrated to 12.3 mg ml$^{-1}$ using the Amicon ultra centrifugal filter unit with 30 kDa MWCO and crystallized in MRC two-well crystallization plates with 50 μl of mother liquor set up using the TTPLabtech Mosquito pipetting robot system using the drop ratios 150 nl:200 nl and 200 nl:200 nl (protein:reservoir). Initial screens were performed using JCSG + HT, Index Screen, Morpheus Screen, PACT Premier screen and Crystal Screen at room temperature. Several hits were obtained from Crystal Screen and the condition F3, containing 0.5 M $(NH_4)_2SO_4$, 0.1 M $Na_3$ citrate pH 5.6 and 1 M $Li_2SO_4$ was used as a template for optimization screening by varying the $(NH_4)_2SO_4$ and $Li_2SO_4$ concentrations. The best crystals were obtained at 1 M $(NH_4)_2SO_4$ and 0.5 M to 0.7 M $Li_2SO_4$.

## X-ray data collection, model building and refinement

Crystals were cryo-protected using 20% glycerol, flash-frozen in liquid nitrogen and diffraction datasets collected at beamline ID30B at the European Synchrotron Radiation Facility (ESRF, France) under cryogenic conditions. The collected datasets were processed with XDS and converted to the mtz file format using XDSCONV[77]. The phase problem was solved with Phaser-MR[78], using its AlphaFold[79] prediction as a search model. The structure was further refined in iterative cycles of the manual model building using COOT[80] and maximum-likelihood refinement using the PHENIX software suite[81]. The final stages of refinement used the automated addition of hydrogens, and TLS refinement with one TLS group per chain. The models were validated with MolProbity[82] and PDBREDO[83]. Figures were created using PyMOL (The PyMOL Molecular Graphics System, v.2.0, Schrödinger) (Supplementary Table 12). Anomalous datasets were collected at ID30B at a wavelength of 1.8929 Å, close to the manganese anomalous scattering absorption edge, and at a wavelength of 1.4825 Å, close to the nickel anomalous scattering absorption edge. The anomalous datasets were processed as described above and the obtained mtz files were refined using the finalized model obtained from the native dataset. The anomalous maps obtained from refinement were averaged using phenix.ncs_average supplying the refined pdb structure file and the corresponding anomalous map in ccp4 file format. Averaged anomalous maps were visualized using PyMol.

## Substrate-dependent oxygen uptake measurements

Whole-cell substrate oxidation kinetics were determined from oxygen-uptake measurements as previously described[34,48,84]. Here oxygen-uptake measurements were performed using a microrespirometry (MR) system equipped with a four-channel MicroOptode meter (Opto-F4 UniAmp) and $O_2$ MicroOptodes. Real-time $O_2$ concentration monitoring was supported through SensorTrace Rate software (Unisense).

*N. inopinata* biomass was cultivated in batch cultures in the same growth medium as described above and ammonium (1 mM) or urea (0.5 mM) as sole substrates. Ammonium and guanidine were also used as co-substrates and here ammonium grown cultures (1 mM) were supplemented with guanidine (10–20 μM) around 12 h before MR experiments to induce the expression of the guanidine transporter and the guanidinase. In all cases, active *N. inopinata* biomass was collected (3,000*g*, 6 min, 20 °C) from substrate replete cultures, washed and resuspended in identical but substrate-free medium, and incubated in a recirculating water bath (>30 min, 37 °C). Samples were taken for chemical analysis to ensure the absence of detectable ammonium, nitrite, nitrate and urea before MR experiments. All chemical species were determined photometrically as described above.

MR experiments were conducted in a glass MR chamber (~2 ml) containing a glass-coated magnetic stir bar, on an MR2 stirring rack (350 rpm), in a recirculating water bath (37 °C). MR chambers were over-filled with concentrated biomass to ensure the absence of a gaseous headspace, closed with an MR injection lid and submerged in the water bath. An $O_2$ MicroOptode was inserted into each MR chamber and left to equilibrate (~1 h), before a stable background signal was determined (15–30 min). The background rate of oxygen depletion was subtracted from all subsequent rate determinations in each MR chamber. A Hamilton syringe (10 μl; Hamilton) was used for all substrate (ammonium, urea, guanidine) injections. Both single- and multiple-trace oxygen uptake measurements were performed.

For single-trace measurements, a single-substrate injection was performed, and the oxygen uptake was recorded until complete substrate depletion in the presence of excess $O_2$ (>30 μM $O_2$). The single-injection scheme was used to determine the molar ratio of urea and guanidine consumed per $O_2$. The whole-cell kinetics of *N. inopinata* with urea and guanidine as substrates, respectively, were performed with single-injection traces. Here a single injection of urea (~20 μM) or guanidine (~20 μM) into the MR chamber was performed. Moreover, the whole-cell kinetics of total ammonium oxidation in *N. inopinata* precultivated with urea (0.5 mM) or ammonium plus guanidine (1 mM and ~20 μM) was determined with single-trace measurements. Here a single injection of $NH_4Cl$ (~25 μM) was performed. In all cases, the experiments were halted after complete substrate depletion in the presence of excess $O_2$ (>30 μM $O_2$). Nitrate was the only detectable end product in all MR chambers used for whole-cell urea and guanidine kinetic calculations.

Multiple-trace measurements were used to determine the inhibitory effect of guanidine on the rate of maximum ammonium oxidation in *N. inopinata*. The maximum rate of ammonium oxidation was achieved with an initial injection of $NH_4Cl$ (250–500 μM). Subsequently, several injections of varying guanidine concentrations were performed, and discrete slopes of oxygen depletion were calculated after each injection (~2–5 min).

In all cases, for both single and multiple injections, MR chamber contents were immediately centrifuged (19,000$g$, 15 min, 20 °C) after the measurements and the cell pellets and supernatant were stored separately for protein and chemical analysis, respectively (−20 °C). For protein analysis, the total protein content was determined photometrically using the Pierce BCA Protein Assay Kit (Thermo Fisher Scientific). The chemical analyses (ammonium, nitrite, nitrate and urea) were performed as described above.

## In vitro enzyme activity assay of guanidinase

Guanidine degradation by the heterologously expressed and purified guanidinase (in the presence of different Ni$^{2+}$ concentrations; see above) was measured at 37 °C, pH 7.5, in a buffer containing 20 mM Tris-HCl and 50 mM NaCl by measuring urea production over 25 min of incubation. The measurements of the enzyme expressed in the presence of 1 mM Ni$^{2+}$ were done in the presence of 1 mM Ni$^{2+}$. Kinetics were calculated from measurements at 50, 100, 250, 500, 1,000, 2,500, 5,000, 10,000, 25,000, 50,000 and 100,000 µM guanidine starting concentrations. For screening alternative substrate use, the guanidinase expressed in the presence of 1 mM Ni$^{2+}$ was used. Then, 10 mM of methylguanidine, agmatine, arginine, creatine, guanidinobutyrate and guanidinopropionate each were incubated with the purified guanidinase enzyme or BSA at 37 °C for 30 min in three or six replicates. Guanidinase pH dependence was screened at 37 °C with incubations at pH 5.5, 6, 6.5, 7, 7.5, 8, 8.5, 9, 9.5, 10, 10.5 and 11 (set by addition of HCl or NaOH). Temperature dependence was screened at pH 7.5 with incubations at 14, 20, 28, 37, 46, 50, 55, 60, 65, 70, 80 and 90 °C. These incubations were done in triplicates.

## Calculation of cellular substrate oxidation kinetic properties

The cellular kinetic properties of total ammonium, urea and guanidine oxidation were calculated from single-trace substrate-dependent oxygen uptake measurements. The substrate oxidation rates were calculated from oxygen uptake measurements using a substrate-to-oxygen consumption ratio. For total ammonium oxidation, a substrate-to-oxygen ratio of 1:2 was used. Single-trace experiments were used to confirm the substrate-to-oxygen ratio for urea (3.9 ± 0.31, $n = 3$) and guanidine (6.17 ± 0.24, $n = 4$) oxidation. Thus, for total urea oxidation and total guanidine oxidation, substrate-to-oxygen ratios of 1:4 and 1:6 were used, respectively. All substrate oxidation rates were normalized to total cellular protein in each MR chamber. In the case of total ammonium oxidation, the $K_{m(app)}$ for unprotonated NH$_3$ was calculated based on the $K_{m(app)}$ for total ammonium, incubation temperature, pH and salinity[85].

The cellular kinetic properties of total ammonium, urea and guanidine oxidation were determined with a Michaelis–Menten model fit to the data using equation (2) where $V$ is the reaction rate (µM per mg protein per h), $V_{max}$ is the maximum reaction rate (µM per mg protein per h), $S$ is the total substrate concentration (µM), and $K_{m(app)}$ is the reaction half saturation concentration (µM). An unconstrained nonlinear least-squares regression analysis was used to estimate the $K_{m(app)}$ and $V_{max}$ values[86,87].

$$V = (V_{max} \times [S]) \times (K_{m(app)} + [S])^{-1} \qquad (2)$$

The reaction half-inhibition concentration for total ammonium oxidation ($K_i$, µM), inhibition by guanidine, was also determined. The $K_i$ was determined graphically with a Dixon plot analysis[88]. Inverse total ammonium oxidation rates were plotted against total guanidine concentration. Total ammonium oxidation rates resulting in a linear trend were used for these analyses. Linear best fit trendlines from each biological replicate were used to determine intersection focal points and estimate $K_i$ values. Furthermore, a linear regression of the percentage of the total ammonium oxidation rate at varying guanidine concentrations was used to determine the $K_i$.

The specific substrate affinity ($a^o$; litres per g wet cells per h) of ammonium, urea and guanidine oxidation was calculated using equation (3). The factor of 5.7 g wet cell weight per g of protein was used[32,48,89].

$$a^o = (V_{max} \times 5.7^{-1}) \times K_{m(app)}^{-1} \qquad (3)$$

## WWTP community structure analyses

The Ribe WWTP (GPS coordinates: 55.33, 8.74) has biological N and P removal (enhanced biological phosphorus removal) and treats municipal wastewater with 20% industrial contribution (organic loading) corresponding to a total of 25,000 person equivalents. It is designed with recirculation and has return sludge sidestream hydrolysis. It does not have primary settling. Suspended solids around the time of sampling were ~3.1 g l$^{-1}$. The Haderslev WWTP (GPS coordinates: 55.25, 9.51) has biological N and P removal and treats municipal wastewater with 5% industrial contribution corresponding to a total of 100,000 person equivalents. It is designed with alternating conditions and includes side stream hydrolysis. It does not have primary settling. Suspended solids around the time of sampling were ~3.2 g l$^{-1}$. The Klosterneuburg WWTP (GPS coordinates: 48.29, 16.34) treats municipal wastewater corresponding to a total of 50,000 person equivalents with a two-stage, biological hybrid process. Suspended solids around the time of sampling were ~4.4 g l$^{-1}$.

For characterizing the community structure, amplicon sequencing of the V1 to V3 regions of bacterial 16S rRNA genes was performed on samples from the Ribe and Haderslev WWTPs from the MiDAS BioBank collection. Applied PCR primers were 27F (5′-AGAGTTTGATCCTGG CTCAG-3′) and 534R (5′-ATTACCGCGGCTGCTGG-3′) with barcodes and Illumina adapters (IDT). PCR reactions (25 µl) were run in duplicate for each sample, using 1× PCRBIO Ultra Mix (PCR Biosystems), 400 nM of both the forward and reverse primer, and 10 ng template DNA. The PCR conditions were 95 °C for 2 min; followed by 20 cycles of 95 °C for 20 s, 56 °C for 30 s and 72 °C for 60 s; and a final elongation step at 72 °C for 5 min. The PCR products were purified using 0.8× CleanNGS beads and eluted in 25 µl nuclease-free water. The amplicon libraries were pooled separately in equimolar concentrations, diluted to 4 nM and paired-end sequenced (2 × 300 bp) on the Illumina MiSeq sequencer using v3 chemistry (Illumina). A 20% phage PhiX control library was added to mitigate low-diversity library effects. The forward and reverse sequence reads were merged using the software usearch[60] with the -fastq_mergepairs command, filtered to remove phiX sequences using usearch -filter_phix and quality filtered using usearch -fastq_filter with parameter -fastq_maxee set to 1.0. Dereplication was performed by usearch -fastx_uniques with the option -sizeout, and amplicon sequence variants (ASVs) were resolved using the usearch -unoise3 command. An ASV table was created by mapping the quality-filtered reads to the ASVs using the usearch -otutab command with the -zotus and -strand plus options. Taxonomy was assigned to ASVs using the usearch -sintax command with the parameters -strand both and -sintax_cutoff 0.8. The absence of comammox organisms in the sample from Klosterneuburg used for guanidine degradation measurements was confirmed by PCR using comammox clade A and clade B specific primer sets[38]. Ribe and Haderslev sample DNA in the same concentration were used as positive controls.

## Substrate incubation experiments with biomass from WWTPs

Activated sludge samples were collected from the aerated tanks of the Ribe and Haderslev WWTPs on 22 October 2021. Four litres of sludge from each WWTP were scooped into large sterile plastic bottles. The samples were transported to the laboratory on the same day, and were stored in the dark at ambient temperature, that is, ranging from 4 to 10 °C, until the incubations were started. The incubations with sludge from Ribe were started on the same day as collection, and the incubation with samples from Haderslev were started on the

day after collection. Before each incubation, the sludge was diluted approximately 1:4 as follows: the sludge was allowed to completely settle (1 h), then 1.5 l of the clear supernatant was gently collected to a new sterile flask without disturbing the flocs and, finally, 0.5 l of the remaining sludge was fully resuspended and added to the 1.5 l of supernatant. Well-mixed aliquots of 100 ml of the diluted sludge were then distributed to 200 ml sterile glass microcosms and covered with aluminium foil to enable gas exchange with the atmosphere. Substrates were added to the following final concentrations: guanidine, 50 μM; ammonia, 150 μM; and urea, 75 μM. These different concentrations were chosen to account for the number of amino groups among the molecules. No substrate controls were also included. The samples were incubated at 23 °C with shaking at 100 rpm. All substrate and control treatments were performed in triplicate. Microcosms were subsampled immediately before and after initial substrate additions at T0. Additional subsamples from the Ribe incubation series were taken at 3, 6, 12 and 24 h. Additional subsamples from the Haderslev incubation series were taken at 2.5, 4, 8, 16, 24 and 48 h. Subsamples for metatranscriptomics were immediately flash-frozen with liquid-$N_2$ and stored at −80 °C until processing. Parallel samples (1 ml) for chemical analyses were centrifuged at 12,000$g$ for 5 min, and the supernatant was taken and frozen immediately at −80 °C.

## RNA extraction and purification
Total nucleic acids were extracted from activated sludge samples (500 μl), which were thawed on ice and centrifuged (5 min, maximum speed, 4 °C), using the RNeasy PowerMicrobiome Kit (Qiagen) according to the manufacturer's instructions with the addition of phenol:chloroform:isoamyl alcohol (25:25:1) and β-mercaptoethanol (10 μl ml$^{-1}$ final concentration). Bead beating (40 s at 6 m s$^{-1}$, four times with 2 min interval on ice) on the Fastprep FP120 (MP Biomedicals) system was performed for cell lysis instead of vortexing to improve lysis of bacteria with rigid cell walls. The total nucleic acid extracts were subjected to DNase treatment to remove DNA contaminants using the TURBO DNA-free kit (Invitrogen), and further cleaned up and concentrated with RNAclean XP beads (Beckman Coulter) before rRNA depletion. The integrity and quality of the purified total RNA were assessed on a Tapestation 2200 (Agilent) with the Agilent RNA Screen-Tape (Agilent) system, and the concentration was measured using the Qubit RNA BR Assay Kit (Thermo Fisher Scientific). The average RNA integrity number was above 7.0 for all of the samples.

## rRNA depletion, library preparation and sequencing
Total RNA was rRNA-depleted using the NEBNext rRNA Depletion Kit for Bacteria (New England Biolabs) with 100–300 ng total RNA as input. The NEBNext Ultra II Directional RNA Library Prep Kit (New England Biolabs) was used to prepare cDNA sequencing libraries according to the manufacturer's instructions. The libraries were pooled in equimolar concentration and 2.0 nM was sequenced on an S4 flow cell on the NovaSeq 6000 platform (Illumina) using the v1.5 300 cycle kit (Illumina, 20012863).

## Identification of differentially transcribed genes
rRNA-depleted reads were adapter-screened, quality-filtered and mapped to published MAGs using bbmap v.38.92. Adapter removal and quality filtering was conducted using bbduk (ktrim=r k=21 mink=11 hdist=2 minlen=119 qtrim=r trimq=15). Metatranscriptomic reads from WWTP Ribe were mapped to genome accession GCA_016722055.1 and reads from WWTP Haderslev were mapped to genome accession GCA_016712165.1, which were the dominant comammox MAGs in the respective WWTP[37]. Both mappings were carried out using bbmap (minid=0.98 idfilter=0.98 ambiguous=toss pairedonly=t killbadpairs=t mappedonly=t bamscript=bs.sh) to produce bamfiles. Counts for each gene were calculated using bedtools coverage (-counts) using BAM files from bbmap and GFF files downloaded from GenBank for each

genome. Counts for each coding gene were examined for potential outliers, which identified MBK8278324.1 and MBK9947797.1 as potentially misannotated small RNAs and were removed from subsequent calculations. Differential transcription was evaluated by treating different timepoints as replicates and comparing treatments as factors using DESeq2[90]. TPM was calculated and used for visualization purposes only.

## Soil incubations
Soil was collected from a long-term fertilization experiment managed by the Austrian Agency for Health and Food Safety located at the Ritzlhof field experiment (48° 11′ 17.9′′ N 14° 15′ 16.5′′ E) in May 2023. The soil is classified as a Cambisol and has been fertilized since 1991 with solid cattle manure at an application rate of 525 kg N per ha per year[91]. Soil incubations were conducted in 125 ml Wheaton bottles capped with grey butyl stoppers. In brief, 30 g soil was added to each replicate ($n$ = 3) bottle, and amended with 820 μl water, ammonium, guanidine or ammonium + guanidine for a final concentration of 30 μg N per g dry-weight soil. Soils were incubated at 23 °C and sampled at 0, 1, 2, 3, 5, 7, 12 and 27 days. Acetylene (0.02%, v/v) was used to inhibit all lithotrophic ammonia oxidation. Acetylene was supplied by adding 0.3 ml of 10% acetylene gas to sealed bottles. Bottles were opened every 1–3 days, and acetylene was resupplied. For chemical analyses, around 2 g soil was extracted in water and 2 M KCl and extracts were frozen at −20 °C until analysis. Nitrate and nitrite were quantified in water extracts and ammonium, urea, and guanidine were quantified in KCl extracts as described above. Approximately 1 g soil was sampled for molecular analysis and was frozen at −80 °C until analysis. DNA extracts were performed using the ZymoBIOMICS DNA/RNA Miniprep Kit according to the manufacturer's instructions. AOB, AOA and comammox clade A and B $amoA$ qPCRs were carried out as previously described[38,92,93].

## Statistics
Statistical analysis on chemical, protein and qPCR data from physiological experiments and WWTP sample incubations were performed using two-tailed $t$-tests in SigmaPlot v.14.5 and R. No statistical methods were used to predetermine sample size, and blinding and randomization of samples were not used.

## Inclusion and ethics statement
All collaborators on this study fulfil the criteria for authorship required by Nature journals, they have been included as authors as their work was essential in designing and performing the study. The roles and responsibilities were agreed among collaborators ahead of the research. No living animals or animal-derived material were used in this study, except dropped animal manure and urine. Animals were not forced to excrete.

## Reporting summary
Further information on research design is available in the Nature Portfolio Reporting Summary linked to this article.

## Data availability
The crystal structures have been deposited in the Protein Data Bank under accession code 9FEK. The anomalous scattering datasets are available at the European Synchrotron Radiation Facility[94] (https://data.esrf.fr/doi/10.15151/ESRF-DC-1801440672). WWTP metatranscriptome reads have been deposited under BioProject ID PRJNA1118285. The MS proteomics data have been deposited at the ProteomeXchange Consortium via the PRIDE[95] partner repository under dataset identifier PXD038826. A high-resolution version of Extended Data Fig. 4a is available at Figshare[96] (https://doi.org/10.6084/m9.figshare.26139127). Source data are provided with this paper.

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

**Acknowledgements** We thank the staff of the ESRF and EMBL Grenoble, in particular A. McCarthy, for assistance and support in using beamline(s) ID30B under proposal number MX2591, the staff at the Mass Spectrometry Facility at Max Perutz Labs for proteomics analyses using the Vienna BioCenter Core Facilities (VBCF) instrument pool; V. Pevala for thermostability analysis of *N. inopinata* guanidinase; S. Hofbauer and P. Furtmüller as well as A. Sponga for their help with the kinetic modelling; M. Stockhausen for ICP-MS analysis; A. Schintlmeister for support with the NanoSIMS analyses; A. Spiegel and T. Sandén for providing access to the soil sample; and E. Selberherr and G. Terler for providing access to animal manure samples. N.J. thanks M. von Bergen and the funding of the UFZ for the ProMetheus platform for proteomics. This work was made possible in part by the OpenMIMS software, the development of which is funded by the NIH/NIBIB National Resource for Imaging Mass Spectrometry, NIH/NIBIB 5P41 EB001974-10. We are grateful for support by the Wittgenstein grant of the Austrian Science Fund FWF (Z383-B to M.W.), the FWF Young Investigators Research Grant program (ZK74; to C.J.S. and A.T.G.), the FWF grant P30570-B29 (to H.D.), the Comammox Research Platform of the University of Vienna (to H.D., M.W. and K.D.-C.), the Christian Doppler Laboratory for High-Content Structural Biology and Biotechnology (to K.D.-C.), the Vienna Science and Technology Fund Chemical Biology project LS17-008 (to K.D.-C.), and the Austrian-Slovak Interreg Project B301 StruBioMol (to K.D.-C.). M.K.D.D. and P.H.N. were supported by the Villum Foundation (grant 16558). This research was funded in part by the Austrian Research Fund (FWF) (https://doi.org/10.55776/COE7) (to M.W., K.K., H.D., K.D.-C., M.Z. and T.H.). For the purpose of open access, the authors have applied a CC BY public copyright licence to any Author Accepted Manuscript version arising from this publication.

**Author contributions** Conceptualization: M.W., H.D., K.D.-C., P.H.N., K.K. and A.T.G. Methodology: R.G. and M.Z. (guanidine measurements). Data curation: C.W.H., K.D.-C. and D.P. Formal analysis: D.P., C.W.H., C.J.S., M.K.D.D., M.P., A.T.G. and K.K. Investigation: M.P. (AOB activity, enzyme kinetics, enzyme specificity, comammox growth experiments), C.J.S. (whole-cell kinetics), N.J. (comammox proteome), D.P. (guanidinase structure and metal analyses), N.K. and J.K. (heterologous expression and protein purification), K.W. (WWTP experiments), C.S. (protein purification), J.P. (cultivation, qPCR), T.H. (guanidinase metal content), M.K.D.D. (WWTP amplicon analyses), A.L. (nanoSIMS analyses), K.K. (growth experiments, nanoSIMS), A.T.G. (soil incubations) and C.W.H. (bioinformatics analyses). Visualization: C.W.H., D.P., J.K., M.P. and K.K. Funding acquisition: M.W., H.D., C.J.S., K.D.-C., P.H.N., M.Z., K.K., T.H. and A.T.G. Project administration: M.W. Supervision: M.W., H.D., K.D.-C., D.P., M.Z. and K.K. Writing—original draft: M.W., C.W.H., M.P., C.J.S., H.D., D.P., K.D.-C. and K.K. All of the authors contributed to reviewing and editing the manuscript.

**Competing interests** The authors declare no competing interests.

**Additional information**
**Correspondence and requests for materials** should be addressed to Michael Wagner.

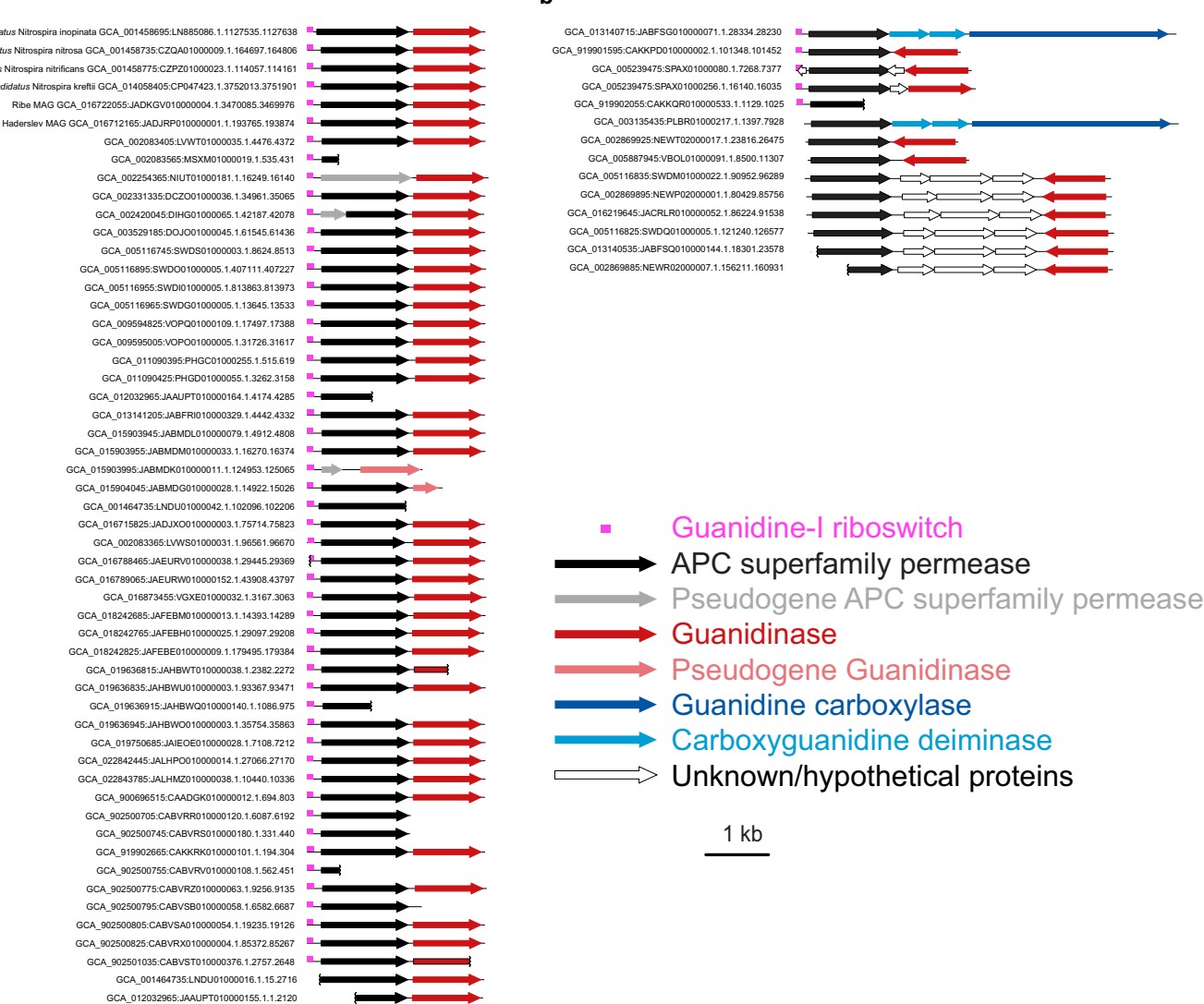

a

*Candidatus* Nitrospira inopinata GCA_001458695:LN885086.1.1127535.1127638
*Candidatus* Nitrospira nitrosa GCA_001458735:CZQA01000009.1.164697.164806
*Candidatus* Nitrospira nitrificans GCA_001458775:CZPZ01000023.1.114057.114161
*Candidatus* Nitrospira kreftii GCA_014058405:CP047423.1.3752013.3751901
Ribe MAG GCA_016722055:JADKGV010000004.1.3470085.3469976
Haderslev MAG GCA_016712165:JADJRP010000001.1.193765.193874
GCA_002083405:LVWT01000035.1.4476.4372
GCA_002083565:MSXM01000019.1.535.431
GCA_002254365:NIUT01000181.1.16249.16140
GCA_002331335:DCZO01000036.1.34961.35065
GCA_002420045:DIHG01000065.1.42187.42078
GCA_003529185:DOJO01000045.1.61545.61436
GCA_005116745:SWDS01000003.1.8624.8513
GCA_005116895:SWDO01000005.1.407111.407227
GCA_005116955:SWDI01000005.1.813863.813973
GCA_005116965:SWDG01000005.1.13645.13533
GCA_009594825:VOPQ01000109.1.17497.17388
GCA_009595005:VOPO01000005.1.31726.31617
GCA_011090395:PHGC01000255.1.515.619
GCA_011090425:PHGD01000055.1.3262.3158
GCA_012032965:JAAUPT010000164.1.4174.4285
GCA_013141205:JABFRI010000329.1.4442.4332
GCA_015903945:JABMDL010000079.1.4912.4808
GCA_015903955:JABMDM010000033.1.16270.16374
GCA_015903995:JABMDK010000011.1.124953.125065
GCA_015904045:JABMDG010000028.1.14922.15026
GCA_001464735:LNDU01000042.1.102096.102206
GCA_016715825:JADJXO01000003.1.75714.75823
GCA_002083365:LVWS01000031.1.96561.96670
GCA_016788465:JAEURV010000038.1.29445.29369
GCA_016789065:JAEURW010000152.1.43908.43797
GCA_016873455:VGXE01000032.1.3167.3063
GCA_018242685:JAFEBM010000013.1.14393.14289
GCA_018242765:JAFEBH010000025.1.29097.29208
GCA_018242825:JAFEBE010000009.1.179495.179384
GCA_019636815:JAHBWT010000038.1.2382.2272
GCA_019636835:JAHBWU010000003.1.93367.93471
GCA_019636915:JAHBWQ010000140.1.1086.975
GCA_019636945:JAHBWO010000003.1.35754.35863
GCA_019750685:JAIEOE010000028.1.7108.7212
GCA_022842445:JALHPO010000014.1.27066.27170
GCA_022843785:JALHMZ010000038.1.10440.10336
GCA_900696515:CAADGK010000012.1.694.803
GCA_902500705:CABVRR010000120.1.6087.6192
GCA_902500745:CABVRS010000180.1.331.440
GCA_919902665:CAKKRK010000101.1.194.304
GCA_902500755:CABVRV010000108.1.562.451
GCA_902500775:CABVRZ010000063.1.9256.9135
GCA_902500795:CABVSB010000058.1.6582.6687
GCA_902500805:CABVSA010000054.1.19235.19126
GCA_902500825:CABVRX010000004.1.85372.85267
GCA_902501035:CABVST010000376.1.2757.2648
GCA_001464735:LNDU01000016.1.15.2716
GCA_012032965:JAAUPT010000155.1.1.2120
GCA_015903965:JABMDJ010000152.1.25664.28164
GCA_019636915:JAHBWQ010000349.1.963.2659

b

GCA_013140715:JABFSG010000071.1.28334.28230
GCA_919901595:CAKKPD010000002.1.101348.101452
GCA_005239475:SPAX01000080.1.7268.7377
GCA_005239475:SPAX01000256.1.16140.16035
GCA_919902055:CAKKQR010000533.1.1129.1025
GCA_003135435:PLBR01000217.1.1397.7928
GCA_002869925:NEWT02000017.1.23816.26475
GCA_005887945:VBOL01000091.1.8500.11307
GCA_005116835:SWDM01000022.1.90952.96289
GCA_002869895:NEWP02000001.1.80429.85756
GCA_016219645:JACRLR010000052.1.86224.91538
GCA_005116825:SWDQ01000005.1.121240.126577
GCA_013140535:JABFSQ010000144.1.18301.23578
GCA_002869885:NEWR02000007.1.156211.160931

■ Guanidine-I riboswitch
➤ APC superfamily permease
➤ Pseudogene APC superfamily permease
➤ Guanidinase
➤ Pseudogene Guanidinase
➤ Guanidine carboxylase
➤ Carboxyguanidine deiminase
▷ Unknown/hypothetical proteins

1 kb

**Extended Data Fig. 1 | Co-occurrence of guanidine-I riboswitch, APC superfamily permease, and guanidinase genes on genome scaffolds from comammox *Nitrospira* group A (a) and B (b).** We identified a guanidine-I riboswitch in 67% (56/83) of the screened comammox genomes and found that the presence of the riboswitch was more widespread in comammox group A genomes (52/63; including the two metagenome-assembled comammox genomes from the wastewater treatment plants Ribe and Haderslev) than comammox group B genomes (4/20). The majority (75/83) of screened comammox genomes encoded urease. All comammox genomes with a guanidine-I riboswitch (56/56) encode an APC superfamily permease immediately downstream from the guanidine-I riboswitch, and the majority (46/56) encode guanidinase immediately downstream from the permease. It should be noted that the majority of comammox genomes are MAGs that are incomplete, preventing the triad of a guanidine-I riboswitch, an APC transporter, and a guanidinase from being located together in every genome. More specifically, for nine cases in which guanidinase is not found immediately downstream from the permease, the scaffold edge can be found within the permease or immediately downstream from the permease. In the final case, the genes immediately downstream from the riboswitch and permease are genes involved in the guanidine carboxylase pathway. Labels indicate the genome and scaffold accession IDs as well as genomic coordinates. For scaffold neighbourhoods with a riboswitch, the genomic coordinates indicate the coordinates of the riboswitch. For entries without a riboswitch, the accession IDs and end-point coordinates of the displayed genomic segment region are indicated. All genome regions visualized have been oriented so that the riboswitch and permease would be transcribed from left to right. Grey (permease) and light red (guanidinase) indicate genes that contain extensive frameshifts and therefore could be pseudogenes. We also screened all comammox genomes for the presence of the other three known guanidine riboswitches (mini-*ykkC*, *ykkc*-III and IV[63] and examined whether genes involved in guanidine degradation were located nearby (Supplementary Table 3). Two additional guanidine-responsive riboswitches, the mini-*ykkC* and *ykkC*-III riboswitches, were identified in five and two comammox genomes, respectively, and no genes known to be involved in guanidine degradation, transportation nor nickel incorporation were found in the vicinity of these. More generally, no obvious, dedicated nickel loading protein could be detected in any of the comammox genomes.

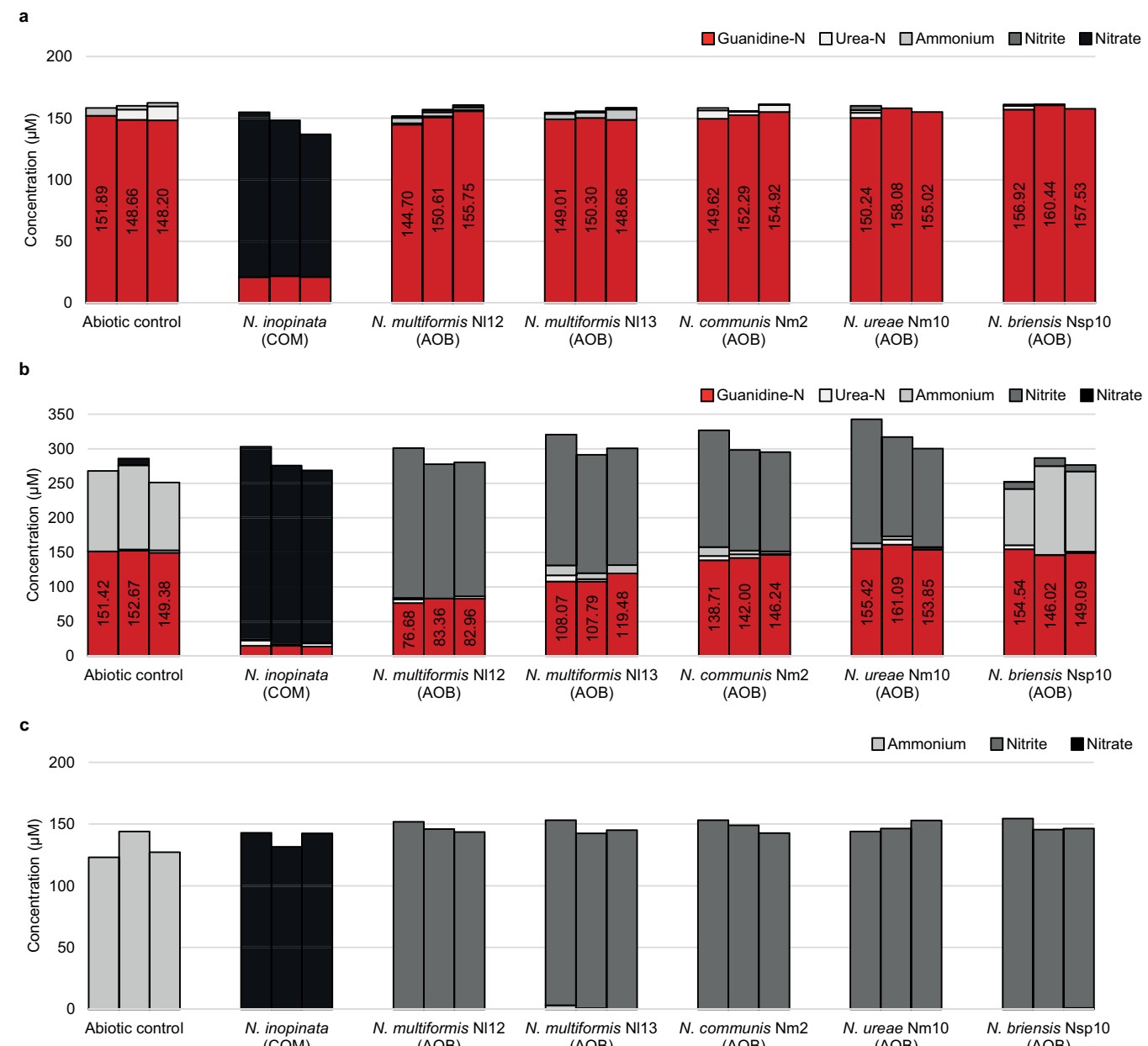

**Extended Data Fig. 2 | Potential of guanidine utilization by selected ammonia-oxidizing microbes.** Nitrogen balances for individual replicate cultures after 2 weeks of incubation with (**a**) 50 µM guanidine; (**b**) 50 µM guanidine plus 150 µM ammonium; and (**c**) 150 µM ammonium as the only substrate (activity control). Genomes of all five AOB strains harbour the complete genetic repertoire for guanidine transportation and utilization (guanidine/organocation transporter; guanidine carboxylase with signature amino acids[15,16]; carboxyguanidine deiminase; allophanate hydrolase; urease); only *Nitrosomonas communis* Nm2 lacks urease. *N. inopinata* possesses a guanidine/organocation transporter, a guanidinase, and a urease, and is among the tested organisms the only strain that was able to utilize guanidine in the absence of ammonium. In the presence of ammonium, three AOB strains were also able to significantly degrade guanidine (two-tailed student's *t*-test, d.f. = 4 for all): *N. multiformis* Nl12 ($t = 24.932$, $P = 0.0000154$), *N. multiformis* Nl13 ($t = 9.482$, $P = 0.000690$), and *N. communis* Nm2 ($t = 3.837$, $P = 0.0185$). All incubations were done in triplicates. Starting biomass was set equal among all strains, replicates, and treatments (with equal protein biomass density of the AOB strains as the comammox culture which showed rapid guanidine consumption).

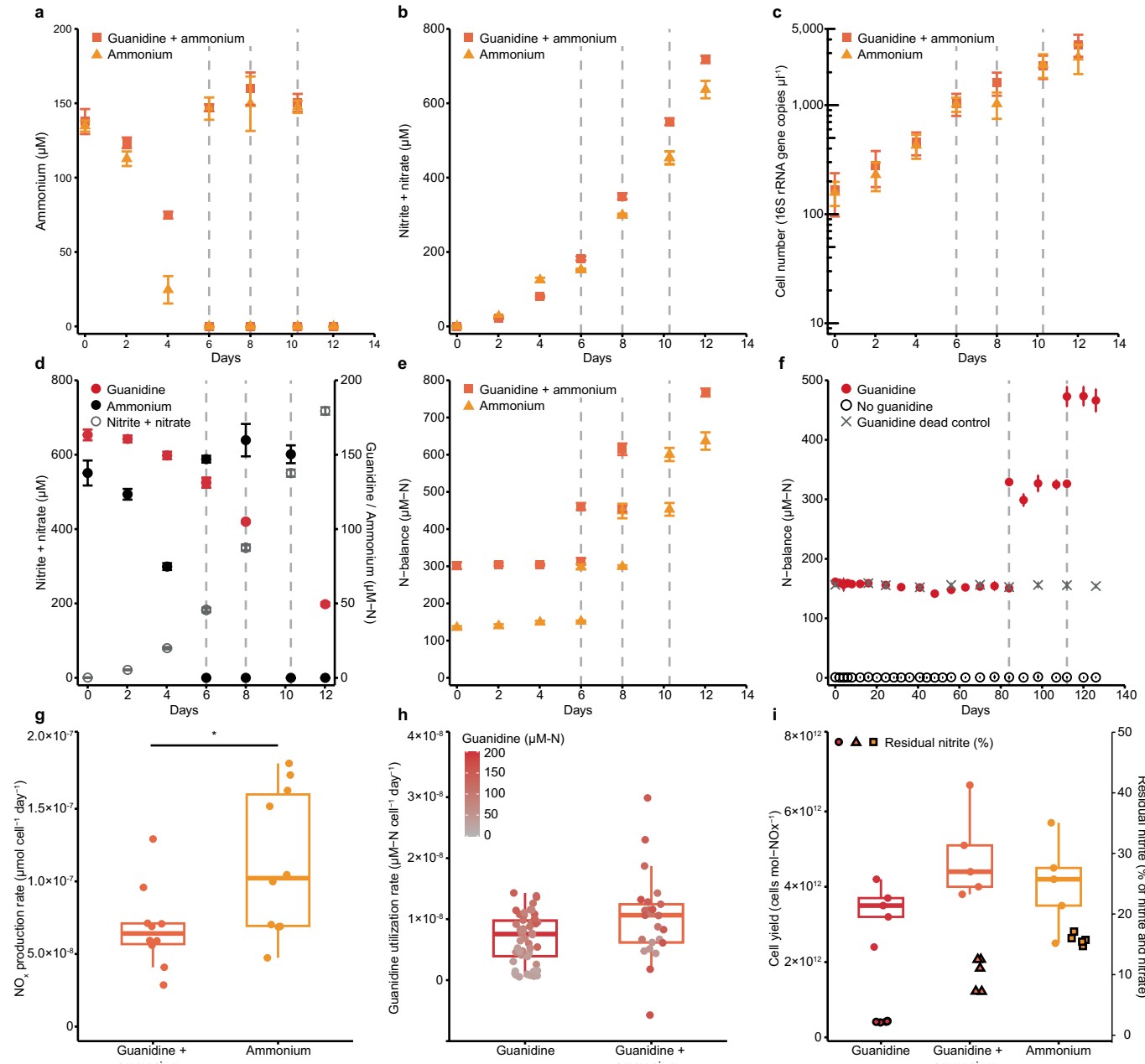

**Extended Data Fig. 3 | *N. inopinata* growth experiments.** (**a**) Ammonium consumption over time in treatments containing either ammonium and guanidine, or ammonium only. 150 µM N for both guanidine and ammonium were added to a washed culture of *N. inopinata* (after pre-incubation with guanidine and ammonium for several months) and incubated for 12 days. On days 6, 8 and 10, additional spikes of around 150 µM of ammonium were added to the incubations (dashed grey lines). (**b**) Nitrite and nitrate production (combined). (**c**) 16 S rRNA gene copy numbers measured by qPCR. (**d**) Concomitant utilization of guanidine and ammonium in treatments containing both substrates. (**e**, **f**) N balance for treatments receiving guanidine and ammonium, and ammonium only (**e**), and for guanidine, no guanidine (starved) and guanidine dead control (**f**). N-balances include guanidine (where added), ammonium, nitrite and nitrate. Urea concentrations were not included as they were always <2.5 µM. (**g**) Nitrite and nitrate (combined) production rate per cell and day in treatments receiving guanidine and ammonium, or ammonium only. Rates were calculated for time intervals before complete ammonium depletion (incl. day 4), by normalizing the difference in NO$_x$ concentration to average cell numbers between time points and the duration of the time interval. * *N. inopinata* NO$_x$ production when incubated with guanidine and ammonium versus ammonium only, Welch two sample *t*-test:

$t = -2.4714$, d.f. $= 14.174$, $P = 0.02673$. (**h**) Guanidine utilization rate per cell and day in treatments receiving only guanidine, or guanidine and ammonium. Rates were calculated across all time intervals as guanidine concentrations never dropped to 0, by normalizing the difference in guanidine concentration to average cell numbers between time points and the duration of the time interval. Colour gradient indicates the average guanidine concentration between time intervals. Higher utilization rates coincided with higher guanidine concentrations. No significant differences were found between treatments (Welch two sample *t*-test). (**i**) Cell yield per mol nitrite and nitrate produced across treatments, calculated between beginning and end of the incubation (12 days for treatments receiving guanidine and ammonium, or ammonium only; 126 days for the treatment receiving guanidine only). Note that all treatments contained residual nitrite (given as %), which may affect the overall energy conserved, and thus the cell yield per mol nitrite and nitrate produced. No significant differences were found between treatments (anova). All experiments were done in five biological replicates, datapoints show means, error bars standard deviations. Boxplots in (g-i) depict the 25–75% quantile range, with the centre line depicting the median (50% quantile); whiskers encompass data points within 1.5x the interquartile range.

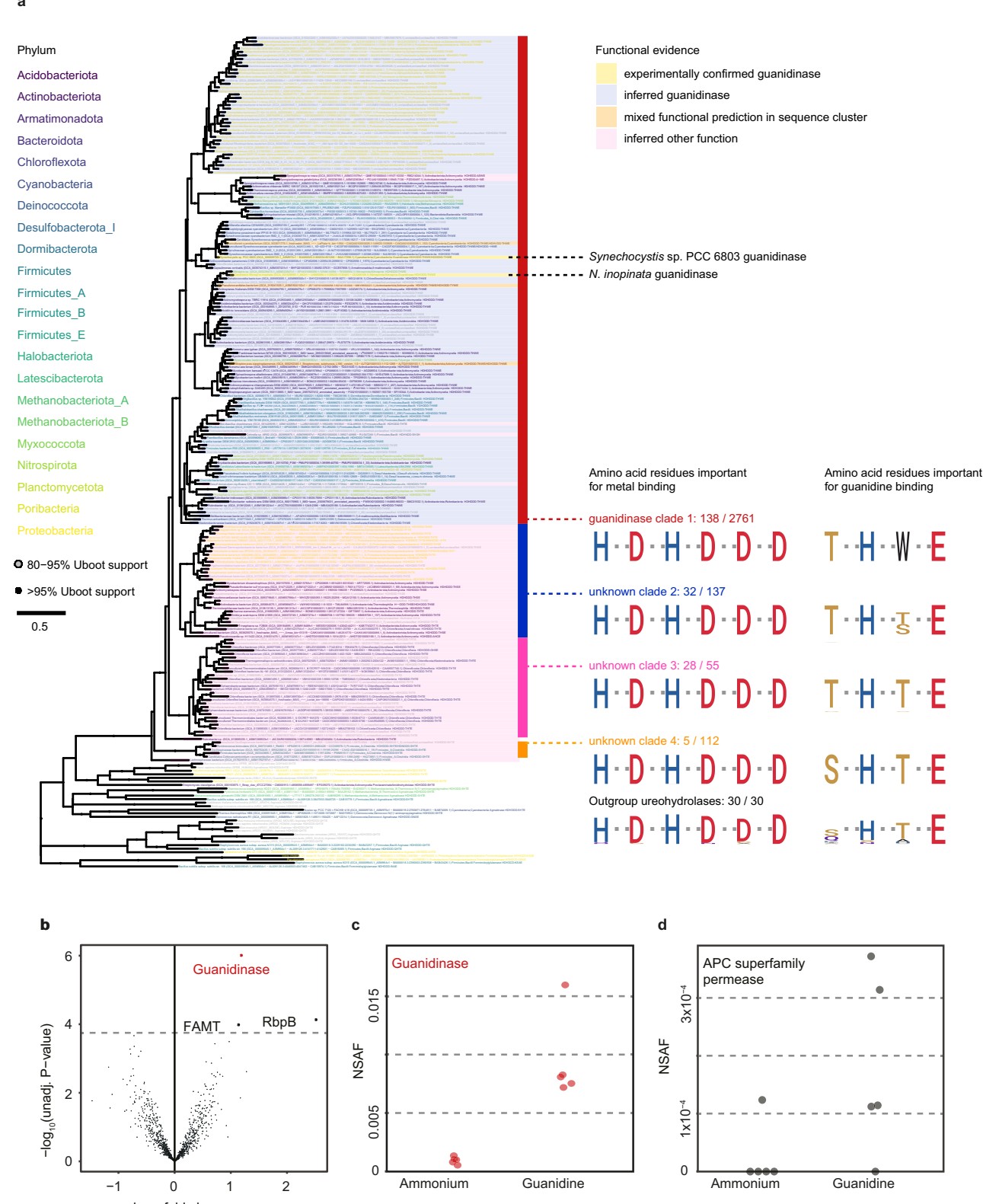

**Extended Data Fig. 4** | See next page for caption.

**Extended Data Fig. 4 | Phylogeny of guanidinases and proteomics of *N. inopinata* grown on guanidine and ammonium.** (**a**) Phylogeny of guanidinases according to an HMM alignment based on pfam model PF0049, using other biochemically characterized ureohydrolase family members as outgroup. The tree has been split into four general clades for simplification, which are indicated with coloured side bars. Specific residues that coordinated with nickel/manganese ions (*N. inopinata* guanidinase positions H182, D207, H209, D211, D299, D301) or have been proposed by Funck et al.[18] to be important for guanidine hydrolysis (*N. inopinata* guanidinase positions T105, H222, W313, E344) of guanidinase are indicated as a logo for each of the general clades as well as the outgroup (for details see Supplementary Tables 4 and 9). Each shown tip label is a centroid representative of a cluster of sequences (90% identity over 90% of HMM alignment). The logo title indicates an arbitrary clade name n/N where n is how many representatives in the shown tree belong to the clade and N is the total number of sequences represented by the centroids in the tree. Tip labels are coloured according to the phylum-level classification of the centroid representative. Guanidinase-specific residues were used to classify each centroid representative as "inferred guanidinase" (all members possess equivalent of T105 and W313), inferred not guanidinase (no members possess equivalent of both T105 and W313). "Mixed" refers to centroids that represent a cluster with mixed inferred function. The two experimentally characterized guanidinases from *Synechocystis sp*. PCC 6803 and *N. inopinata* are indicated. A high-resolution version of the phylogenetic tree can be found under https://doi.org/10.6084/m9.figshare.26139127. (**b**) Volcano plot of the differential expression of proteins in *N. inopinata* according to DEqMS[72]. Each point represents a single protein. Positive fold-change corresponds to higher expression levels during growth on guanidine, and negative fold-change corresponds to higher expression during growth on ammonium. Horizontal dashed lines show the significance threshold for *P*-values according to the Benjamini-Hochberg procedure with a false discovery rate of 0.05. Guanidinase (CUQ66148.1), RbpB (CUQ66942.1) and FAMT (CUQ67192.1) showed significant differential expression ($\log_{10}[FC] = 1.2$, $P_{adj} = 0.001$, $\log_{10}[FC] = 2.5$, $P_{adj} = 0.036$, and $\log_{10}[FC] = 1.1$, $P_{adj} = 0.036$, respectively). Protein expression of the (**c**) guanidinase and (**d**) APC superfamily permease in terms of normalized spectral abundance factor (NSAF). All experiments were performed in five replicates. Dots in (b) and (c) show data for individual replicates.

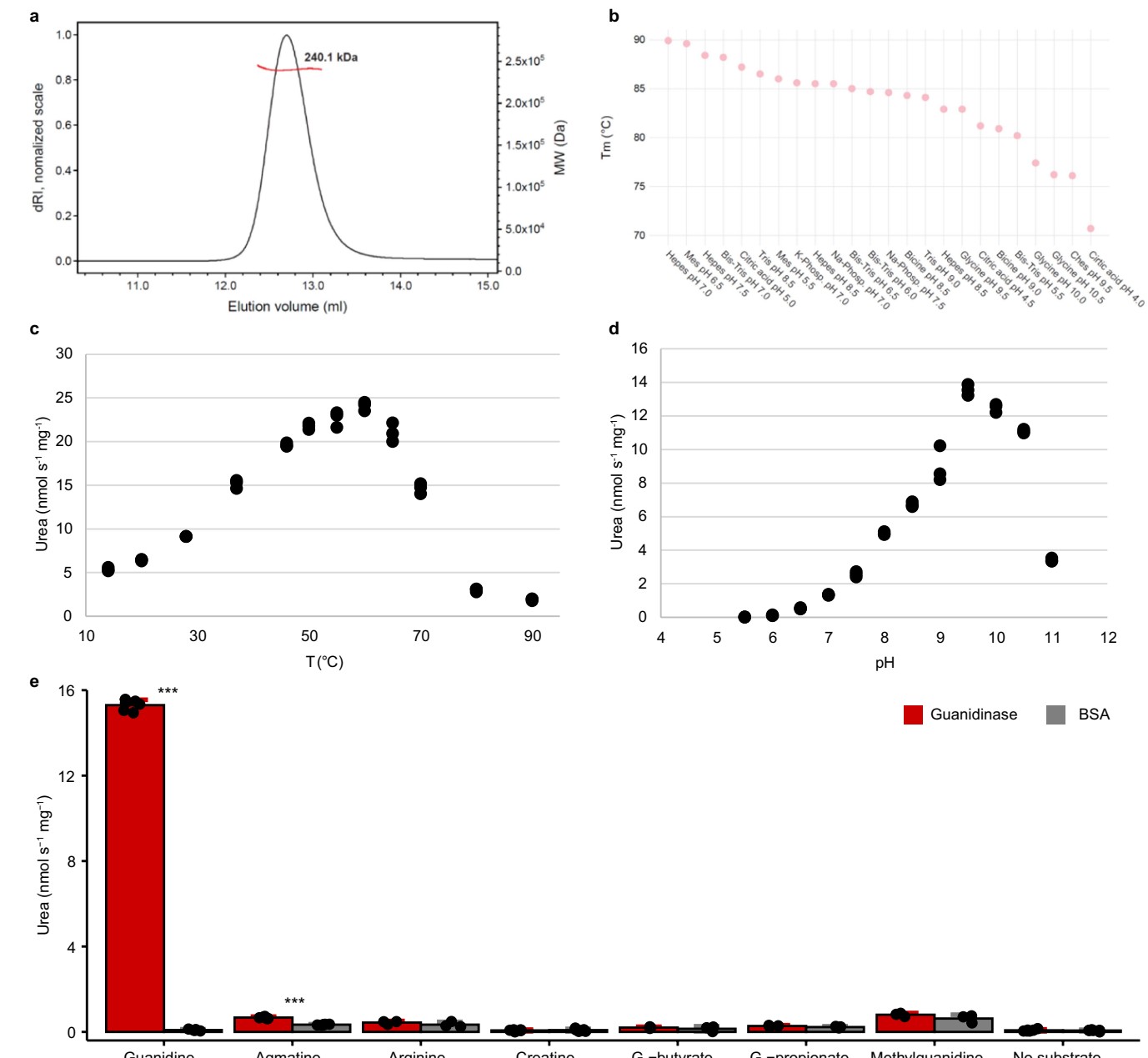

**Extended Data Fig. 5 | Oligomeric state thermostability, pH and temperature dependence and possible alternative substrates of the heterologously expressed *N. inopinata* guanidinase.** (**a**) Size exclusion chromatography coupled to multi-angle-light scattering (SEC-MALS) elution profile of the guanidinase. The red line crossing the SEC profile indicates the molecular mass of the protein. Guanidinase forms hexamers in solution as the calculated molecular mass of a subunit is 41.66 kDa. (**b**) Melting temperatures ($T_m$) of guanidinase in various buffer/pH conditions estimated using the two-state reversible unfolding model implemented in MoltenProt[75]. 24 buffer/pH conditions with the highest $T_m$ are shown. (**c**) Temperature dependence of heterologously expressed *N. inopinata* guanidinase at pH 7, 10 mM guanidine starting concentration. (**d**) pH dependence of heterologously expressed *N. inopinata* guanidinase at 37 °C, 1 mM guanidine starting concentration. (**e**) Urea production by heterologously expressed *N. inopinata* guanidinase from guanidine and different guanidino compounds. 10 mM of substrates were incubated with guanidinase or BSA at 37 °C. Bars show the means, error bars the standard deviation of 3 replicates (for arginine, guanidinobutyrate, guanidinopropionate, methylguanidine) or 6 replicates (guanidine, except BSA control 5 replicates; agmatine, creatine, no substrate). \*\*\* Urea production from guanidino compounds when incubated with guanidinase versus bovine serum albumin (BSA). Guandine one-sided *t*-test: $t = 133.92635$, d.f. = 9, $P = 1.83 \times 10^{-16}$. Agmatine one-sided *t*-test: $t = 17.44749$, d.f. = 10, $P = 4.06 \times 10^{-9}$.

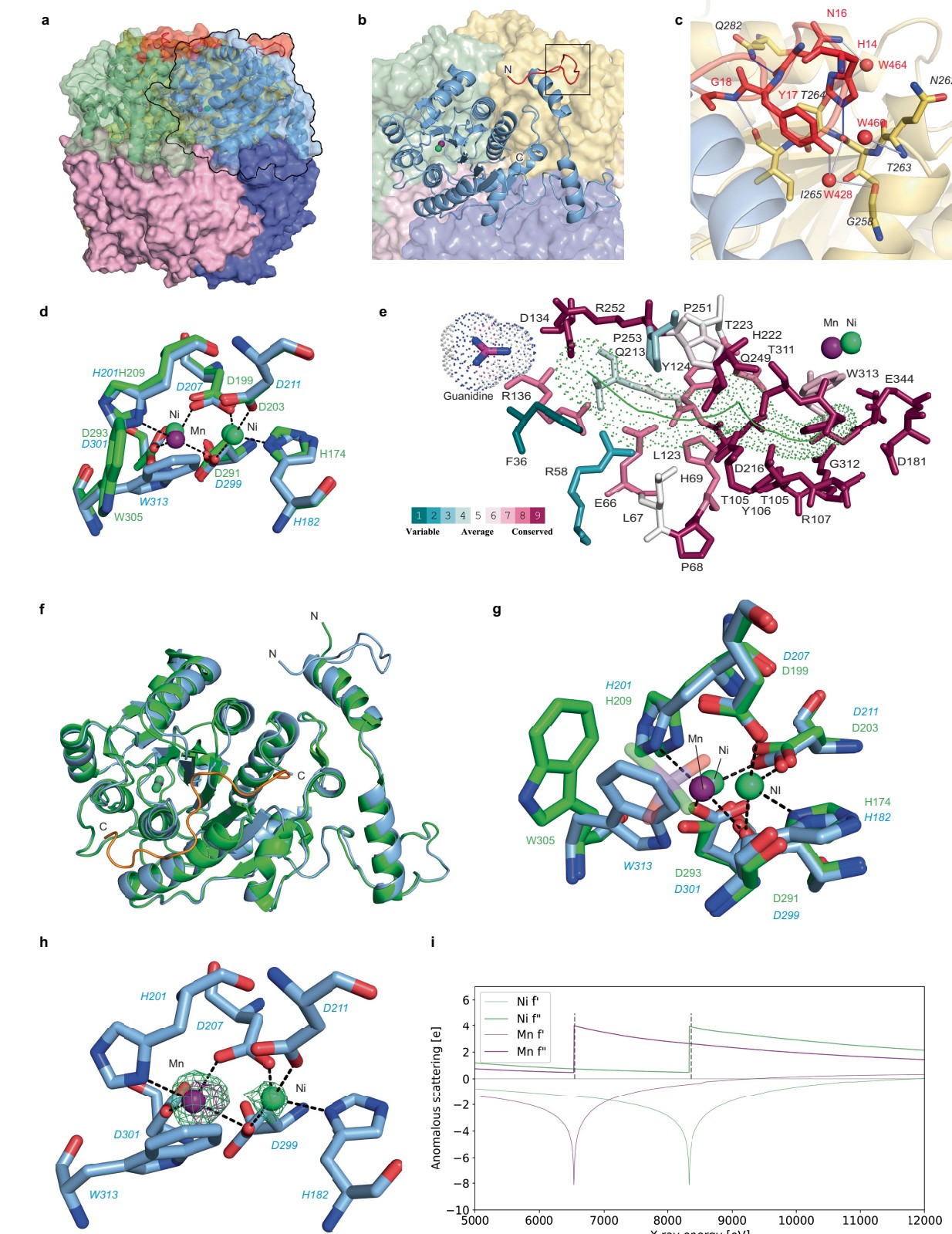

**Extended Data Fig. 6** | See next page for caption.

**Extended Data Fig. 6 | Structure of the *N. inopinata* guanidinase and comparison to the *Synechocystis* guanidinase.** (**a**) Overview of the *N. inopinata* guanidinase hexamer shown with surface representation, subunits are individually coloured. Nickel ions are shown as green, manganese ions as purple spheres. The N-terminal extension of 14 amino acids which partly protrudes together with the first N-terminal helix over the neighbouring subunit is highlighted in red (PDB ID 9FEK). (**b**) Cartoon representation of the *N. inopinata* guanidinase subunit with the colour code for the N-terminal extension and metal ions as in (a). (**c**) Close up view of the interaction of the *N. inopinata* N-terminal extension (red cartoon and sticks) with the residues of the neighbouring subunit (yellow cartoon and sticks) through three hydrogen bonds and four water bridges (Supplementary Table 7). Coordinated waters are shown as red spheres. (**d**) Superposition of the active sites of guanidinase from *N. inopinata* (green) and the *Synechocystis* GdmH (7OI1) (cyan). Nickel ions are shown as green, manganese ions as purple spheres. (**e**) Model of guanidine and the tunnel shown as in Fig. 2d. Residues lining the tunnel are shown as sticks and colour coded according to conservation value, as provided by ConSurf[97]. The tunnel has an average width of 1.4 Å and is flanked by highly conserved residues. The width of the tunnel corresponds to the size of a water molecule and would not allow the passage of the larger guanidinium cation. Hence, some molecular plasticity/flexibility is needed to allow minor rearrangements leading to the required widening of the tunnel. (**f**) Subunit comparison between *Synechocystis* (green) and *N. inopinata* guanidinases (cyan), with C-terminal extension of *Synechocystis* (residues 369–386 of 7OI1) highlighted in orange, which is engaged in a series of interactions with the same subunit on the solvent-exposed side of the hexamer (Supplementary Table 8), and which is absent from the *N. inopinata* guanidinase. Nickel ions are shown as green, manganese ions as purple spheres. (**g**) Superposition of the active sites of *Synechocystis* (green, PDB ID 7OI1) and *N. inopinata* guanidinases (light blue, PDB ID 9FEK). Particularly noteworthy is the tilt of *N. inopinata* guanidinase Trp313 [$\chi1$ (N, C$\alpha$, C$\beta$, C$\gamma$) = 51.3°] towards the active site nickel/manganese ions, compared to the corresponding Trp305 in *Synechocystis* GdmH 7OI1 [$\chi1$ (N, C$\alpha$, C$\beta$, C$\gamma$) = −80.5°] which is flipped out, creating a cavity occupied by ethylene glycol or a cacodylate ion (PDB ID 7ESR). Nickel ions are shown as green, manganese ions as purple spheres. Cacodylate ion from *Synechocystis* guanidinase (PDB 7OI1) is shown as transparent sticks. (**h**) Active site of *N. inopinata* guanidinase (light blue) with averaged anomalous difference fourier maps contoured at 2 rmsd. The nickel ion is shown as green sphere, the nickel anomalous map is shown as a green mesh. The manganese ion is shown as purple sphere, the manganese anomalous map is shown as a purple mesh. (**i**) Anomalous scattering signal vs x-ray energy plotted for manganese (purple) and nickel (green) ions using data obtained from http://skuld.bmsc.washington.edu/scatter/AS_form.html, accessed on 20.02.2024. X-ray energies used to collect datasets of anomalous maps from C are indicated by dashed lines.

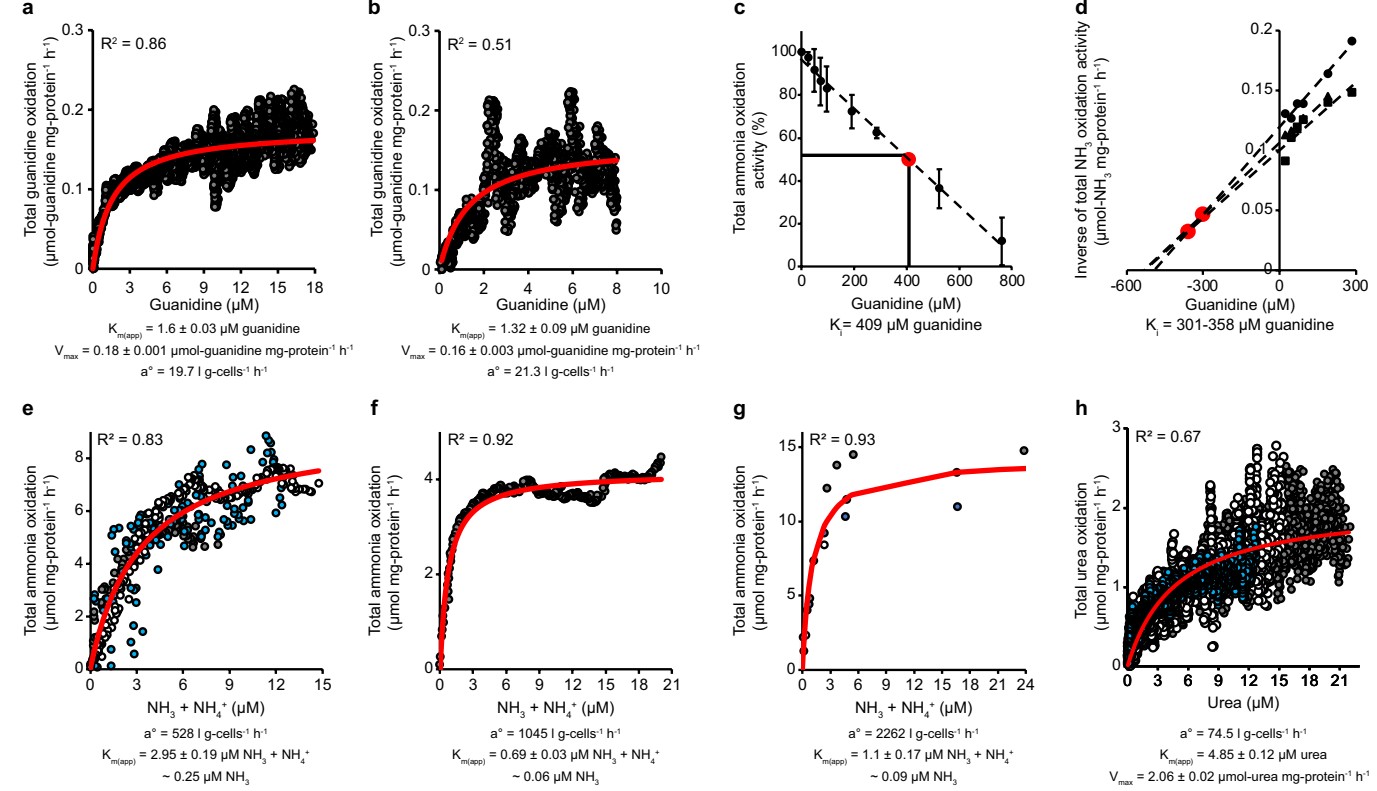

**Extended Data Fig. 7 | Extended kinetics parameters of *N. inopinata*.**
(**a**) and (**b**) Additional replicates for whole-cell guanidine oxidation rates of
*N. inopinata* pre-induced with guanidine for ~12 h. A Michaelis-Menten model
(red line) was used to determine the $K_{m(app)}$, and $V_{max}$. (**c**) The effect of guanidine
on the rate of total ammonia oxidation (points are means ± s.d., n = 3). A linear
regression was fit (dashed line), and the 50% inhibition rate is marked in red.
(**d**) A Dixon reciprocal plot for the inhibition of total ammonia oxidation by
guanidine (n = 3; circles, triangles, and squares; the same biological replicates
from panel (c) were used). Ammonia oxidation rates determined at guanidine
concentrations that formed a linear inhibition trend were used to fit a linear
regression for each replicate (dashed lines). Regression intersection points
(red dots) were used to determine the inhibition constant value ($-K_i$) (**e**-**h**)
Michaelis-Menten plots for *N. inopinata* pre-grown in medium containing
ammonium and guanidine (~1 mM and ~15 μM, respectively for ~12 h, (**e**)), urea

(0.5 mM for >1 week (**f** and **h**)), or ammonium (~1 mM for >1 week (**g**)). Data
presented in panel (**g**) was previously published and is shown for comparison[48].
Total ammonia (**e**, **f**, and **g**) and urea oxidation rates (**h**) were determined
from $O_2$ consumption rates and normalized to total protein concentration.
A Michaelis-Menten model (red line) was used to determine the $K_{m(app)}$ and $V_{max}$.
The $K_{m(app)}$ for unprotonated ammonia was calculated from total ammonium
kinetics. In all cases, biological replicates are shown in white, blue, and grey.
It should be noted that recently whole-cell urea oxidation kinetics of *N. inopinata*
were also reported in another publication and differ from those shown here[33].
Qin et al., reports a $K_{m(app)}$ of ~0.14 μM urea, a $V_{max}$ of ~8 μM urea mg protein$^{-1}$ h$^{-1}$,
and an $a^o$ of ~10,000 l g cells$^{-1}$ h$^{-1}$. In addition, no difference in the $a^o$ for $NH_3$ or
urea for *N. inopinata* was reported. While the cause of these discrepancies is
unknown, it is possible that this reflects differences related to concentrating
cultures or lack thereof before microrespiration measurements.

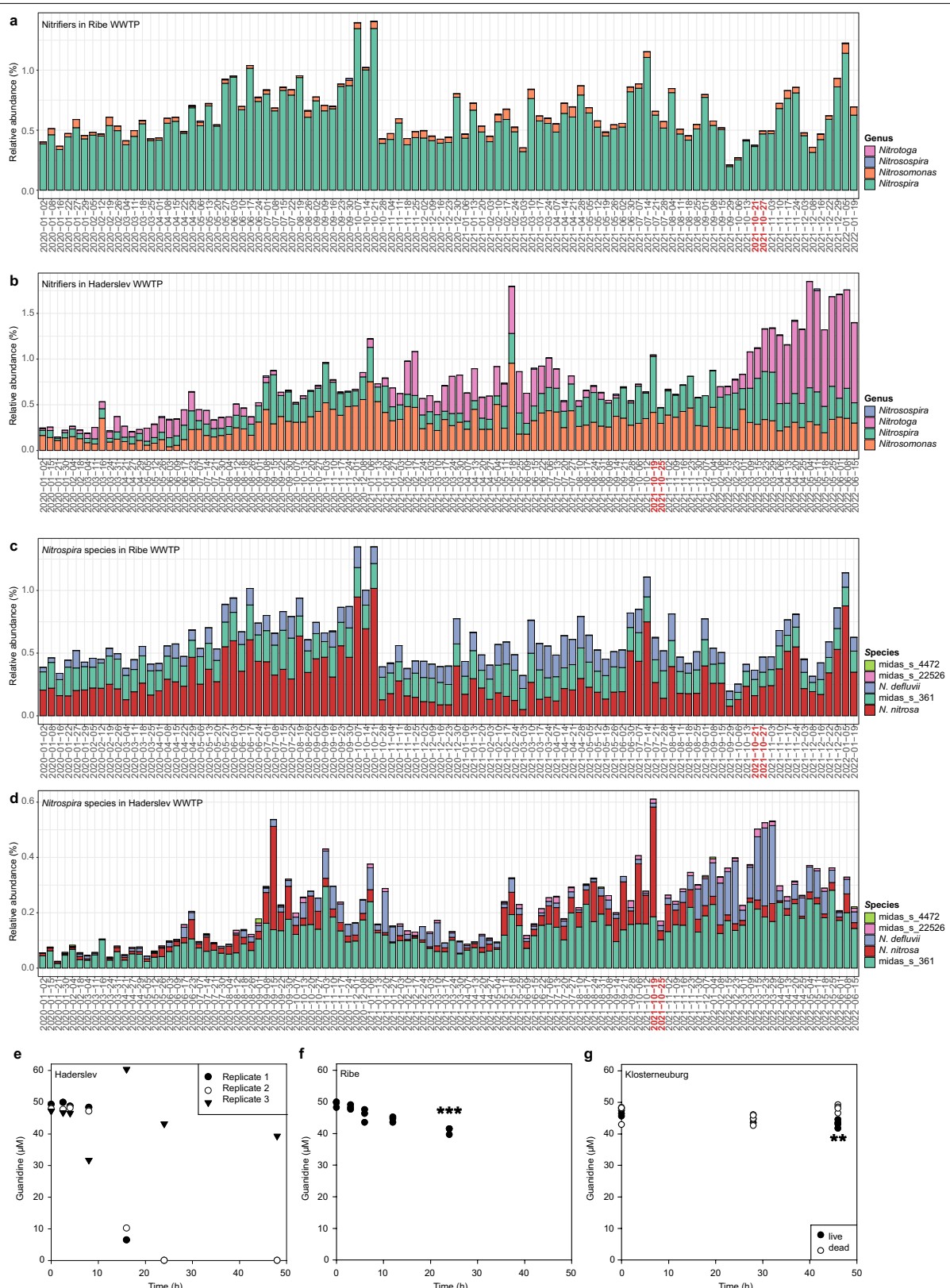

**Extended Data Fig. 8** | See next page for caption.

**Extended Data Fig. 8 | Community structure of nitrifying bacteria in the WWTPs Ribe and Haderslev, and Guanidine degradation in WWTPs with and without comammox microbes.** (a-d) V1-V3 16 S rRNA gene amplicon sequencing revealed that, over a period of two years, (**a**) the nitrifying community in WWTP Ribe was dominated by members of the genus *Nitrospira*, while co-occurring AOB related to *Nitrosomonas* and *Nitrosospira* were less abundant. (**b**) The nitrifying community in WWTP Haderslev was more diverse and contained, in addition to the consistently abundant *Nitrospira*, also relatively large populations of *Nitrosomonas* and nitrite-oxidizing bacteria from the genus *Nitrotoga*, whereas *Nitrosospira*-related AOB were rare. The *Nitrospira* populations in the two WWTPs consisted of organisms closely related with the comammox species *N. nitrosa*, the canonical nitrite oxidizer *N. defluvii*, and yet uncharacterized *Nitrospira*-like bacteria referred to as 'midas_s_361', 'midas_s_4472' (both WWTPs), and 'midas_s_22526' (only WWTP Haderslev) (**c** and **d**). Samples taken shortly before and after the sampling of activated sludge for metatranscriptomic analyses from both plants are labelled in red. The partial 16 S rRNA gene sequence midas_s_361 shows >99% identity to a 16 S rRNA gene from a lab-scale reactor *Nitrospira* sp. UBC3 MAG (GCA_022226955), which possesses no genes for ammonia oxidation. The midas_s_4472 16 S rRNA

and midas_s_22526 16 S rRNA gene sequence are related to the canonical nitrite oxidizers *N. marina* and *N. defluvii*, respectively. Thus, most likely these three partial 16 S rRNA gene sequences do not represent comammox organisms. (**e-g**) 50 μM guanidine was added to the (**e**) Haderslev, (**f**) Ribe (both with comammox), and (**g**) Klosterneuburg (no comammox) activated sludge samples and guanidine concentrations were determined over time. Dead controls showed no significant changes in guanidine concentrations, suggesting that chemical degradation or sorption of guanidine did not influence the results. The activated sludge biomass in the incubations was normalized by Total Suspended Solid levels and subsequently confirmed by BCA protein content measurements. Relative guanidine degradation rates normalized by protein content: Haderslev active replicates: 12.9 μM per h; Ribe: 0.95 μM per h; Klosterneuburg living: 0.19 μM per h. All incubations were done in triplicates. Two-tailed *t*-test of significance: panel (b): *** between 0 h and 24 h timepoints: $t = 10.418$, d.f. = 4, $P = 0.000480$; panel (c): ** Living biomass between 0 h and 48 h timepoints: $t = 4.602$, d.f. = 6, $P = 0.00369$; Dead biomass between 0 h and 48 timepoints: no significant difference. We have no good hypothesis, why one replicate of the Haderslev sample showed different results (e).

**Extended Data Table 1 | Comparison of kinetic properties and metal loading of purified guanidinase enzymes from *N. inopinata* and *Synechocystis***

| Source of guanidinase enzyme (kinetics pH) | $K_m$ (mM) | $V_{max}$ (nmol s$^{-1}$ mg$^{-1}$) | $K_{cat}$ (s$^{-1}$) | $K_{cat}/K_M$ (s$^{-1}$ mM$^{-1}$) | Ni loading (atom / subunit) | Mn loading (atom / subunit) | Reference |
|---|---|---|---|---|---|---|---|
| *N. inopinata;* 0.5 µM Ni in expr. (pH=7.5) | 11.2 ±0.51 | 0.56 ±0.42 | 0.013 | 0.0012 | 0.26 ±0.004 | 0.18 ±0.006 | This study |
| *N. inopinata;* 20 µM Ni in expr. (pH=7.5) | 16.1 ±0.40 | 2.02 ±0.02 | 0.083 | 0.0052 | 0.35 ±0.016 | 0.55 ±0.025 | This study |
| *N. inopinata;* 1 mM Ni in expr. (pH=7.5) | 13.6 ±0.76 | 34.73 ±0.62 | 1.42 | 0.1044 | 0.42 ±0.006 | 0.03 ±0.001 | This study |
| *Synechocystis* PCC6803 (pH=7.5) | 5.3 ±1.2 | 0.91 ±0.23 | 0.040 | 0.0075 | | | Wang *et al.*, 2021 [17] |
| *Synechocystis* PCC6803 (pH=8) | 7.8 ±1.8 | 84.8 ± 6.79 | 3.830 | 0.4337 | 1.9 | 1 | Funck *et al.*, 2022 [18] |

The *Synechocystis* enzyme was either overexpressed in *Synechocystis*[15,17,18] or in *E. coli*[18]. In the latter heterologous expression experiment, Ni-loading chaperons of *Synechocystis* were co-overexpressed. For the *N. inopinata* guanidinase, three different nickel concentrations (indicated after the source species name) were used during expression. For the 20 µM nickel expression batch, nickel was present during all purification steps in the same concentration, and only removed for the ICP-MS analysis of metal loading. This was not possible for the 1 mM Ni$^{2+}$ treatment as it resulted in protein denaturation during purification.

**Extended Data Table 2 | Guanidine concentrations in animal and WWTP samples**

| Sample | Biological Replicates | Concentration ($\mu$mol kg$^{-1}$ (dry weight) or $\mu$mol l$^{-1}$) |
|---|---|---|
| **Samples measured in this study** | | |
| Cow Urine* | 5 | 13.6 ± 4.5 |
| Sheep Urine | 2 | 6.50 - 6.62 |
| Cow Feces* | 3 | 150 ± 17 |
| Sheep Feces | 4 | 69 ± 22 |
| Chicken Feces | 4 | 108 ± 46 |
| Pig Feces | 4 | 143 ± 40 |
| WWTP Klosterneuburg Influent | 1 | 0.5 |
| **Literature data** | | |
| Human Urine [36,37] | 22 | 2.17 - 19.8 |
| Rosettes of *Arabidopsis* [14] | 4 | 100-200 |
| Legume Seeds [14] | 3 | 2400 |

Concentrations are reported as mean±s.d. or as a range of biological replicates in µmol per kg (dry weight) for faecal and plant samples or µmol per l for urine and wastewater samples[14,35,36]. Based on the guanidine concentrations measured in cow faeces and urine samples (*), we roughly extrapolated to the amount of guanidine that is globally excreted by cows. Using the measured water content of our cow faeces samples (mean: 57% (w/w)) and the molar mass of guanidine, we calculated a guanidine concentration per faeces wet mass of 3.8 mg/kg. With an estimated global cow population of 940 million and estimated amounts per cow and day for feces (30 kg) and urine (13 L), we obtained > 100 tons of guanidine globally excreted by cows per day. Cow, sheep, pig and chicken samples were collected at the University of Veterinary Medicine Vienna.

# Reporting Summary

## Statistics

For all statistical analyses, confirm that the following items are present in the figure legend, table legend, main text, or Methods section.

| n/a | Confirmed | |
|---|---|---|
| ☐ | ☒ | The exact sample size (*n*) for each experimental group/condition, given as a discrete number and unit of measurement |
| ☐ | ☒ | A statement on whether measurements were taken from distinct samples or whether the same sample was measured repeatedly |
| ☐ | ☒ | The statistical test(s) used AND whether they are one- or two-sided<br>*Only common tests should be described solely by name; describe more complex techniques in the Methods section.* |
| ☐ | ☒ | A description of all covariates tested |
| ☐ | ☒ | A description of any assumptions or corrections, such as tests of normality and adjustment for multiple comparisons |
| ☐ | ☒ | A full description of the statistical parameters including central tendency (e.g. means) or other basic estimates (e.g. regression coefficient) AND variation (e.g. standard deviation) or associated estimates of uncertainty (e.g. confidence intervals) |
| ☐ | ☒ | For null hypothesis testing, the test statistic (e.g. *F*, *t*, *r*) with confidence intervals, effect sizes, degrees of freedom and *P* value noted<br>*Give P values as exact values whenever suitable.* |
| ☒ | ☐ | For Bayesian analysis, information on the choice of priors and Markov chain Monte Carlo settings |
| ☒ | ☐ | For hierarchical and complex designs, identification of the appropriate level for tests and full reporting of outcomes |
| ☒ | ☐ | Estimates of effect sizes (e.g. Cohen's *d*, Pearson's *r*), indicating how they were calculated |

*Our web collection on statistics for biologists contains articles on many of the points above.*

## Software and code

Policy information about availability of computer code

| Data collection | hmmsearch, infernal v1.1.3, XDS, SensorTrace Rate,  bbmap v38.92, C1000-CFX96 BioRad, NovaSeq 6000 |
|---|---|
| Data analysis | SigmaPlot v14.5, MS Excel 2016, IQ-TREE2, Proteome Discoverer v.2.5, Astra, MoltenProt, MaxEnt1, XDSCONV, Phaser-MR, AlphaFold, COOT, PHENIX, MolProbity, PDBREDO, PyMol v 2.0, DESeq2, R (packages: DEqMS, phangorn, ape, ggtree, ggseqlogo, Decipher,phytools), usearch, FastTree2, OpenMIMS v 3.0.5, Fiji ImageJ 1.54f, bbmap v38.92, bbduk, bedtools |

For manuscripts utilizing custom algorithms or software that are central to the research but not yet described in published literature, software must be made available to editors and reviewers. We strongly encourage code deposition in a community repository (e.g. GitHub). See the Nature Portfolio guidelines for submitting code & software for further information.

## Data

Policy information about availability of data

All manuscripts must include a data availability statement. This statement should provide the following information, where applicable:
- Accession codes, unique identifiers, or web links for publicly available datasets
- A description of any restrictions on data availability
- For clinical datasets or third party data, please ensure that the statement adheres to our policy

The crystal structures have been deposited in the Protein Data Bank under Code: 9FEK. WWTP metatranscriptome reads have been deposited under BioProject ID PRJNA1118285. The mass spectrometry proteomics data have been deposited to the ProteomeXchange Consortium via the PRIDE 94 partner repository with the

## Research involving human participants, their data, or biological material

Policy information about studies with human participants or human data. See also policy information about sex, gender (identity/presentation), and sexual orientation and race, ethnicity and racism.

| | |
|---|---|
| Reporting on sex and gender | Not applicable. No human participants were used in this study. |
| Reporting on race, ethnicity, or other socially relevant groupings | Not applicable. No human participants were used in this study. |
| Population characteristics | Not applicable. No human participants were used in this study. |
| Recruitment | Not applicable. No human participants were used in this study. |
| Ethics oversight | Not applicable. No human participants were used in this study. |

Note that full information on the approval of the study protocol must also be provided in the manuscript.

# Field-specific reporting

Please select the one below that is the best fit for your research. If you are not sure, read the appropriate sections before making your selection.

☐ Life sciences          ☐ Behavioural & social sciences          ☒ Ecological, evolutionary & environmental sciences

For a reference copy of the document with all sections, see nature.com/documents/nr-reporting-summary-flat.pdf

# Ecological, evolutionary & environmental sciences study design

All studies must disclose on these points even when the disclosure is negative.

| | |
|---|---|
| Study description | Ability of Nitrospira inopinata to grow on guanidine as the sole source of energy, reductant, and nitrogen was tested. Proteomics, enzyme kinetics, and the crystal structure of a N. inopinata guanidinase homologue was determined. Transcription of comammox guanidinases was induced in wastewater treatment plant microbiomes upon incubation with guanidine and the nitrifying community which also included comammox, present in natural soil used guanidine as a substrate. |
| Research sample | Nitrospira inopinata (complete ammonia-oxidizer) bacterial pure culture. Heterologously expressed guanidinase homologue from N.inopinata. Activated sludge samples from wastewater treatment plants. Agricultural soil. Dropped animal manure and urine from cows, pigs, sheep and chicken. |
| Sampling strategy | For physiological experiments 3 - 5 biological replicates were performed. For shotgun proteomics, 5 biological replicates were performed. |
| Data collection | Data was collected by small batch culturing of bacteria, chemical analysis, qPCR, shotgun proteomics, chrystal structure analysis, ICP-MS and metatranscriptomics by Christopher J. Sedlacek, Kenneth Wasmund, Nico Jehmlich, Richard Gruseck, Julius Kostan, Dominic Pühringer, Katharina Kitzinger, Andrew Giguere and Marton Palatinszky |
| Timing and spatial scale | Activated sludge samples were collected from the aerated tanks of the Ribe and Haderslev WWTPs on October 22, 2021, Klosterneuburg WWTP on June 1, 2022. Agricultural soil was collected from 48°11'17.9"N 14°15'16.5"E on May 9, 2023. |
| Data exclusions | No data was excluded from analysis. |
| Reproducibility | All performed experiments are reported - even if they show different behaviour in one replicate. |
| Randomization | Biological triplicates, quadruplicates or quintuplicates per treatment were considered as one experimental group. |
| Blinding | Blinding was not applicable as the study does not involve living animal or human individuals. |

Did the study involve field work?    ☒ Yes    ☐ No

# Field work, collection and transport

| | |
|---|---|
| Field conditions | Activated sludge from three wastewater treatment plants were sampled. Agricultural soil was sampled from 48°11'17.9"N 14° |

| Field conditions | 15'16.5"E under mild weather conditions. |
|---|---|
| Location | Ribe, Denmark (GPS: 55.33, 8.74), Haderslev, Denmark (GPS: 55.25, 9.51), Klosterneuburg, Austria (GPS: 48.29, 16.34), Agricultural soil 48°11'17.9"N 14°15'16.5"E |
| Access & import/export | Sampling of the wastewater treatment plants happened under the supervision of the respective facility management personnel. Living cultures did not cross any borders, they were incubated and sub-sampled in local laboratories. Austrian agricultural soil was sampled and all experiments were performed in Austria, which does not require specific permits. |
| Disturbance | Sampling the tanks of the wastewater treatment plants or the agricultural soils (longterm field experiment from AGES Austrian Agency for Health and Food Safety) did not cause disturbance to their microbial communities. |

# Reporting for specific materials, systems and methods

We require information from authors about some types of materials, experimental systems and methods used in many studies. Here, indicate whether each material, system or method listed is relevant to your study. If you are not sure if a list item applies to your research, read the appropriate section before selecting a response.

## Materials & experimental systems

| n/a | Involved in the study |
|---|---|
| ☒ ☐ | Antibodies |
| ☒ ☐ | Eukaryotic cell lines |
| ☒ ☐ | Palaeontology and archaeology |
| ☒ ☐ | Animals and other organisms |
| ☒ ☐ | Clinical data |
| ☒ ☐ | Dual use research of concern |
| ☒ ☐ | Plants |

## Methods

| n/a | Involved in the study |
|---|---|
| ☒ ☐ | ChIP-seq |
| ☒ ☐ | Flow cytometry |
| ☒ ☐ | MRI-based neuroimaging |

## Plants

| Seed stocks | not applicable |
|---|---|
| Novel plant genotypes | not applicable |
| Authentication | not applicable |

