## [Peer Review File · Nature]

Manuscript Title: Growth of complete ammonia oxidizers on guanidine

Reviewer Comments & Author Rebuttals

Reviewer Reports on the Initial Version:

Referees' comments:

Referee #1 (Remarks to the Author):

Manuscript Growth of complete ammonia oxidizers on guanidine

By Palatinszky et al describes the growth of the complete ammonia oxidiser *Nitrosospira* on guanidine and the elucidation of its guanidinase enzyme system.

Guanidine is an important nitrogenous compound, but there is virtually no record of microorganisms growing on guanidine. The present study elegantly employed various complementary techniques such as proteomics, enzyme kinetics, and crystallography to demonstrate that comammox bacteria use guanidine as sole nitrogen and energy source, and that they contain a genuine guanidinase.

Furthermore, the transcription of comammox guanidinase gene was observed to be induced in wastewater treatment plant microbiomes when incubated with micromolar quantities of guanidine. The discovery of guanidine as a selective growth substrate for comammox bacteria provides a very valuable insight into the unique niche of these important nitrifiers and presents novel opportunities for their isolation. Overall, this is an excellent contribution to the field of microbiology and environmental biotechnology. At some points in the manuscript, the authors exaggerate their findings and could tune down a bit their claim.

Specific comments

Line 24 "it occurs widely in the environment" would need proper documentation on how much guanidine really occurs in ecosystems

Line 33-35 are overselling the scientific value by a large extend

Fig 1 do the depicted microbes have urease elsewhere in the genome? That is not so clear. In line 105 something is mentioned

Line 75 "massive N loss" please quantify and tune down statement with proper reference

Line 81 "very narrow substrate range" is in sharp contrast to the titles of other studies by the consortium that highlight the versatility of this group of nitrifiers

Expanded metabolic versatility of ubiquitous nitrite-oxidizing bacteria from the genus *Nitrospira*. Koch et al 2PNAS 015

Activity and metabolic versatility of complete ammonia oxidizers in full-scale wastewater treatment systems Yang et mBio 2020

A nitrite-oxidising bacterium constitutively consumes atmospheric hydrogen
PM Leung et al ISME J 2022

Line 81 the abbreviation AOM can represent many other processes, seems an odd choice to me

Line 106 "much more energy efficient" you can not compare N source with an energy source in terms of efficiency, please rephrase

Line 112 why was 50 micromolar chosen as the start concentration?

Would the other nitrifiers have converted guanidine if you used a lower or higher concentration and if you would have waited longer? Now it seems everything was geared towards comammox.

Line 134-136 does comammox occurrence and guanidine in ecosystems match? Or are ammonia and oxygen more strong drivers for niche separation?

Line 174 ref 24 does not tell us much on the inhibition of guanidine on growth of nitrifiers.

For guanidine to be toxic it must have entered the cells; Is uptake of guanidine inhibited by the presence of urea or ammonia?

Line 251 the affinity of 16 mM is in sharp contrast to the claimed low environmental concentrations?

How do the cells take up the guanidine? Is there a large internal pool?

Line 269: 200 μM is still a very high K_s value for cells, if environmental guanidine concentrations are very low and conversion is not very efficient. Please provide K_s and V_{max} values of urea and ammonia by whole cells of comammox to put the values in perspective

Line 298 elegant experiments with WWTP sludge indicate that comammox bacteria are geared to handle guanidine. Can transcripts of the comammox guanidinase gene be found in the transcripts of the native WWTP biomass? This would be an indication that this conversion is actually happening in situ. Maybe there are studies on micropollutants that have measured in and out flow concentrations of metformin and/or guanidine that could give you an indication of this in situ metabolism.

The discussion contains quite some iteration of previous points and could be condensed.

Referee #2 (Remarks to the Author):

Palatinszky et al. report the use of guanidine as a source of energy, reductant and nitrogen by comammox microbes. Overall, the manuscript is well written and shows convincingly that comammox bacteria can produce nitrite and nitrate upon guanidine addition, strongly suggesting that after breakdown to ammonia it can be used as an energy source. This is a novel result and suggests a new role for guanidine beyond what was shown in Funck et al (2022) and Wang et al (2021), who showed guanidine N assimilation.

The evidence provided in the main manuscript that guanidine is used by the culture of *Nitrospira inopinata* as a nitrogen source and to gain energy for growth is less convincing (see below). Similarly, there is not sufficient support for the conclusion that guanidine utilization provides a niche for comammox bacteria in the environment. Overall, the manuscript is very long and could be more focused. Below, I made some suggestions how to reduce its length to better emphasize the main new finding – guanidine is an alternative energy source for comammox bacteria.

1) Growth of *Nitrospira inopinata* on guanidine

The title of the manuscript is “Growth of complete ammonia oxidizers on guanidine” and the abstract states that “*Nitrospira inopinata* and likely most other comammox microbes grow on guanidine as the sole source of energy, reductant, and nitrogen”. However, the evidence provided in the main manuscript that these microorganisms grow on guanidine is weak. In figure 3, the authors show results from a batch incubation in which ~50 μM guanidine was added to a washed culture of *N. inopinata* that had been pre-incubated guanidine for a week. The guanidine is clearly converted into nitrite and nitrate, which clearly shows that it is used by these microorganisms as an electron donor. However, since there is no concomitant change in the 16S rRNA gene copies, there is no direct evidence that *Nitrospira inopinata* is growing in the first 10 days. In a second set of batch incubations presented in the extended data figure 4, there is an increase in gene copies; however, the data is at a very low temporal resolution. Since this is the main finding of the current manuscript, I would expect a more compelling growth curve (with more data points) based on an increase in cell numbers, OD600 or 16S rRNA copy numbers. The direct link between nitrate and nitrite production and growth is missing, and this is particularly essential since there is an imbalance in the nitrogen budget (as mentioned in the legend figure 3). Without more compelling evidence, it cannot be ruled out that the use of guanidine is instead a survival mechanism.

2) Use of guanidine as a nitrogen source

Based on the imbalance in the nitrogen budget, the presented data does not provide evidence for direct use of guanidine as an N source for assimilation. This could be provided by using ^{15}N labeled guanidine followed by bulk isotope ratio mass spectrometry or nanoSIMS.

3) Purified enzyme kinetics and crystallography

Even though the crystallography data and the analyses of the crystal structure is very well done, it does

not make a substantial advance on what was shown in the previous guanidine hydrolase papers (Funck et al. (2022) and Wang et al. (2021)). I would suggest shortening this section significantly and moving much of the data to a supplement as it is not relevant to the main novel message of the manuscript that these microorganisms gain energy from metabolizing guanidine.

It is surprising that the purified protein kinetics are worse than the whole cell kinetics. The authors state that this might be because the enzyme is not nickel loaded, which could well be the case. However in order to advance the previous work, it would have been interesting to attempt to solve this by using higher Ni concentrations or using Ni in the purification buffer at different concentrations. In addition, considering that the work is carried out using a pure culture, the authors could directly purify the enzyme from the native cells and characterize the protein afterwards. If this is not feasible, at least the enzyme kinetics measurements could be carried out on cell-free extracts. This would be novel and add more to our understanding of this enzymatic pathway.

The purified protein is a part of the arginase family. In the paper by Funck et al., activity of the guanidinase was also tested on alternative substrates such as arginine, creatine and other guanidine-containing compounds. I would suggest the authors to determine whether the enzyme they purified is a specific guanidinase as reported for the one in the Funck et al. paper or whether it has a wider range of substrate spectrum. The rather low affinity of the purified enzyme might as well point towards the possibility that the guanidine hydrolysis is rather the side reaction of this enzyme rather than its bona fide activity.

Furthermore, the manuscript states that the enzyme has a high thermostability, but activity is not measured over a range of temperatures or pH. Instead, this conclusion is based on a protein unfolding model, however, the enzyme might stop being active long before it “unfolds”. Therefore, further activity assays are needed to support the conclusion.

4) Whole cell kinetics

I am wondering why guanidine, ammonium and urea were not directly measured in the whole cell kinetics assays. I think these experiments should be repeated with these measurements, which would make them more compelling. Particularly considering that these compounds were used to determine the kinetics for the purified enzyme. I understand that this data would not be at the same resolution as the O₂ respirometry, but it would be important to show what is actually occurring as currently the range in which guanidine appears to be utilized is rather narrow and very high.

5) Wastewater treatment sludge experiments

A large section of the manuscript is devoted to experiments carried out using wastewater treatment sludge which the authors added guanidine to and subsequently examined transcription of guanidinase and ACP. However, it is unclear what the aim of these experiments were. If guanidine utilization is an important factor in the success of commamox bacteria in WWTP, then why are the relevant enzymes

not already being transcribed? This might suggest that guanidine is not a substrate the comammox microorganisms encounter in WWTP. These experiments suffer from the lack of in situ guanidine measurements, which could easily be measured using the newly developed assay.

If these experiments were carried out with the intention of supporting the statement that guanidine can be used to enrich comammox bacteria, then I would expect to see a comammox enrichment on guanidine. The use of guanidine as a potential substrate to enrich comammox is a key point in the discussion and abstract, but so far no evidence is shown to support this statement.

6) Guanidine utilization as a niche for comammox bacteria

The abstract states that

“Nitrospira inopinata and likely most other comammox microbes grow on guanidine as the sole source of energy, reductant, and nitrogen” and that “The discovery of guanidine as selective growth substrate for comammox shows a unique niche of these globally important nitrifiers”

In the first case, the text should be adjusted to say that they can use guanidine as a source of energy (and if shown more convincingly, nitrogen). The current text implies that they grow solely off guanidine in the environment, which is not supported by the data provided in the manuscript. Similarly with the second sentence, I do not see that guanidine utilization is a niche for comammox in the environment. *N. inopinata* has an apparent substrate affinity ($K_m(\text{app})$) for guanidine of $\sim 202 \mu\text{M}$ – in comparison its affinity for ammonium is $0.84 \mu\text{M}$. Do the authors expect that guanidine would therefore be used in environments where ammonium is already present and that the presence of this compound therefore provides a niche for these bacteria? In order to keep this statement, the authors must provide more evidence that there is a niche in the environment where guanidine might be important – which likely would require more measurements of its concentration in situ.

Minor comments:

Please do not use the term AOM, it is also used to refer to anaerobic oxidation of methane and therefore can lead to confusion. “Ammonia oxidizers” could be used instead of AOM.

In the WWTP section the title of the figure “Metatranscriptomic response to guanidine amendment by comammox activated sludge microbiome members” is confusing and should be adjusted.

What is meant by “ancient trait for comammox”, ancient implies you have an idea as to the timing. Rephrase.

Reformulate sentence in abstract:

“It occurs widely in nature and is used by microbes as a nitrogen source, but microorganisms growing on

guanidine have not yet been discovered”

For the general readership of Nature, it is probably unclear what the difference is between guanidine being used as a nitrogen source and microbes growing on guanidine. This sentence could be rephrased as: “It ... microorganisms using guanidine as an energy source”

The meaning of “Riboswitch” and its relevance needs explaining.

Referee #3 (Remarks to the Author):

The authors demonstrate that *Nitrospira inopinata*, a “comammox” bacterium that completely oxidizes ammonia to nitrate, can grow on guanidine as the sole source of nitrogen, energy, and reductant. Following up work of earlier researchers, these investigators identified a putative guanidinase-encoding gene in the majority of comammox strains, along with genes for guanidine transporters and guanidine-specific riboswitches. They expressed the *N. inopinata* gene in *E. coli*, defined the kinetics of the guanidine hydrolase, and structurally characterized the purified nickel-containing enzyme. Whole cell kinetic assays were carried out using *N. inopinata*, but those microrespirometric experiments are of minor interest because they are confounded by the need to incorporate rates for guanidine transport, guanidine hydrolysis, urea hydrolysis, and the multi-step ammonia oxidation pathway terminating in oxidase activity (with some unknown process inhibited by guanidine). Of slightly greater interest, they showed that mRNA encoding the enzyme was induced by guanidine in the microbial communities associated with wastewater treatment plants.

The reported experiments were carefully conducted, the interpretations appear to be solid, and the writing is very clear; however, the results presented are of only moderate interest. Complete ammonia oxidizers such as *N. inopinata* obtain nitrogen, energy, and reducing equivalents from ammonia, so any compound that can be degraded to ammonia will supply these items. Thus, it is no surprise that such microbes would acquire genes encoding guanidine hydrolase (and urease, appropriate transporters, and riboswitch regulatory sequences) as observed by DNA sequence analyses. The proteomics analysis of *N. inopinata* cells grown on guanidine versus ammonia showed, not unexpectedly, that the guanidinase protein is upregulated by the presence of its substrate. The crystal structure of the enzyme had previously been characterized for guanidine hydrolase isolated from a strain of *Synechocystis* (PDB: 7oi1), and few new insights were obtained for the nearly identical structure reported here (PDB: 8c0h). The enzyme described here possessed very low levels of metalcenter (only 0.56 Ni/subunit for what should be a dinuclear site) and low activity (e.g., 0.08 s^{-1} compared to that of the cyanobacterial enzyme = 3.83 s^{-1} at pH 8). The differences in properties might be attributed to the lack of genes for auxiliary components that function in metalcenter assembly for the *N. inopinata* gene when expressed in *E. coli*, compared to the cyanobacterial gene expression that included two nickel-delivery genes. The very high nickel concentrations used here appear to be insufficient for fully metalating the enzyme in *E. coli*. Perhaps the authors could transform their *E. coli* cells with a compatible plasmid containing DNA fragments from *N. inopinata* and screen for enhanced guanidine hydrolase activity (or reduced guanidine toxicity) to identify the associated genes.

Author Rebuttals to Initial Comments:

Referees' comments:

Referee #1 (Remarks to the Author):

Manuscript Growth of complete ammonia oxidizers on guanidine by Palatinszky et al. describes the growth of the complete ammonia oxidiser *Nitrosospira* on guanidine and the elucidation of its guanidinase enzyme system.

Guanidine is an important nitrogenous compound, but there is virtually no record of microorganisms growing on guanidine. The present study elegantly employed various complementary techniques such as proteomics, enzyme kinetics, and crystallography to demonstrate that comammox bacteria use guanidine as sole nitrogen and energy source, and that they contain a genuine guanidinase. Furthermore, the transcription of comammox guanidinase gene was observed to be induced in wastewater treatment plant microbiomes when incubated with micromolar quantities of guanidine. The discovery of guanidine as a selective growth substrate for comammox bacteria provides a very valuable insight into the unique niche of these important nitrifiers and presents novel opportunities for their isolation. Overall, this is an excellent contribution to the field of microbiology and environmental biotechnology.

We thank the reviewer very much for these very positive comments.

At some points in the manuscript, the authors exaggerate their findings and could tune down a bit their claim.

Taking also into account the comments of the reviewer below, we went through our manuscript and either provided additional data to support our statements or tuned them down as recommended.

Specific comments

Line 24 “it occurs widely in the environment” would need proper documentation on how much guanidine really occurs in ecosystems

For the revised manuscript we have now performed guanidine measurements in a sewage sample as well as in urine and feces of agricultural livestock (see new Extended Data Table 2). Furthermore, from published data it is well known that guanidine occurs in human urine, and during revision of our manuscript, a pre-print appeared that confirmed older literature that guanidine is also produced by plants (see citations in the revised manuscript).

Line 33-35 are overselling the scientific value by a large extend

We have removed the biodegradation statement as we show no data supporting this opportunity (although we believe it exists). While we are convinced that guanidine additions are worth to be tested for providing a selective advantage for the low-N₂O emitting comammox in agricultural soil, experimentally exploring this will take several years and we thus removed the respective statement from the abstract. However, we would like to keep the “new opportunities for comammox isolation” part as there are no indications for guanidine metabolism by AOA, and among the 5 AOB strains (harboring the complete genetic repertoire for guanidine utilization) we tested, only three strains could degrade a fraction of the added guanidine, but only in the presence of ammonia (see new Extended Data Figure 2).

Fig 1 do the depicted microbes have urease elsewhere in the genome? That is not so clear. In line 105 something is mentioned

Yes, as mentioned in the legend of Figure 1, these organisms encode urease in their genomes.

Line 75 “massive N loss” please quantify and tune down statement with proper reference

The review paper by Erisman *et al.* (2008) states that 40% of fertilizer nitrogen is lost to the environment (via nitrification & denitrification) ⁵.

Galloway *et al.*, 2003 report that 121 Tg N per year are lost as NO_x, NH₃, N₂O, and N₂, out of 170 Tg N applied, which works out to 71 % lost ⁶.

Billen *et al.*, 2013 state that of the globally 80 Tg N yr⁻¹ of synthetic fertilizer applied, 43 Tg N yr⁻¹ is lost to waterways, presumably as nitrate ⁷.

All references show massive nitrogen fertilizer loss, but the numbers vary and we would therefore prefer not to provide a number in the manuscript. However, we have added the Erisman *et al.* 2008 reference and have also slightly rephrased the respective sentence.

Line 81 “very narrow substrate range” is in sharp contrast to the titles of other studies by the consortium that highlight the versatility of this group of nitrifiers

As requested by the reviewer, we have rephrased this statement but would like to mention that two of the three publications listed below are on canonical nitrite-oxidizing members of the genus

Nitrospira and the third one (on uncultured comammox from WWTPs) is a meta-omics-study without physiological data.

Expanded metabolic versatility of ubiquitous nitrite-oxidizing bacteria from the genus *Nitrospira*. Koch *et al.*, 2015 PNAS

This publication describes new insights into the metabolism of the canonical nitrite-oxidizer *Nitrospira moscoviensis* - we showed that it can grow on urea in co-culture with ammonia-oxidizers and that it can grow aerobically on formate.

Activity and metabolic versatility of complete ammonia oxidizers in full-scale wastewater treatment systems Yang *et al.*, 2020 mBio

In this publication, genomes and *in situ* transcriptomes of comammox organisms from two wastewater treatment plants were analyzed. This provided hypotheses on the metabolic versatility of the comammox microbes (without recognizing the guanidinase which was annotated as an agmatinase) without experimental proof.

A nitrite-oxidising bacterium constitutively consumes atmospheric hydrogen

PM Leung *et al.*, 2022 ISME J

This publication describes that atmospheric hydrogen supports the growth of the canonical nitrite-oxidizer *Nitrospira moscoviensis*.

Line 81 the abbreviation AOM can represent many other processes, seems an odd choice to me
While AOM is sometimes used in the nitrification literature, we agree with the reviewer and have removed it from our manuscript.

Line 106 “much more energy efficient” you can not compare N source with an energy source in terms of efficiency, please rephrase

Please note that our sentence refers specifically to the conversion of guanidine to ammonia, which is indeed more energy efficient in comammox than in N-assimilating microbes. Therefore, we have not modified the statement. Aside from this, we agree with the reviewer that the energy demand for utilizing a substrate as an energy source, which is required for all cellular processes, likely has a different impact on the cell's overall energy budget than the energy demand for utilizing a substrate only as a nitrogen source.

Line 112 why was 50 micromolar chosen as the start concentration?

50 micromolar was selected, because we did not want to use a too high concentration of guanidine to avoid potential toxicity (and showed during revision of the manuscript that 50 micromolar is not strongly toxic for comammox - see also below. While NO_x production is somewhat slower when both ammonium and guanidine are provided compared to ammonium only, *N. inopinata* cells have the same growth yield when incubated on guanidine, ammonium or both substrates).

In addition, we did not want to use an extremely low concentration in order to minimize the time of the experimental procedures (still, the new growth experiments with comammox and 50 micromolar guanidine in the revised manuscript version took us 126 days - see new Figure 3).

Would the other nitrifiers have converted guanidine if you used a lower or higher concentration and if you would have waited longer? Now it seems everything was geared towards comammox. In the revised manuscript we now provide quantitative data of guanidine conversion (with or without added ammonium) by five AOB strains possessing all genes required for guanidine degradation via the ATP-dependent pathway. After two weeks of incubation with 50 micromolar guanidine (in triplicates) of high-cell density cultures (with equal protein biomass density as the comammox culture that showed rapid guanidine consumption), no AOB strain converted guanidine without addition of ammonium to the medium. In the presence of ammonium, three strains partially converted the added guanidine (see new Extended Data Figure 2).

These data show that active AOB with all genes required for guanidine degradation via the ATP-dependent pathway do not degrade 50 micromolar guanidine in the absence of ammonium. Only one of the five AOB strains was clearly inhibited by the 50 micromolar guanidine. We have initiated a new research project investigating in more detail the effect of lower concentrations of guanidine on AOB, with the current hypothesis that the guanidine degradation genes found in some AOB mainly serve for detoxification. In the framework of this new project, we will also study the influence of various guanidine concentrations (especially lower concentrations) on degradation by various AOB strains. We expect results on this within the next 1-2 years. In this context, we would also like to emphasize that the energy requirement of converting guanidine to ammonia is predicted to be significantly higher in AOB than in comammox organisms.

In response to the comment of the reviewer, we have modified the respective statement in our manuscript (p. 7, lines 129-131 in the version with changes indicated).

Line 134-136 does comammox occurrence and guanidine in ecosystems match? Or are ammonia and oxygen more strong drivers for niche separation?

This is a great question, but currently impossible to answer. We have started to measure guanidine in environmental samples (see new Extended Data Table 2), but many more measurements from many more environments are needed before such analyses can be made.

Line 174 ref 24 does not tell us much on the inhibition of guanidine on growth of nitrifiers.

For guanidine to be toxic it must have entered the cells; Is uptake of guanidine inhibited by the presence of urea or ammonia?

We thank the reviewer for this interesting comment. For the revised manuscript, we have repeated the growth experiments with *N. inopinata* including an incubation with ammonium and guanidine. The new data show that guanidine at a 50 micromolar concentration only slightly slows down ammonia oxidation by *N. inopinata* and does not lead to a lower cellular yield compared to incubations with ammonia alone. Furthermore, we demonstrate that guanidine and ammonia are

used in parallel by *N. inopinata* and that degradation rates of guanidine are not significantly lower in the presence of ammonia (see new Figure 3 and new Extended Data Figure 3).

Line 251 the affinity of 16 mM is in sharp contrast to the claimed low environmental concentrations? How do the cells take up the guanidine? Is there a large internal pool?

We agree with the reviewer that the low enzyme affinity, when it is expected to scavenge low environmental concentrations, is initially counter intuitive. Interestingly, the other two characterized cyanobacterial guanidinases also have poor affinities within the same range (5-8 mM guanidine). During revision, we obtained purified guanidinase from a high nickel preparation (1 mM Ni²⁺ in the *E. coli* growth medium) in an attempt to maximize metal loading into the enzyme. The updated pure enzyme kinetics overall showed a much faster maximal rate, but the affinity remained in the same range (13.6 mM, new Figure 5a). However, ICP-MS data on the guanidinase from the high nickel preparation still showed incomplete metal loading of the enzyme. Regarding the competitive success of comammox in the environment, the affinity of the guanidine transporter might be more important than the affinity of the guanidinase. Furthermore, a low substrate affinity of an enzyme does not preclude it from performing important metabolic reactions even if the enzyme might never operate at its maximal capacity. In this context, we also would like to emphasize that the now updated whole cell affinity of *N. inopinata* for guanidine of 1.34 micromolar guanidine (Figure 4b, Extended Data Figure 7a-b) is much higher (see also our response to the point of the reviewer below) and well in the range of measured environmental guanidine concentrations (Extended Data Table 2). This might reflect a more efficient metal-loading of the enzyme in vivo, post-translational modifications of the enzyme, molecular crowding in the cytoplasm, or accumulation of guanidine to higher intracellular concentrations via the amino acid/polyamine/organocation permease (APC superfamily).

Finally, *N. inopinata* has been cultured in our lab for more than a decade on ammonium. This might also have resulted in mutations in the riboswitch, the permease, or the guanidinase that affect its kinetic properties.

Line 269: 200 uM is still a very high K_s value for cells, if environmental guanidine concentrations are very low and conversion is not very efficient. Please provide K_s and V_{max} values of urea and ammonia by whole cells of comammox to put the values in perspective

During revision, we have optimized the whole cell affinity microrespirometry experiments (see comments to reviewer #2) and have now measured a much better apparent K_m value of 1.34 micromolar guanidine (new Figure 4b, Extended Data Figure 7a-b). In response to the comment of the reviewer, we have also determined the K_m and V_{max} values of urea and present them together with the previously determined values for ammonia in the modified Extended Data Figure 7. Interestingly, in our measurements *N. inopinata* has a comparable specific affinity for guanidine and urea, which is significantly lower than its specific affinity for ammonia, highlighting

the hierarchy of substrate acquisition and utilization by *N. inopinata*. This is discussed in the revised manuscript (p. 17-18, lines 343-359 in the version with changes indicated).

Line 298 elegant experiments with WWTP sludge indicate that comammox bacteria are geared to handle guanidine. Can transcripts of the comammox guanidinase gene be found in the transcripts of the native WWTP biomass? This would be an indication that this conversion is actually happening in situ. Maybe there are studies on micropollutants that have measured in and out flow concentrations of metformin and/or guanidine that could give you an indication of this in situ metabolism.

We thank the reviewer for this point. It should be noted that the activated sludge samples were kept without adding fresh wastewater for some time period before the start of the incubation experiments. Thus, guanidine originally present in the activated sludge sample might have been taken up/degraded during this period. We did not add fresh wastewater to the samples during incubation, as its composition changes so fast and conditions are hard to standardize.

Nevertheless, the WWTP-comammox guanidinase had a TPM>0 at T=0 in the incubation experiments, which either indicates that it was being transcribed in the WWTP or that the sludge produced guanidine, which is immediately consumed. We now mention this in the legend of Figure 5. Consistent with this observation, guanidine is known to be present in human urine and was now also measured by us in the influent of a sewage treatment plant (Extended Data Table 2).

The discussion contains quite some iteration of previous points and could be condensed. We have carefully checked the discussion and have condensed it.

Referee #2 (Remarks to the Author):

Palatinszky et al. report the use of guanidine as a source of energy, reductant and nitrogen by comammox microbes. Overall, the manuscript is well written and shows convincingly that comammox bacteria can produce nitrite and nitrate upon guanidine addition, strongly suggesting that after breakdown to ammonia it can be used as an energy source. This is a novel result and suggests a new role for guanidine beyond what was shown in Funck *et al.* (2022) and Wang *et al.* (2021), who showed guanidine N assimilation.

The evidence provided in the main manuscript that guanidine is used by the culture of *Nitrospira inopinata* as a nitrogen source and to gain energy for growth is less convincing (see below). Similarly, there is not sufficient support for the conclusion that guanidine utilization provides a niche for comammox bacteria in the environment. Overall, the manuscript is very long and could be more focused. Below, I made some suggestions how to reduce its length to better emphasize the main new finding – guanidine is an alternative energy source for comammox bacteria.

We thank the reviewer for the positive comments. We have performed during the last 8 months a number of additional experiments and convincingly show growth of *N. inopinata* on guanidine and provide many additional data regarding the importance of guanidine for comammox in the environment (see below).

1) Growth of *Nitrospira inopinata* on guanidine

The title of the manuscript is “Growth of complete ammonia oxidizers on guanidine” and the abstract states that “*Nitrospira inopinata* and likely most other comammox microbes grow on guanidine as the sole source of energy, reductant, and nitrogen”. However, the evidence provided in the main manuscript that these microorganisms grow on guanidine is weak. In figure 3, the authors show results from a batch incubation in which ~50 μM guanidine was added to a washed culture of *N. inopinata* that had been pre-incubated guanidine for a week. The guanidine is clearly converted into nitrite and nitrate, which clearly shows that it is used by these microorganisms as an electron donor. However, since there is no concomitant change in the 16S rRNA gene copies, there is no direct evidence that *Nitrospira inopinata* is growing in the first 10 days. In a second set of batch incubations presented in the extended data figure 4, there is an increase in gene copies; however, the data is at a very low temporal resolution. Since this is the main finding of the current manuscript, I would expect a more compelling growth curve (with more data points) based on an increase in cell numbers, OD600 or 16S rRNA copy numbers. The direct link between nitrate and nitrite production and growth is missing, and this is particularly essential since there is an imbalance in the nitrogen budget (as mentioned in the legend figure 3). Without more compelling evidence, it cannot be ruled out that the use of guanidine is instead a survival mechanism.

We thank the reviewer for this point. In response to it, we have designed a completely new growth experiment and performed this in high replication and high temporal resolution for 126 days. New Figure 3 as well and new Extended Data Figure 3 now clearly show that *N. inopinata* grows on guanidine as the only source of energy, electrons, and nitrogen.

2) Use of guanidine as a nitrogen source

Based on the imbalance in the nitrogen budget, the presented data does not provide evidence for direct use of guanidine as an N source for assimilation. This could be provided by using ^{15}N labeled guanidine followed by bulk isotope ratio mass spectrometry or nanoSIMS.

We thank the reviewer for this suggestion and have performed the requested NanoSIMS experiments that clearly show that *N. inopinata* uses guanidine as nitrogen (and carbon) source (see new Figure 3).

3) Purified enzyme kinetics and crystallography

Even though the crystallography data and the analyses of the crystal structure is very well done, it does not make a substantial advance on what was shown in the previous guanidine hydrolase papers (Funck et al. (2022) and Wang et al. (2021)). I would suggest shortening this section significantly and moving much of the data to a supplement as it is not relevant to the main novel message of the manuscript that these microorganisms gain energy from metabolizing guanidine. Following the suggestion of this reviewer, we have further condensed the crystal structure section in the main text, and we have also condensed two Extended Data Figures (6 and 7) into one, and have moved the rest of the respective figure material into Supplementary Figure 3. However, we would also like to take the opportunity to highlight two new aspects that were revealed by the crystal structure of the guanidinase of *N. inopinata*. Firstly, during revision of the manuscript we also used X-ray anomalous scattering for characterizing the metals in the heterologously expressed guanidinase and found nickel and manganese to form a binuclear active site, which sets it apart from the cyanobacterial enzyme. We now display this in the respective figures. However, we also mention that ultimatum proof would require purification and analyses of the enzyme from *N. inopinata*, which would take again many additional months of work. Secondly, the tilted Trp313 in the *N. inopinata* guanidinase facilitates access to the active site by allowing a passage from the surface, unlike the blocking Trp305 in the cyanobacterial enzyme. The enzyme's hexamer features negatively charged ridges and tunnels leading to the active sites, suggesting an electrostatic mechanism of attraction of the positively charged guanidinium cation that is the dominant form under physiological conditions.

It is surprising that the purified protein kinetics are worse than the whole cell kinetics. The authors state that this might be because the enzyme is not nickel loaded, which could well be the case. However in order to advance the previous work, it would have been interesting to attempt to solve this by using higher Ni concentrations or using Ni in the purification buffer at different concentrations.

As whole cell kinetics assesses the entire pathway from guanidine uptake and degradation through ammonia and nitrite oxidation to terminal oxidase activity in the actual cellular environment, this analysis reflects more accurately the *in situ* capability of this organism to convert guanidine. For example, a higher affinity of the guanidine permease can lead to elevated guanidine concentrations in the cytoplasm compared to the environment. Furthermore, a low affinity of an enzyme does not preclude it from performing important metabolic reactions, even if the enzyme might never operate at its maximal capacity. In this context, we also would like to emphasize that the now updated whole cell affinity of *N. inopinata* for guanidine of 1.34 micromolar guanidine (Figure 4b, Extended Data Figure 7a-b) is much higher than reported in the original manuscript and well in the range of measured environmental guanidine concentrations (Extended Data Table 2). This might reflect a more efficient metal-loading of the enzyme *in vivo*, post translational modifications of the enzyme, molecular crowding in the cytoplasm, or as mentioned above, accumulation of guanidine to higher intracellular concentrations via the amino acid/polyamine/organocation permease (APC superfamily). Finally, *N. inopinata* has been cultured in our lab for more than a

decade on ammonium. This might also have resulted in mutations in the riboswitch, the permease, or the guanidinase that affect its kinetic properties.

In response to the comment of the reviewer, we have performed additional heterologous expression of the enzyme at elevated (20 μ M, 1 mM) Ni^{2+} concentration in the *E. coli* growth medium. Kinetic analyses of these newly produced enzymes showed a much faster maximal rate, but the affinity remained in the same range (new Figure 4a and Extended Data Table 1). However, even at strongly elevated Ni^{2+} concentrations in the *E. coli* growth medium, the heterologously expressed enzyme was not fully metal-loaded (Extended Data Table 1). Biosynthesis and insertion of nickel active sites often require specific and elaborated maturation pathways, and we hypothesize that the concomitant heterologous expression of other (unknown) *N. inopinata* proteins would be required to increase the metal loading of the guanidinase.

In addition, considering that the work is carried out using a pure culture, the authors could directly purify the enzyme from the native cells and characterize the protein afterwards. If this is not feasible, at least the enzyme kinetics measurements could be carried out on cell-free extracts. This would be novel and add more to our understanding of this enzymatic pathway.

We discussed this point extensively in our team but then decided to not perform these experiments as we would need to grow a considerable amount of *N. inopinata* biomass with guanidine (only then the guanidinase is induced). As growth is slower on this substrate than on ammonium (see new Figure 3), this would have been (in addition to all the other experiments we have performed in response to the reviewer comments) a considerable additional time investment.

The purified protein is a part of the arginase family. In the paper by Funck et al., activity of the guanidinase was also tested on alternative substrates such as arginine, creatine and other guanidine-containing compounds. I would suggest the authors to determine whether the enzyme they purified is a specific guanidinase as reported for the one in the Funck et al. paper or whether it has a wider range of substrate spectrum. The rather low affinity of the purified enzyme might as well point towards the possibility that the guanidine hydrolysis is rather the side reaction of this enzyme rather than its bona fide activity.

We thank the reviewer for this valuable comment and now provide new experimental data on the *N. inopinata* guanidinase activity with six alternative substrates, demonstrating absent or negligible urea formation (p.16. Lines 207-315 in the version with changes indicated; Extended Data Figure 5e). This is also consistent with the observation of a very narrow substrate channel in the enzyme structure (Figure 2, Extended Data Figure 6; p.16 Lines 315-319) that would likely exclude efficient access of larger substrates to the active site of the enzyme. The calculated molecular volumes for the potential alternative substrates are between 127% and 287% larger than that of guanidine, which barely fits the entry channel. Furthermore, the guanidine-I riboswitch recognizes every functional group of the ligand through hydrogen bonding to guanine bases and phosphate oxygens. Guanidinium binding is further stabilized through cation- π interactions with

guanine bases. This allows the riboswitch to recognize guanidinium while excluding other bacterial metabolites with a guanidino group, including the amino acid arginine⁸.

Together with the high whole cell affinity of *N. inopinata*, our new data and the above-mentioned considerations show that guanidine hydrolysis is most likely the *bona fide* activity of this enzyme.

Furthermore, the manuscript states that the enzyme has a high thermostability, but activity is not measured over a range of temperatures or pH. Instead, this conclusion is based on a protein unfolding model, however, the enzyme might stop being active long before it “unfolds”. Therefore, further activity assays are needed to support the conclusion.

We thank the reviewer for this comment. We have added experimental activity data of the enzyme at different pH values and temperatures (see modified Extended Data Figure 5), which confirm the theoretical predictions.

4) Whole cell kinetics

I am wondering why guanidine, ammonium and urea were not directly measured in the whole cell kinetics assays. I think these experiments should be repeated with these measurements, which would make them more compelling. Particularly considering that these compounds were used to determine the kinetics for the purified enzyme. I understand that this data would not be at the same resolution as the O₂ respirometry, but it would be important to show what is actually occurring as currently the range in which guanidine appears to be utilized is rather narrow and very high.

The reviewer raises a good point, and during revision we discovered that in fact we had intermediates building up and not being fully converted to nitrate in our microrespirometry experiments. This made interpreting the whole cell kinetics challenging. While we had measured all relevant compounds at the start and at the end of the microrespirometry experiments, no subsamples had been taken during these experiments for chemical measurements, because the microrespiration is performed in a small (~2 ml), headspace free glass vial.

During the revision, we have updated our whole cell kinetics results with single trace measurements, which measure the decrease in O₂ concentration until all substrate (including all intermediates) is consumed. We confirmed the absence of the intermediates (ammonium, nitrite, urea) at the start and end of these experiments. This approach provides information on the whole process from initial guanidine transport into the cell all the way to electron transfer at the terminal oxidase (see new Figure 4b).

5) Wastewater treatment sludge experiments

A large section of the manuscript is devoted to experiments carried out using wastewater treatment sludge which the authors added guanidine to and subsequently examined transcription of

guanidinase and APCP. However, it is unclear what the aim of these experiments were. If guanidine utilization is an important factor in the success of comammox bacteria in WWTP, then why are the relevant enzymes not already being transcribed? This might suggest that guanidine is not a substrate the comammox microorganisms encounter in WWTP. These experiments suffer from the lack of in situ guanidine measurements, which could easily be measured using the newly developed assay.

Published data show that (i) guanidine is present in human urine and thus also in sewage, and (ii) that functionally redundant microbes can coexist by partitioning low-concentration substrates even though they compete for one dominant substrate. We discuss these points now more extensively in the revised manuscript. It is a major question in wastewater treatment microbiology which factors enable functional redundancy and thus preserve microbiome services in these systems. In wastewater treatment plants the co-existence of various ammonia-oxidizers is often observed, but little is known which traits help the respective guild members to avoid competitive exclusion. Therefore, we wanted to experimentally test whether guanidine addition triggers transcription of wastewater treatment plant comammox strains and whether guanidine is degraded in plants with comammox microbes. This cannot be taken for granted as (i) *in silico* predictions of metabolic traits can always be incorrect, and (ii) there might be competition for guanidine from heterotrophic microbes which use this compound for assimilation. Our results suggest that guanidine as a minor substrate is used by comammox in these engineered systems and thus likely supports co-existence of comammox with other ammonia-oxidizers not capable of growing on this compound.

It should also be noted that the activated sludge samples were transported for several hours from the treatment plants to the lab and were subsequently stored there (partly overnight). During this time, a large fraction of guanidine present in the sludge would have been degraded. Nevertheless, we still observed transcription of guanidine utilization genes in the sludge at T0, suggesting the presence of guanidine in the original activated sludge or production of guanidine by sludge microbiome members. We mention this important point in the revised manuscript.

While we had no access during revision of the manuscript to the sewage of the two Danish treatment plants, we now sampled again sewage from the Austrian control wastewater treatment plant that does not contain comammox microbes and measured 0.5 $\mu\text{mol/L}$ guanidine in its influent. We provide this data in the revised manuscript in the new Extended Data Table 2 together with new guanidine measurements of urine and fecal samples from farm animals.

If these experiments were carried out with the intention of supporting the statement that guanidine can be used to enrich comammox bacteria, then I would expect to see a comammox enrichment on guanidine. The use of guanidine as a potential substrate to enrich comammox is a key point in the discussion and abstract, but so far no evidence is shown to support this statement.

Enriching and isolating nitrifiers is a very time-consuming effort and we have started such experiments and expect publishable results in a few months to years. The wastewater treatment

experiments in the manuscript were not specifically intended to support the enrichment statement (although they are consistent with the hypothesis that this might be a promising strategy) as they were lasting for maximally 2 days, which is much too short to expect any enrichment of comammox microbes.

6) Guanidine utilization as a niche for comammox bacteria

The abstract states that “*Nitrospira inopinata* and likely most other comammox microbes grow on guanidine as the sole source of energy, reductant, and nitrogen” and that “The discovery of guanidine as selective growth substrate for comammox shows a unique niche of these globally important nitrifiers”

In the first case, the text should be adjusted to say that they can use guanidine as a source of energy (and if shown more convincingly, nitrogen). The current text implies that they grow solely off guanidine in the environment, which is not supported by the data provided in the manuscript.

Our newly performed experiments (see the new Figure 3 and Extended Figure 3) unambiguously demonstrate that *N. inopinata* grows on guanidine and also uses it as a nitrogen source.

Similarly with the second sentence, I do not see that guanidine utilization is a niche for comammox in the environment. *N. inopinata* has an apparent substrate affinity ($K_m(\text{app})$) for guanidine of $\sim 202 \mu\text{M}$ – in comparison its affinity for ammonium is $0.84 \mu\text{M}$. Do the authors expect that guanidine would therefore be used in environments where ammonium is already present and that the presence of this compound therefore provides a niche for these bacteria? In order to keep this statement, the authors must provide more evidence that there is a niche in the environment where guanidine might be important – which likely would require more measurements of its concentration in situ.

(i) With our methodologically improved whole cell affinity measurements we now demonstrate a much higher affinity of *N. inopinata* ($K_m(\text{app})$, $1.34 \pm 0.25 \mu\text{M}$) for guanidine (see the new Figure 4b).

(ii) We have added a new set of experiments, which demonstrate that guanidine added at concentrations typically used for N-fertilization to comammox-containing agricultural soil is nitrified in these soils. If nitrification is inhibited by acetylene addition, guanidine degradation in the soil was found to be much slower. Thus, in the revised manuscript we provide experimental support for the use of guanidine by comammox-containing nitrifying microbial communities in wastewater treatment systems and agricultural soil.

(iii) Guanidine is present in human urine and sewage. We have also now detected guanidine and quantified its concentrations in urine and fecal samples of various animals, which play an important role in manure production for agriculture (see the new Extended Data Table 2). Therefore, additional sources of guanidine for comammox strains (and guanidine-assimilating microbes) thriving in wastewater treatment plants and agricultural soil have been identified. In regard to agricultural soil, it should also be noted that plants produce guanidine (see references in our manuscript).

(iv) Recent literature has nicely demonstrated that functionally redundant microbes can coexist by partitioning low-concentration substrates even though they compete for one dominant substrate^{3,4}. Thus, the capability of using guanidine (low-concentration substrate) for growth is providing an additional niche for comammox organisms in guanidine-containing environments and is likely important for their co-existence with other ammonia-oxidizing microbes, with which they compete for the dominant substrate ammonia. Co-existence instead of competitive exclusion enables the functional redundancy of ammonia-oxidizers as frequently observed in soils and wastewater treatment plants, preserving their overall ecosystem service. In the revised manuscript, we have added this important aspect in the discussion section.

Minor comments:

Please do not use the term AOM, it is also used to refer to anaerobic oxidation of methane and therefore can lead to confusion. “Ammonia oxidizers” could be used instead of AOM.

Done as suggested.

In the WWTP section the title of the figure “Metatranscriptomic response to guanidine amendment by comammox activated sludge microbiome members” is confusing and should be adjusted.

We have rephrased the figure title.

What is meant by “ancient trait for comammox”, ancient implies you have an idea as to the timing. Rephrase.

Almost all environmental and cultured comammox have the genes for guanidine utilization, and the guanidinase and *amoA* phylogenies are largely congruent. The most parsimonious explanation for these findings is that the last common ancestor of the known comammox strains already possessed this trait. Therefore, we think that this is an accurate statement although we cannot time when the LCA of comammox lived.

Reformulate sentence in abstract:

“It occurs widely in nature and is used by microbes as a nitrogen source, but microorganisms growing on guanidine have not yet been discovered”

For the general readership of Nature, it is probably unclear what the difference is between guanidine being used as a nitrogen source and microbes growing on guanidine. This sentence could be rephrased as: “It microorganisms using guanidine as an energy source”

We have rephrased this sentence to improve clarity.

The meaning of “Riboswitch” and its relevance needs explaining.

Done as suggested.

Referee #3 (Remarks to the Author):

The authors demonstrate that *Nitrospira inopinata*, a “comammox” bacterium that completely oxidizes ammonia to nitrate, can grow on guanidine as the sole source of nitrogen, energy, and reductant. Following up work of earlier researchers, these investigators identified a putative guanidinase-encoding gene in the majority of comammox strains, along with genes for guanidine transporters and guanidine-specific riboswitches. They expressed the *N. inopinata* gene in *E. coli*, defined the kinetics of the guanidine hydrolase, and structurally characterized the purified nickel-containing enzyme. Whole cell kinetic assays were carried out using *N. inopinata*, but those microrespirometric experiments are of minor interest because they are confounded by the need to incorporate rates for guanidine transport, guanidine hydrolysis, urea hydrolysis, and the multi-step ammonia oxidation pathway terminating in oxidase activity (with some unknown process inhibited by guanidine).

We agree with the reviewer that from a biochemistry/enzymatic perspective, whole cell kinetic assays are less informative. However, from a microbial ecology perspective, whole cell assays are much more informative than assays with a purified enzyme. The structural and functional analyses of the heterologously expressed guanidinase from *N. inopinata* mainly served the purpose to prove that it actually represents a *bona fide* guanidinase (a conclusion that cannot be made based on *in silico* analyses only). The whole cell assays, which we further improved significantly during manuscript revision, then allow us to experimentally infer how well comammox competes for a dilute substrate in its environment. This is essential, as recent literature has nicely demonstrated that functionally redundant microbes can coexist by partitioning low-concentration substrates even though they compete for one dominant substrate^{3,4}. Thus, the capability of using guanidine (low-concentration substrate) for growth is providing an additional niche for comammox organisms in guanidine-containing environments and is likely important for their co-existence with other ammonia-oxidizing microbes, with which they compete for the dominant substrate ammonia. Co-existence instead of competitive exclusion enables the functional redundancy of ammonia-oxidizers that is frequently observed in soils and wastewater treatment plants, preserving their overall ecosystem service. In the revised manuscript, we have added this important aspect in the discussion section.

Of slightly greater interest, they showed that mRNA encoding the enzyme was induced by guanidine in the microbial communities associated with wastewater treatment plants.

Nitrification is of major importance in wastewater treatment plants and as human urine contains guanidine, sewage also contains guanidine (now also measured by us, see new Extended Data Table 2). Nitrification in wastewater treatment plants is sensitive to perturbations (with major ecological consequences), and thus it is desired to maintain a high functional redundancy of nitrifiers in these systems to preserve nitrification robustness. Thus, our finding that comammox can grow on guanidine and might be able to co-exist with other ammonia-oxidizers in wastewater treatment plants should be of broad interest.

Furthermore, we also now demonstrate that guanidine is nitrified in agricultural soils (new Figure 6) and show that major farm animals excrete guanidine in urine and feces. This is an important finding, as comammox emits less nitrous oxide than ammonia-oxidizing bacteria. To reduce the emission of this potent greenhouse gas and ozone-depleting substance from agricultural soils, an in-depth understanding of factors shaping the community composition and activity of nitrifier communities in these systems is essential.

The reported experiments were carefully conducted, the interpretations appear to be solid, and the writing is very clear; however, the results presented are of only moderate interest. Complete ammonia oxidizers such as *N. inopinata* obtain nitrogen, energy, and reducing equivalents from ammonia, so any compound that can be degraded to ammonia will supply these items. Thus, it is no surprise that such microbes would acquire genes encoding guanidine hydrolase (and urease, appropriate transporters, and riboswitch regulatory sequences) as observed by DNA sequence analyses. The proteomics analysis of *N. inopinata* cells grown on guanidine versus ammonia showed, not unexpectedly, that the guanidinase protein is upregulated by the presence of its substrate.

We respectfully disagree with the reviewer on this point. Ammonia can be enzymatically produced from many substrates, but nevertheless despite decades of intensive research in the field, only urea and cyanate have been proven to be used by nitrifiers as ammonia sources (other than ammonia itself) for growth. In contrast to urea and cyanate, guanidine seems to be used by virtually all members of one of the three major groups of ammonia-oxidizers (AOB, AOA, comammox) and among them, the energy-efficient pathway for its decomposition is exclusively found in comammox organisms. This is particularly important as guanidine is a widespread ammonia-source, as indicated by our guanidine measurements in several samples and the widespread occurrence of guanidine riboswitches in bacterial genomes. As discussed in the manuscript, the unexpected finding that comammox organisms thrive on guanidine opens many new opportunities for nitrifier research ranging from the targeted isolation of the fastidious comammox organisms to the manipulation of nitrifier community composition.

The crystal structure of the enzyme had previously been characterized for guanidine hydrolase isolated from a strain of *Synechocystis* (PDB: 7oi1), and few new insights were obtained for the nearly identical structure reported here (PDB: 8c0h). The enzyme described here possessed very low levels of metalcenter (only 0.56 Ni/subunit for what should be a dinuclear site) and low activity (e.g., 0.08 s⁻¹ compared to that of the cyanobacterial enzyme = 3.83 s⁻¹ at pH 8).

We agree that there are major similarities between both enzymes and we clearly mention this in our manuscript. We have further condensed the crystal structure section in the main text, and we also condensed two Extended Data Figures (6 and 7) into one, and moved the rest of the respective figure material into Supplementary Figure 3. However, we would also like to take the opportunity to highlight two new aspects that were revealed by the crystal structure of the guanidinase of *N. inopinata*. Firstly, during revision of the manuscript, we also used X-ray anomalous scattering for characterizing the metals in the heterologously expressed guanidinase and found nickel and manganese to form a binuclear active site, which sets it apart from the cyanobacterial enzyme. We now display this in the respective figures. However, we also mention that ultimate proof of the *in situ* metal loading of the active site would require purification and analyses of the enzyme from *N. inopinata* (as we observed different metal loadings dependent of the conditions of the heterologous expression, see new Extended Data Table 1), which would take again many additional months of work. Secondly, the tilted Trp313 in the *N. inopinata* guanidinase facilitates access to the active site by allowing a passage from the surface, unlike the blocking Trp305 in the cyanobacterial enzyme. The enzyme's hexamer features negatively charged ridges and tunnels leading to the active sites, suggesting an electrostatic mechanism of attraction of the positively charged guanidinium cation that is the dominant form under physiological conditions.

As a side note, the comammox guanidinase expressed heterologously under high nickel conditions has a substantially higher K_{cat} of 1.42 s⁻¹.

The differences in properties might be attributed to the lack of genes for auxiliary components that function in metalcenter assembly for the *N. inopinata* gene when expressed in *E. coli*, compared to the cyanobacterial gene expression that included two nickel-delivery genes. The very high nickel concentrations used here appear to be insufficient for fully metalating the enzyme in *E. coli*. Perhaps the authors could transform their *E. coli* cells with a compatible plasmid containing DNA fragments from *N. inopinata* and screen for enhanced guanidine hydrolase activity (or reduced guanidine toxicity) to identify the associated genes.

This is an interesting suggestion and we indeed plan to experimentally screen in various ways for nickel-delivery enzymes in *N. inopinata*. However, as there are no obvious candidate genes, we feel that this will be a very large project on its own that is well beyond the scope of this manuscript. In this context, we would also like to highlight that we have already spent more than 6 months of additional experimental work by involving many of the previous as well as four new co-authors to significantly expand our study based on the comments of the reviewers.

We would like to thank all three reviewers for their constructive criticism, which has helped us a lot to put together a revised manuscript that we are convinced is even more complete and interesting.

Bibliography

1. Wang, J., Wang, J., Rhodes, G., He, J.-Z. & Ge, Y. Adaptive responses of comammox *Nitrospira* and canonical ammonia oxidizers to long-term fertilizations: Implications for the relative contributions of different ammonia oxidizers to soil nitrogen cycling. *Sci. Total Environ.* **668**, 224–233 (2019).
2. Zhang, K. *et al.* Comammox plays a functionally important role in the nitrification of rice paddy soil with different nitrogen fertilization levels. *Appl. Soil Ecol* **193**, 105120 (2024).
3. Yu, X. A. *et al.* Low-level resource partitioning supports coexistence among functionally redundant bacteria during successional dynamics. *ISME J.* **18**, (2024).
4. Brochet, S. *et al.* Niche partitioning facilitates coexistence of closely related honey bee gut bacteria. *eLife* **10**, (2021).
5. Erisman, J. W., Sutton, M. A., Galloway, J., Klimont, Z. & Winiwarter, W. How a century of ammonia synthesis changed the world. *Nature Geosci.* **1**, 636–639 (2008).
6. Galloway, J. N. *et al.* The Nitrogen Cascade. *Bioscience* **53**, 341 (2003).
7. Billen, G., Garnier, J. & Lassaletta, L. The nitrogen cascade from agricultural soils to the sea: modelling nitrogen transfers at regional watershed and global scales. *Philos. Trans. R. Soc. Lond. B Biol. Sci.* **368**, 20130123 (2013).
8. Reiss, C. W., Xiong, Y. & Strobel, S. A. Structural Basis for Ligand Binding to the Guanidine-I Riboswitch. *Structure* **25**, 195–202 (2017).

Reviewer Reports on the First Revision:

Referees' comments:

Referee #1 (Remarks to the Author):

The manuscript "Growth of Complete Ammonia Oxidizers on Guanidine" by Palatinszky et al. showcased the remarkable new finding of the ability of comammox Nitrospira to thrive on guanidine, and elucidated the structure and properties of its guanidinase enzyme system.

The authors have gone to great length and depth to conduct additional experiments to address the concerns raised by the reviewers. The revised version shows much better the occurrence and relevance of guanidine in the environment and possible use by comammox bacteria. The discrepancy between kinetic properties and environmental low concentrations have been investigated and addressed by improved protocols and measurements.

we look forward to the mentioned new studies and hope that they will not take as long as predicted. Good luck!

Referee #2 (Remarks to the Author):

The revised manuscript "Growth of complete ammonia oxidisers on guanidine" by Palatinszky et al contains substantial amounts of new data. The extensive work carried out by the authors during the revision process now shows convincingly that comammox bacteria can grow on guanidine as a sole nitrogen-source. The updated whole-cell kinetics results are much more compelling and the new experimental data indicating that the guanidase is guanidine specific is an important addition. The authors have also added a new experiment examining guanidine use in soils.

However, one of my previous major concerns still remains. The claim that "guanidine use provides a unique niche for comammox in the environment" is to my opinion not supported by the presented data, as outlined below:

First, the evidence for a "unique niche" for comammox bacteria is not provided. The presented data show that other ammonia-oxidizing bacteria (AOB) can also use guanidine, so the word 'unique' is not appropriate.

It might be that comammox bacteria outcompete AOB for guanidine, but these data are currently not provided. Although the AOB cultures did not use as much guanidine as the comammox cultures, individual guanidine consumption and growth rates would be required for the AOB cultures, which could then be compared to the comammox results. Alternatively, the authors could perform a competition experiment between *N. inopinata* and a guanidinase-encoding AOB. Without this additional data, there is not sufficient experimental support for the conclusion that guanidine can be used to select for comammox in agricultural fields and thus reduce N₂O emissions. As it is, the data shows for the first

time that several ammonia-oxidizing bacteria, including comammox, can oxidize and grow on guanidine, which is an exciting result in itself.

Secondly, the relevance of this metabolism in the environment i.e. waste-water treatment plant (wwtp) and agricultural soils is over-stated. The authors present new data showing that guanidine is present in animal urine and feces, as well as at the Klosterneuburg wwtp influent. At Klosterneuburg wwtp, the measured influent sample has a guanidine concentration of 500 nM, whereas ammonium is typically present in domestic wastewater at a concentration of around 2 mM. Based on the comparable yields of comammox on guanidine and ammonium, if all substrates would be used by comammox bacteria, the amount of biomass that could be obtained from guanidine would be 4000 times less than the biomass obtained from ammonium. Taken together with the fact that comammox bacteria also have much lower guanidine specific activity than ammonium specific activity, this suggests that guanidine is likely of minor importance for comammox in wwtp.

To investigate guanidine metabolism in soils, new experimental data has been added. The data shows that the addition of acetylene slows down guanidine degradation and inhibits nitrate production, which indicates an involvement of ammonia oxidizing microbes. However, there is no direct evidence for the involvement of comammox in guanidine degradation in the soil experiment; in fact, no growth of comammox by qPCR was detected in these incubations. While this experiment strongly supports an involvement of ammonia-oxidizers in guanidine degradation in soil in general, no direct link to comammox bacteria can be made. As such, while I appreciate the time and resources the authors have invested to this experiment, its relevance for the current main conclusion of the manuscript (i.e. guanidine use provides a unique niche for comammox in the environment) is unclear to me. Instead, these results do indicate that ammonia-oxidizers, including comammox, can oxidize guanidine in soils, which is interesting and could be emphasized in the manuscript.

In summary, unless additional data is provided as suggested above, the conclusions concerning the unique niche for comammox in the environment should be toned down accordingly, as should the subsequent speculation regarding use of guanidine to select for comammox to reduce N₂O emissions from agricultural fields.

Other comments:

The data in the manuscript all seem to point to a slow growth of *N. inopinata* on guanidine. The authors argue that this is due to deleterious mutations in the guanidinase gene/transporter or riboswitch due to extended cultivation; but if that is the case, then the same argumentation needs to be considered for the other results, for example the lack of guanidinase activity on other substrates. Furthermore, the other investigated cultures of ammonia oxidizing bacteria might also suffer from similar deleterious mutations, making them less efficient at using guanidine than environmental AOB. These considerations should be added in the respective discussion.

The new data do not show how quantitatively important guanidine is for growth of *N. inopinata* in comparison to ammonia. Growth rates for both substrates should be added.

The detection limit for guanidine concentrations in the soils is not clear. All soil additions are given in units of $\mu\text{g-N g}^{-1}$ dry weight soil, whereas the detection limit is given as $<50 \text{ nM}$. Please make these units in the legend of figure 6 consistent.

New Ext data fig. 3. Based on the rest of the paper, panel g on this figure is rather surprising. It seems to show that the addition of guanidine in fact reduces the total oxidation rate. If this interpretation is correct, it needs to be addressed in the manuscript.

New data has been added showing that guanidine is present in animal urine and feces, as well as at the Klosterneuburg WWTP influent; but at extremely low concentrations, and this should be acknowledged or at least compared to ammonium concentrations in the respective samples/environments.

Abstract L34: “*N. inopinata* and likely most other comammox microbes grow on guanidine as the sole source of energy, reductant, and nitrogen”. As it is, the sentence implies that this is their only metabolism, please adjust to clarify that they can also use ammonium. Change as follows; “*N. inopinata* and likely most other comammox microbes can grow on guanidine as the sole source of energy, reductant, and nitrogen”.

Referee #3 (Remarks to the Author):

The revisions to the manuscript adequately addressed my prior concerns. The authors have better emphasized the significance of their work, shortened appropriately some sections, and enhanced the quality of several studies. The demonstration that the two metal ions per subunit in crystallized guanidinase include one Ni and one Mn is fascinating, although this enzyme was obtained from recombinant *E. coli* and so may not be identical to the situation for enzyme isolated from the original microorganism.

Author Rebuttals to First Revision:

Referee #1 (Remarks to the Author):

The manuscript "Growth of Complete Ammonia Oxidizers on Guanidine" by Palatinszky et al. showcased the remarkable new finding of the ability of comammox *Nitrospira* to thrive on guanidine, and elucidated the structure and properties of its guanidinase enzyme system.

The authors have gone to great length and depth to conduct additional experiments to address the concerns raised by the reviewers. The revised version shows much better the occurrence and relevance of guanidine in the environment and possible use by comammox bacteria. The discrepancy between kinetic properties and environmental low concentrations have been investigated and addressed by improved protocols and measurements.

we look forward to the mentioned new studies and hope that they will not take as long as predicted. Good luck!

Many thanks for these very positive comments and for acknowledging the many additional experiments performed during revision.

Referee #2 (Remarks to the Author):

The revised manuscript "Growth of complete ammonia oxidisers on guanidine" by Palatinszky et al contains substantial amounts of new data. The extensive work carried out by the authors during the revision process now shows convincingly that comammox bacteria can grow on guanidine as a sole nitrogen-source. The updated whole-cell kinetics results are much more compelling and the new experimental data indicating that the guanidase is guanidine specific is an important addition. The authors have also added a new experiment examining guanidine use in soils.

However, one of my previous major concerns still remains. The claim that "guanidine use provides a unique niche for comammox in the environment" is to my opinion not supported by the presented data, as outlined below:

First, the evidence for a "unique niche" for comammox bacteria is not provided. The presented data show that other ammonia-oxidizing bacteria (AOB) can also use guanidine, so the word 'unique' is not appropriate.

It is correct that in our experiments three betaproteobacterial AOB strains converted guanidine, but only in the presence of ammonium. Without ammonium addition, guanidine was not degraded by the AOB (Extended Data Figure 2). Furthermore, we would like to stress here that the degradation pathway of guanidine encoded by AOB requires ATP, whereas the degradation pathway used by comammox organisms is predicted to be more energy efficient. Consistently, in Extended Data Figure 2, a much faster guanidine degradation was observed for *N. inopinata*

than for the AOB strains, although these experiments were performed at the optimum growth temperature of the AOB strains and 9 degrees below the temperature optimum for *N. inopinata*. Keeping these points in mind, we referred to “a unique niche” for comammox organisms in our manuscript, but we agree with the reviewer that this can lead to misinterpretations. Therefore, we have modified the respective sections in the manuscript accordingly.

It might be that comammox bacteria outcompete AOB for guanidine, but these data are currently not provided. Although the AOB cultures did not use as much guanidine as the comammox cultures, individual guanidine consumption and growth rates would be required for the AOB cultures, which could then be compared to the comammox results.

For feasibility reasons, the replicated AOB experiments in the Extended Data Figure 2 were done in microtiter plates and, due to the limited available volume per incubation, only end-point measurements were performed. Therefore, rate calculations were not possible. However, growth or degradation rate comparisons between comammox and AOB would be very challenging to interpret as the tested AOB can degrade guanidine only in the presence of ammonium.

Alternatively, the authors could perform a competition experiment between *N. inopinata* and a guanidinase-encoding AOB. Without this additional data, there is not sufficient experimental support for the conclusion that guanidine can be used to select for comammox in agricultural fields and thus reduce N₂O emissions. As it is, the data shows for the first time that several ammonia-oxidizing bacteria, including comammox, can oxidize and grow on guanidine, which is an exciting result in itself.

We agree with the reviewer that the suggested competition experiments would be interesting. In fact, we have been trying without success for many months to set up stable co-cultures of *N. inopinata* and different AOB strains for subsequent competition experiments. One major challenge with these co-cultures is that *N. inopinata* is a moderate thermophile, while most AOB are mesophiles. Although the predicted energy requirements for guanidine degradation by comammox vs. AOB suggest a selective advantage for the comammox organisms, and efficient growth of AOB on guanidine appears unlikely based on our pure culture experiments (where AOB only co-oxidized guanidine with ammonium), we agree with the reviewer that we have not experimentally demonstrated this directly. Thus, we toned down and significantly shortened the section in the discussion on the use of guanidine to select for comammox in agricultural fields to reduce N₂O emissions.

Secondly, the relevance of this metabolism in the environment i.e. waste-water treatment plant (wwtp) and agricultural soils is over-stated. The authors present new data showing that guanidine is present in animal urine and feces, as well as at the Klosterneuburg wwtp influent. At Klosterneuburg wwtp, the measured influent sample has a guanidine concentration of 500 nM, whereas ammonium is typically present in domestic wastewater at a concentration of around 2 mM. Based on the comparable yields of comammox on guanidine and ammonium, if all substrates would be used by comammox bacteria, the amount of biomass that could be

obtained from guanidine would be 4000 times less than the biomass obtained from ammonium. Taken together with the fact that comammox bacteria also have much lower guanidine specific activity than ammonium specific activity, this suggests that guanidine is likely of minor importance for comammox in wwtp.

The wastewater treatment experiments were performed to demonstrate guanidine degradation by comammox microbes under competitive conditions (e.g., in the presence of other guanidine-assimilating microbes) in complex microbial communities. Indeed, our combined chemical and metatranscriptomic data clearly demonstrated that guanidine is detected by the riboswitches of the comammox strains in nitrifying activated sludges and that concomitantly to guanidine degradation, transcription of the comammox guanidinase is upregulated.

Also in the previous version of the manuscript, we have not interpreted the wastewater treatment plant experiments regarding the relevance of the comammox guanidine catabolism for the biotechnological process. Indeed, much additional work would be needed to obtain a comprehensive picture e.g. on the guanidine concentrations in raw sewage in different treatment plants and countries (agricultural impact might be very relevant) and the guanidine contribution to total nitrogen concentrations in the effluent of wastewater treatment plants. As a minor comment, we would also like to mention that the degradation of one guanidine molecule releases three ammonium molecules.

To investigate guanidine metabolism in soils, new experimental data has been added. The data shows that the addition of acetylene slows down guanidine degradation and inhibits nitrate production, which indicates an involvement of ammonia oxidizing microbes. However, there is no direct evidence for the involvement of comammox in guanidine degradation in the soil experiment; in fact, no growth of comammox by qPCR was detected in these incubations. While this experiment strongly supports an involvement of ammonia-oxidizers in guanidine degradation in soil in general, no direct link to comammox bacteria can be made. As such, while I appreciate the time and resources the authors have invested to this experiment, its relevance for the current main conclusion of the manuscript (i.e. guanidine use provides a unique niche for comammox in the environment) is unclear to me. Instead, these results do indicate that ammonia-oxidizers, including comammox, can oxidize guanidine in soils, which is interesting and could be emphasized in the manuscript.

We agree with the reviewer that our soil experiments do not prove that the guanidine degradation was performed by the comammox microbes in these soils (although growth of comammox was not an expected readout as even in the ammonium treatment no significant changes of any ammonia oxidizer abundances were detected) and thus we did not postulate this in our manuscript. In the revised version, we now specifically emphasize that our data do not directly prove guanidine degradation by comammox strains in the soil, but that they highlight the large contribution of ammonia-oxidizing microbes to the observed guanidine degradation.

In summary, unless additional data is provided as suggested above, the conclusions concerning the unique niche for comammox in the environment should be toned down accordingly, as

should the subsequent speculation regarding use of guanidine to select for comammox to reduce N₂O emissions from agricultural fields.

As requested, we have toned down the conclusions concerning the unique niche for comammox in the environment provided by guanidine as well as the speculation on the use of guanidine to select for comammox in agricultural fields to reduce N₂O emissions.

Other comments:

The data in the manuscript all seem to point to a slow growth of *N. inopinata* on guanidine. The authors argue that this is due to deleterious mutations in the guanidinase gene/transporter or riboswitch due to extended cultivation; but if that is the case, then the same argumentation needs to be considered for the other results, for example the lack of guanidinase activity on other substrates. Furthermore, the other investigated cultures of ammonia oxidizing bacteria might also suffer from similar deleterious mutations, making them less efficient at using guanidine than environmental AOB. These considerations should be added in the respective discussion.

We agree with the reviewer that accumulation of mutations during long-term maintenance of the ammonia-oxidizing (or more general any microbial) cultures in the lab can cause many effects and thus decided to remove this purely speculative statement from the revised version of the manuscript.

The new data do not show how quantitatively important guanidine is for growth of *N. inopinata* in comparison to ammonia. Growth rates for both substrates should be added.

Done as suggested (clean manuscript version, L 184-185, Supplementary Table 6). Please note also that, while the growth rates on guanidine were lower than on ammonium, the cell yield (the number of cells formed per mol of combined nitrite and nitrate produced) did not differ between treatments receiving either guanidine or ammonium, or both substrates.

The detection limit for guanidine concentrations in the soils is not clear. All soil additions are given in units of $\mu\text{g-N g}^{-1}$ dry weight soil, whereas the detection limit is given as <50 nM. Please make these units in the legend of figure 6 consistent.

Many thanks for spotting this. The legend of Figure 6 has been adjusted accordingly.

New Ext data fig. 3. Based on the rest of the paper, panel g on this figure is rather surprising. It seems to show that the addition of guanidine in fact reduces the total oxidation rate. If this interpretation is correct, it needs to be addressed in the manuscript.

This interpretation was already mentioned in the previous version of the manuscript (previous manuscript L 187-188; now 186-187).

New data has been added showing that guanidine is present in animal urine and feces, as well as at the Klosterneuburg WWTP influent; but at extremely low concentrations, and this should be acknowledged or at least compared to ammonium concentrations in the respective samples/environments.

We now mention that the detected guanidine concentrations in the urine and fecal samples were low.

Abstract L34: “*N. inopinata* and likely most other comammox microbes grow on guanidine as the sole source of energy, reductant, and nitrogen”. As it is, the sentence implies that this is their only metabolism, please adjust to clarify that they can also use ammonium. Change as follows; “*N. inopinata* and likely most other comammox microbes can grow on guanidine as the sole source of energy, reductant, and nitrogen”.

Done as suggested.

Referee #3 (Remarks to the Author):

The revisions to the manuscript adequately addressed my prior concerns. The authors have better emphasized the significance of their work, shortened appropriately some sections, and enhanced the quality of several studies. The demonstration that the two metal ions per subunit in crystallized guanidinase include one Ni and one Mn is fascinating, although this enzyme was obtained from recombinant *E. coli* and so may not be identical to the situation for enzyme isolated from the original microorganism.

Many thanks for these very positive comments.

Reviewer Reports on the Second Revision:

Referees' comments:

Referee #2 (Remarks to the Author):

I am satisfied with how the authors have addressed my remaining concerns. The authors have removed the statement that guanidine use provides a unique niche for comammox in the environment. Additionally, the authors have now added the growth rates for *N. inopinata* on ammonia and guanidine, respectively. They have also removed the section on the potential impact of deleterious mutations.

Moreover, in the revised manuscript, the discussion on whether guanidine can be used to select for comammox in agricultural fields has been toned down and instead it is highlighted that ammonia-oxidizing microbes in general might be responsible for guanidine degradation in soils.

This is a great piece of work, and I look forward to future studies on this metabolism in ammonia oxidizers, both in culture and the environment.

Author Rebuttals to Second Revision:

Referees' comments:

Referee #2 (Remarks to the Author):

I am satisfied with how the authors have addressed my remaining concerns. The authors have removed the statement that guanidine use provides a unique niche for comammox in the environment. Additionally, the authors have now added the growth rates for *N. inopinata* on ammonia and guanidine, respectively. They have also removed the section on the potential impact of deleterious mutations.

Moreover, in the revised manuscript, the discussion on whether guanidine can be used to select for comammox in agricultural fields has been toned down and instead it is highlighted that ammonia-oxidizing microbes in general might be responsible for guanidine degradation in soils.

This is a great piece of work, and I look forward to future studies on this metabolism in ammonia oxidizers, both in culture and the environment.

Many thanks for the very positive and encouraging words!